# Domain-substituted IGF2 tag modulates targeting of lentiviral gene therapy for Hunter syndrome

Fabio Catalano[1,2,3], Dejan Stevic [ID][1,2,3,4], Giacomo Zundo[1,2,3,4], Tessa F Huizer[1,2,3], Zina Dammou[1,2,3],
Eva C Vlaar[1,2,3], Drosos Katsavelis [ID][1,2,3], Jeroen C van den Bosch[1,3], Hannerieke J M P van den Hout[2,3],
Esmeralda Oussoren[2,3], Ans T van der Ploeg[2,3], George J G Ruijter[1,3], Gerben Schaaf[1,2,3] &
W W M Pim Pijnappel [ID][1,2,3][✉]

## Abstract

**We present the SWAP design, a novel, structurally cohesive IGF2-based tag for modular receptor targeting during gene therapy for lysosomal storage disorders (LSDs). We found that IGF2's central loop is critical for high-affinity binding to the insulin receptor (IR) and IGF1 receptor (IGF1R)—both involved in glucose homeostasis—but is not required for interaction with the cation independent mannose 6-phosphate/IGF2 receptor (CI-M6P/IGF2R)—a key target for lysosomal delivery. This formed the basis for designing the Substitution of the central-loop With Augmenting Peptides (SWAP) tag. By replacing the central loop with alternative epitopes, SWAP ensures high-affinity multimodal receptor targeting while maintaining structural integrity. In vivo, lentiviral gene therapy employing IDS fused to SWAP variants containing ApoE and RAP12x2 inserts corrected Hunter disease pathology across multiple tissues, including liver, spleen, heart, bone, and brain, matching the efficacy of the traditional IGF2 tag. These findings position SWAP as a novel and effective tag design for IGF2-based therapeutics with a more favourable ligand–receptor interaction.**

**Keywords** Gene Therapy; IGF; IGF1 Receptor; Insulin Receptor; Lysosomal Storage Diseases
**Subject Categories** Genetics, Gene Therapy & Genetic Disease; Methods & Resources; Organelles

## Introduction

Lysosomal storage disorders (LSDs) are monogenic inherited conditions caused by defects in lysosomal function. The collective incidence of LSDs is approximately 1 in 5000 living newborns, with manifestations affecting multiple tissues and organs (Kido et al, 2023). Gene therapy is a promising one-time treatment option

(Kido et al, 2023; Sevin and Deiva, 2021; Massaro et al, 2021), though it may be limited in efficacy by suboptimal delivery of therapeutic enzymes to critical tissues (Miwa et al, 2020; Smith et al, 2022; Liang et al, 2022a; Gleitz et al, 2018; Concolino et al, 2018; Vollebregt et al, 2022). To address this limitation, we previously tested insulin-like growth factor type 2 (IGF2) tagging during hematopoietic stem and progenitor cell-mediated lentiviral gene therapy (HSPC-LVGT). IGF2 tagging confers high-affinity binding to the cation-independent mannose 6-phosphate/insulin-like growth factor type 2 receptor (CI-M6P/IGF2R), resulting in enhanced delivery to the lysosome of HSPC-produced therapeutic enzymes. Using this strategy, we observed a dose-dependent enhancement of therapeutic efficacy of HSPC-LVGT upon IGF2 fusion to GAA (Pompe disease; Liang et al, 2022a) or IDS (Hunter disease; Catalano et al, 2023), with the most significant effects observed in the brain and skeletal muscles (Liang et al, 2022a; Catalano et al, 2023, 2024; Liang et al, 2024). These results have been reproduced by others, both in the context of HSPC-LVGT (Dogan et al, 2022; Yoon et al, 2024) and during AAV gene therapy (Meena et al, 2023), effectively illustrating that IGF2-tagging could be a general strategy to improve therapeutic outcomes of gene therapies for LSDs.

Together with insulin and IGF1, IGF2 is a key component of the Insulin-like Growth Factor (IGF) system, which plays a vital role in growth, development, and metabolism (Blyth et al, 2020). The "Insulin-like" attribute was assigned to IGF2 in 1976, when Rinderknecht and Humbel observed its remarkable structural similarity to pro-insulin (Miller et al, 2022). In fact, like pro-insulin, IGF2's B domain (amino acids (AA) 1–32) and A domain (AA 41–61) form three alpha helices held together by three disulfide bonds, while domain C (AA 33–40) and domain D (AA 62–67) fold into a random-coil central loop and an N-terminal tail, respectively (Blyth et al, 2020). This structural organization is shared also by IGF1, and is accompanied by a high degree of homology in the helical cores of all these ligands (Blyth et al, 2020). Given these similarities, it is not surprising that IGF2, aside from binding to the CI-M6P/IGF2 receptor, also binds with high affinity to both insulin receptor isoform A (IR-A) and the IGF1 receptor

[1]Department of Clinical Genetics, Erasmus MC University Medical Center, Rotterdam 3015GE, The Netherlands. [2]Department of Pediatrics, Erasmus MC University Medical Center, Rotterdam 3015GE, The Netherlands. [3]Center for Lysosomal and Metabolic Diseases, Erasmus MC University Medical Center, Rotterdam 3015GE, The Netherlands. [4]These authors contributed equally: Dejan Stevic, Giacomo Zundo. ✉E-mail: w.pijnappel@erasmusmc.nl

(IGF1R) (Blyth et al, 2020). As a result, concerns arise that the use of IGF2-based therapeutics could lead to unintended activation of the IR-A and IGF1R, potentially triggering both metabolic and mitogenic pathways. An example of this concern is the clinical trial assessing the safety of reveglucosidase alpha—a GAA analog fused with IGF2 and administered intravenously as enzyme replacement therapy (ERT) for late-onset Pompe disease patients—which documented dose-dependent hypoglycemic events shortly after drug infusion. This effect was attributed to IGF2's ability to bind the insulin receptor (IR) (Byrne et al, 2017). The ability of IGF2 to engage IR and IGF1R is primarily mediated by interactions involving its B and A domains, which are also crucial for high-affinity binding to CI-M6P/IGF2R (Frago et al, 2016). However, IGF2 binding to IR-A and IGF1R is also mediated by its central loop (Xu et al, 2020; An et al, 2024), which is not directly involved in binding to CI-M6P/IGF2R (Brown et al, 2008; Frago et al, 2016). In fact, the central loop residues R37 and R38 have been observed to interact with IR-A in structural and mutagenesis studies (An et al, 2024), while residues R30 and R38 have been observed to interact with the IGF1R in structural studies (Xu et al, 2020). This makes IGF2's central loop a preferential engineering target to modulate receptor binding of the IGF2 tag, while preserving its interaction with the CI-M6P/IGF2R.

With the goal of developing an IGF2 tag with a more favourable ligand–receptor interaction profile, we assessed the role of IGF2's central loop in binding to IR-A and IGF1R, and we built on this to generate a new variant called SWAP (Substitution of the central-loop With Augmenting Peptides), characterized by the replacement of the central loop of IGF2 with ligands for other receptors (ApoE and Rap12x2). SWAP retained high affinity for the CI-M6P/IGF2 receptor and now effectively also engaged LRP-1, a blood-brain barrier receptor, while showing complete loss of affinity for IR and reduced affinity for IGF1R depending on the inserted epitope (only SWAP-RAP12x2 exhibited reduced binding to IGF1R). When tested in a Mucopolysaccharidosis type II (MPS II) murine model during HSPC-LVGT using pCCL.MND lentiviral vectors (Catalano et al, 2023, 2024), tagging of IDS (Iduronate 2-sulfatase, defective in MPS II) with SWAP variants resulted in a therapeutic outcome that matched unmodified IDS.IGF2, trended superior compared to central loop-deleted IGF2 (IGF2del), and outperformed untagged IDS. This supports SWAP as an alternative to existing IGF2 tags offering modular targeting of receptors and a more favourable ligand–receptor interaction profile that has significant potential to advance gene therapy for lysosomal storage disorders.

## Results

### The central loop of IGF2 can be deleted or replaced to form a modular, domain-substituted IGF2 tag (SWAP tag)

With the goal of reducing binding to the IR and IGF1R while maintaining high-affinity binding to the CI-M6P/IGF2R, we first investigated whether the central loop of IGF2 is essential for its binding to the CI-M6P/IGF2R. To this end, we tested a previously optimized variant of IGF2 with a deletion of residues 30–40, corresponding to the removal of domain C and a partial removal of domains B and A of IGF2 (LeBowitz & Maga, 2012). We show here

using structural modelling that this results in a complete deletion of the central loop of IGF2 (Fig. 1A,B). We referred to this version of IGF2 as IGF2del (Fig. 1B). To assess its functionality, we fused IGF2del to the C-terminus of Iduronate 2-sulfatase (IDS; IDS.IGF2del) and evaluated whether IDS.IGF2del could bind to the CI-M6P/IGF2R using a direct ELISA, as we previously showed for IDS.IGF2 (Catalano et al, 2023). In this assay, different concentrations of IDS, IDS.IGF2, or IDS.IGF2del were incubated with immobilized domain 11–13, corresponding to the IGF2 binding domain of CI-M6P/IGF2R (Catalano et al, 2023), while ligand binding was measured using an anti-IDS antibody. As expected, no appreciable binding of IDS to the CI-M6P/IGF2R was observed, while IDS.IGF2 and IDS.IGF2del bound efficiently, though IDS.IGF2del showed a slight decrease in binding affinity (Fig. EV1A).

We next tested the binding of IDS.IGF2del to the IR-A via competitive ELISA, where immobilized IR-A was incubated with a fixed concentration of biotinylated insulin in combination with varying concentrations of non-biotinylated ligands. Untagged IDS or IGF2del-tagged IDS failed to compete with insulin for binding to IR-A, whereas IDS.IGF2 competed moderately, and IGF2 peptide strongly (Fig. EV1B). These results demonstrated that the deletion of IGF2 AA 30-40 results in the loss of binding to the IR-A, while preserving binding to the CI-M6P/IGF2R.

Based on these results, we explored whether exogenous epitopes could replace the central loop of IGF2, potentially conferring binding to new, clinically relevant receptors, while abolishing binding to IR-A and retaining binding to CI-M6P/IGF2R. We therefore substituted residues 29–41 of IGF2— corresponding to a partial deletion of domain B and A and a complete deletion of domain C—with exogenous epitopes able to bind to receptors other than those normally bound by IGF2 (Fig. 1A,B). In this design, residues 29 and 41 of IGF2—a serine and a glycine, respectively, which were present at both ends of the deletion in the IGF2del—are replaced with a cysteine, while the inserted epitopes are flanked by flexible linkers (Fig. 1A,B). These modifications were based on the prediction that the inclusion of cysteines at the sides of the insertion would form a disulfide bond, helping to stabilize the helical core of IGF2 and mimicking the structural arrangement of IGF2del, while the flexible linker would ensure the correct exposure of the inserted epitopes (Fig. 1A). In line with this prediction, when this IGF2 variant was fused to IDS, the inclusion of a cysteine at both ends of an ApoE (Croy et al, 2004) insert (IDS.IGF2-central loop substituted-ApoE_with Cys; ApoE AA sequence: LRKLRKRLL) provided an advantage in an uptake assay compared to the same design without cysteine (IDS.IGF2-central loop substituted-ApoE_No Cys). We refer to this design as Substitution of the central-loop With Augmenting Peptides (SWAP) (Fig. EV1C). It is characterized by an "SGGG" linker at the N-terminus of the insertion, an "SGGGGSG" linker at the C-terminus of the insertion, and a cysteine residue at both ends of the insertion (Fig. 1A,B).

We next tested SWAP variants with ApoE (9 AA) and RAP12x2 (25 AA) inserts. These inserts were chosen for their documented ability to undergo transcytosis across the blood-brain barrier via engagement of receptors such as LRP-1 (Catalano et al, 2023; Gleitz et al, 2018; Pflanzner et al, 2011). We fused these SWAP variants to IDS (IDS.SWAP-ApoE, IDS.SWAP-RAP12x2) and proceeded to test their functionality. First, we produced lentiviral vectors

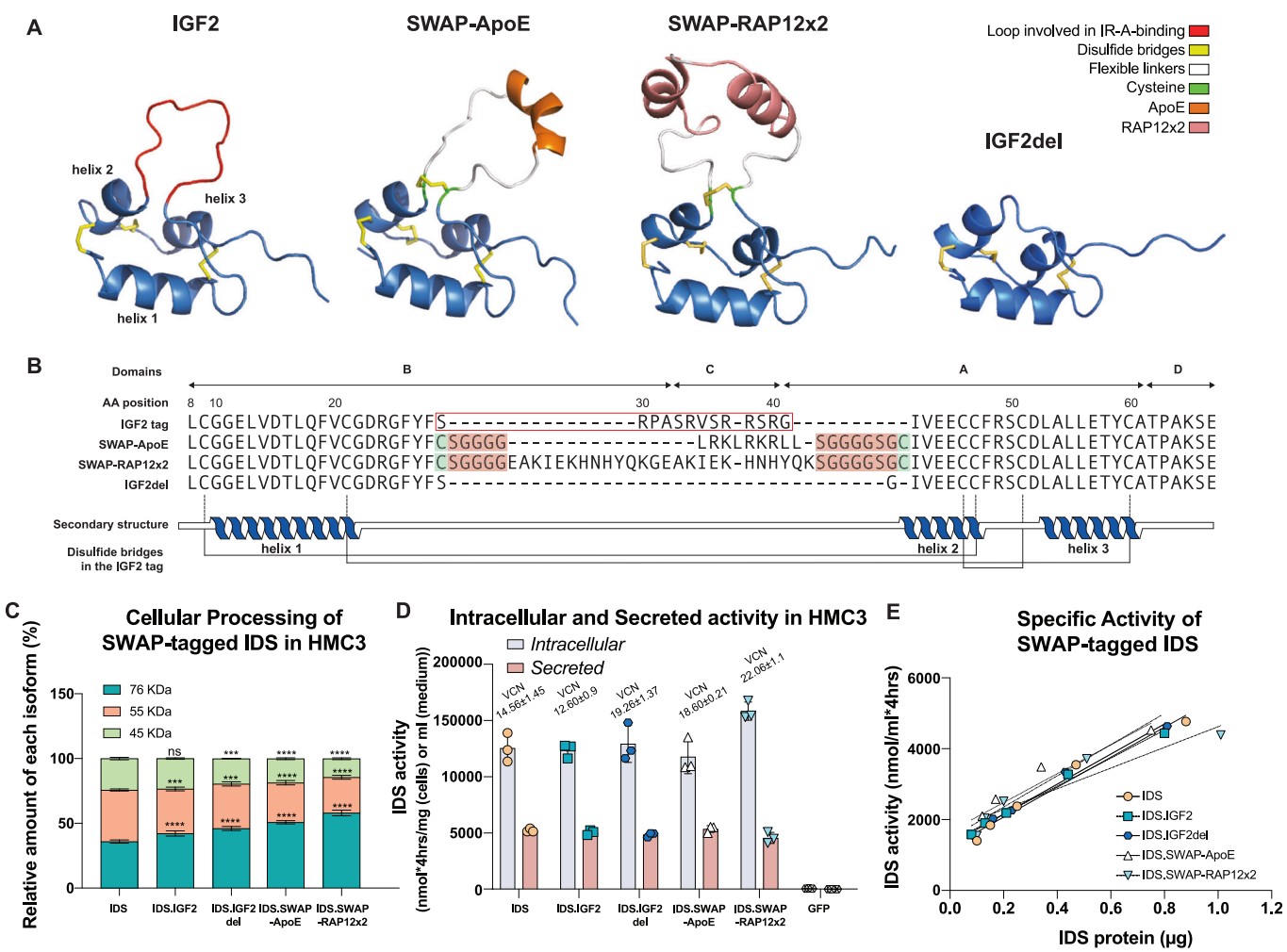

**Figure 1. The central loop of IGF2 can be replaced by epitopes to form a modular, domain-substituted IGF2-based SWAP tag.**

(A) Structure prediction using AlphaFold (Abramson et al, 2024) of IGF2 peptide (left), SWAP with ApoE and RAP12x2 inserts, and central loop-deleted IGF2 peptide (SWAP-ApoE, SWAP-RAP12x2, IGF2del, respectively). Note that the inserted epitopes are predicted to be exposed on the surface without disrupting the native IGF2 tag structure. IGF2 sequence, linkers, inserts, helices and disulfide bonds are indicated. (B) Alignment of sequences from (A). The IGF2 peptide consists of AA 1, 8–67 of mature human IGF2. In the SWAPs, AA 29–41 of the IGF2 peptide (boxed in red) are substituted with AA 141–149 of mature human Apolipoprotein E (ApoE) flanked by flexible linkers, or AA 251–262 of the human receptor associated protein (RAP) in a glycine-spaced tandem repeat (RAP12x2), flanked by flexible linkers (flexible linkers are highlighted in white (A) and in red in (B)). Note that the two cysteine residues located at the edges of the insertion (marked in green in (A) and highlighted in green (B)) are predicted to form a disulfide bridge (highlighted in yellow in (A)). (C) Quantification of intracellular processing of tagged IDS proteins. HMC3 cells were transduced followed by immunoblot analysis of IDS protein 10 days after transduction. See Fig. EV1D for immunoblot analysis. Results are expressed as percentage of the total amount of IDS protein. Significance versus IDS is shown. The adjusted $P$ values were as follows: *76 KDa*, IDS vs. IDS.IGF2 $P = 0.000014$, IDS vs. IDS.SWAP-ApoE $P = <0.000001$, IDS vs. IDS.SWAP-RAP12x2 $P = <0.000001$, IDS vs. IDS.IGF2del $P = <0.000001$. *55 KDa*, IDS vs. IDS.IGF2 $P = 0.000105$, IDS vs. IDS.IGF2del $P = 0.000121$, IDS vs. IDS.SWAP-ApoE $P = <0.000001$, IDS vs. IDS.SWAP-RAP12x2 $P = <0.000001$. 45 KDa, IDS vs. IDS.IGF2del $P = 0.000696$, IDS vs. IDS.SWAP-ApoE $P = 0.000048$, IDS vs. IDS.SWAP-RAP12x2 $P = <0.000001$. (D) As (C), but now assayed for intracellular and secreted IDS enzyme activity. Lentivirus expressing GFP was included as control. Average Vector Copy Number (VCN) is shown. The adjusted $P$ values were >0.9999, except for IDS vs. IDS.SWAP-RAP12x2 ($P = 0.0651$), and IDS vs GFP ($P = <0.0001$). (E) Specific activities of tagged IDS proteins. See also Fig. EV1F and Table EV1. Data information: data represent means ± SD and were analyzed by two-way ANOVA followed by Bonferroni's multiple testing correction (C, D) without interaction (D), or by regression analysis (E). In (C, D), a total of 6 (C) and 7 (D) conditions were analyzed; significant pairwise differences among the 5 (C) and 6 (D) relevant conditions are reported. In (C, D) comparisons vs. IDS were performed. (C–E) $n = 3$ biological replicates per condition. Significant comparisons are indicated by brackets. ***$P \leq 0.001$; ****$P \leq 0.0001$. Source data are available online for this figure.

encoding IDS.SWAPs and controls, and transduced a human microglia cell line (HMC3). Then, we assessed the effect of SWAP tagging on IDS processing by performing SDS-PAGE analysis of the cell lysate and medium supernatant of transduced HMC3 cells 10 days post-transduction (Figs. 1C and EV1D; total protein load is shown in Fig. EV1E). In cells, untagged IDS was present as a

precursor protein of apparent molecular weight of 76 KDa, while tagged version of IDS resulted in a precursor with a noticeably higher apparent molecular weight of ~84 KDa (IDS.IGF2 and IDS.SWAP versions) or ~80 KDa (IDS.IGF2del) (Fig. EV1D, top panel). All the examined proteins underwent processing into the mature forms (apparent molecular weight of 55 KDa and 45 KDa).

Importantly, all tagged IDS variants exhibited a significant but modest skew toward the precursor isoform, with the degree of skew increasing from IDS.IGF2 < IDS.IGF2del <IDS.SWAP-ApoE <IDS.SWAP-RAP12x2 (Figs. 1C and EV1D, top panel), suggesting that the tagging strategy might partially interfere with the cleavage event occurring at the C-terminus of IDS between amino acids 455 and 456 (Millat et al, 1997; Froissart et al, 1995). In medium supernatant, we observed the precursor form of IDS at the same apparent molecular weight as the precursor form observed intracellularly (Fig. EV1D, lower panel). Intracellular and secreted protein levels documented in the SDS-PAGE analysis correlated with the IDS enzyme activity levels observed in the same samples (Fig. 1D). To assess secretion in more details, we transduced HMC3 cells at MOI 5 and 50, and we measured IDS activity in the medium supernatant after 24 h (Fig. EV1G). IDS activity per VCN was consistent across the conditions, with only a slight reduction for IDS.SWAP-ApoE at low MOI (Fig. EV1G). These findings demonstrate that the SWAP design does not interfere with intracellular or secreted IDS activity, although it might slightly affect intracellular processing. We next measured specific activity of IDS tagged with SWAP variants—carried out by quantifying the protein levels and enzyme activity levels in medium supernatant (only precursor protein) obtained from a 24 h culture of transduced HMC3 cells—which confirmed that SWAP tagging does not affect specific activity of IDS (Fig. 1E; Table EV1).

These results demonstrate that, while the central loop of IGF2 is essential for binding to the insulin receptor, it is not for binding to the CI-M6P/IGF2R and can be replaced by an insertion/deletion configuration to form an epitope tag that can be fused to IDS protein without affecting its function.

## SWAP variants display a modified ligand–receptor interaction profile

To investigate binding of SWAP variants to the CI-M6P/IGF2R, we performed a competitive binding ELISA as described above, but now with immobilized domain 11–13 of the CI-M6P/IGF2R and a fixed concentration of biotinylated IGF2 in combination with varying concentrations of non-biotinylated ligands. Whereas IDS failed to compete with IGF2 for binding, IDS.IGF2 and IDS.IGF2-del competed with similar efficacy with $IC_{50}$ values ranging from 35 nM to 60 nM, although IGF2 peptide was the most efficient competitor (Fig. 2A; Table EV1), suggesting that binding of IGF2 to the CI-M6P/IGF2R is partially inhibited when tagged to IDS. IDS.IGF2del displayed a slightly lower affinity than IDS.IGF2, as previously shown (LeBowitz & Maga, 2012), with an $IC_{50}$ of about 50 nM (Fig. 2A; Table EV1). IDS.SWAP variants exhibited similar or slightly higher binding affinities than IDS.IGF2, with $IC_{50}$ values around 30 nM for IDS.SWAP-ApoE and IDS.SWAP-RAP12x2 (Fig. 2A; Table EV1). Importantly, binding of IGF2 to the CI-M6P/IGF2R differed when used as a peptide or when fused to IDS, with IDS.IGF2 showing a 7–10 times increased $IC_{50}$ values compared to the untagged IGF2 peptide (~5 nM; Fig. 2A; Table EV1).

None of the tested IDS.SWAP variants, as well as untagged IDS and IDS.IGF2del, exhibited an appreciable binding to the IR-A in a binding assay as described above, as evidenced by the near-complete lack of competition against biotinylated insulin (Figs. 2B and EV2A; Table EV1). In contrast, IDS.IGF2 bound the IR-A with an $IC_{50}$ of ~250 nM (Fig. 2B; Table EV1). Also in this case, we observed an impact of tagging on binding of IGF2 to the IR-A. Specifically, IDS.IGF2 bound the IR-A with an affinity that was ~65–93-fold lower than the affinity of untagged IGF2 peptide (Table EV1), suggesting a partial hindrance of IGF2 binding to IR-A when tagged to IDS. We also tested binding to the IR-A of IGF2.GAA, the same fusion protein used in the reveglucosidase alpha trial and that caused transient hypoglycemia events after bolus infusions (Byrne et al, 2017). IGF2.GAA was able to bind the IR-A with an affinity that was ~24–34-fold higher compared to IDS.IGF2, and only 2.69 lower than untagged IGF2 peptide (Fig. EV2B; Table EV1). This shows that the IGF2.GAA—fusion at the N-terminus of IGF2—but not IDS.IGF2—fusion at the C-terminus of IGF2—retains significant binding affinity to IR-A. Overall, these findings highlight the pivotal role of the central loop of IGF2 in binding to IR-A, a role that cannot be replaced by the epitopes inserted in the SWAP design (ApoE, RAP12x2). As a result, the SWAP modifications significantly disrupt IGF2's ability to bind the IR-A.

To assess binding to the IGF1R, we performed a competitive ELISA assay using immobilized IGF1R and biotinylated IGF1 peptide (Fig. 2C; Table EV1). In this assay, untagged IDS and IDS.IGF2del exhibited no significant binding to the IGF1R compared to IDS.IGF2. In contrast, IDS.SWAP-ApoE bound to the IGF1R with an affinity similar to that of IDS.IGF2, while IDS.SWAP-RAP12x2 bound with an affinity that was ~3.5 times lower compared to IDS.IGF2. Also in this case, IDS.IGF2 showed an affinity for the IGF1R that was ~6 times lower compared to untagged IGF2 peptide. These findings demonstrate for the first time that deletion of amino acids 30–40 of IGF2 disrupts IGF1R binding, highlighting the critical role of the central loop in this interaction, but they also show that replacing amino acids 30–40 with epitopes can also result in retainment of IGF1R affinity (in the case of IDS.SWAP-ApoE). There are several scenarios that may explain why SWAP variants showed some affinity for the IGF1R, including either the presence of IGF1R binding motifs within the SWAP sequences, and/or a structural, sequence-independent requirement of a central loop structure in IGF2 and IGF2-derived tags for binding to the IGF1R, as further elaborated on in the "Discussion".

The ApoE (Croy et al, 2004) and RAP12 (Ruan et al, 2018) peptides have been reported to bind to cluster IV of the LRP-1 receptor. To investigate whether these peptides are functional while inserted within SWAP, we performed a direct ELISA assay with immobilized LRP-1 (Figs. 2D and EV2C,D). IDS.IGF2 and IDS.IGF2del bound LRP-1 with low affinity, while untagged IDS showed no binding (Table EV1). In contrast, insertion of ApoE and RAP12x2 caused a strong increase in the affinity for the LRP-1, with apparent $K_d$ of either 25.63 nM (IDS.SWAP-ApoE) or 38.08 nM (IDS.SWAP-RAP12x2) (Table EV1). In separate LRP-1 ELISAs assay, we compared these two SWAP variants to IDS.ApoEII (tandem repeat of the ApoE sequence "LRKLRKRLL") and IDS.RAP12x2 (Fig. EV2C,D) (Catalano et al, 2023, 2024; Gleitz et al, 2018). All the IDS.SWAP variants showed apparent $K_d$ values that were comparable to those of IDS.ApoEII (Fig. EV2C,D; Table EV1), while IDS.RAP12x2 showed no binding to LRP-1 (Fig. EV2D), as previously shown (Catalano et al, 2023). These findings demonstrate that, when incorporated into SWAP, ApoE binds to LRP-1 with an affinity comparable to that of its tandem-repeat variant (ApoEII). Importantly, these data also show that

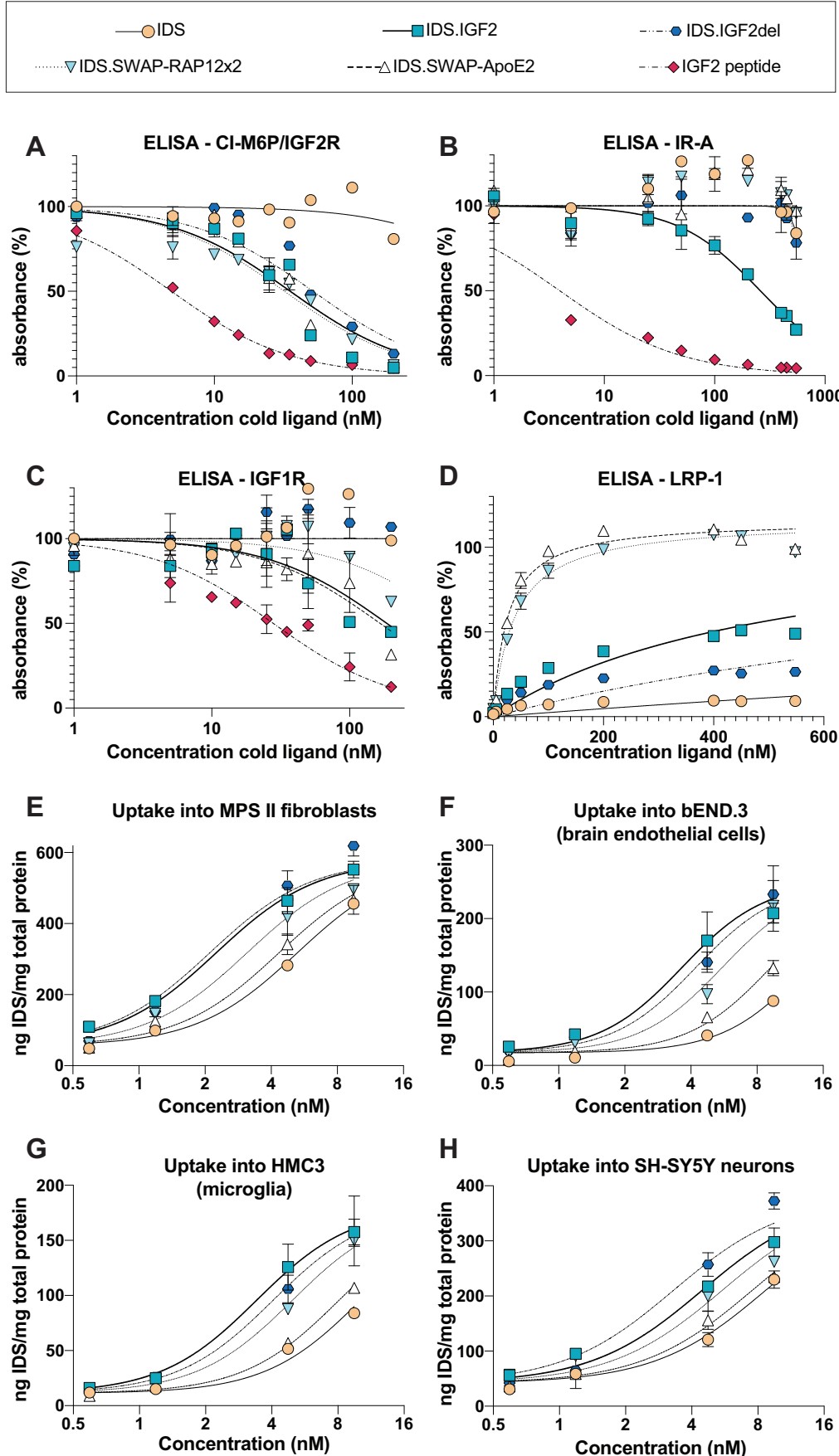

◀

**Figure 2. Ligand-receptor binding profiles of IDS.SWAP proteins.**

(A–C) Competitive IGF2R ELISA using biotinylated IGF2 (A), IR-A ELISA using biotinylated insulin (B) and IGF1R ELISA using biotinylated IGF1 (C) with the indicated competing ligands at the indicated concentrations. (D) Direct LRP-1 ELISA with the indicated ligands at the indicated concentrations. $IC_{50}$ (A–C) or $K_d$ (D) affinity values are shown in Table EV1. (E–H) Uptake into MPS II fibroblasts (E), bEND.3 cells (F), HMC3 (G), or SH-SY5Y cells (H) over 24 h at the indicated input concentrations. Data information: data represent means ± SD. Regression analyses of (A–H) are shown in Table EV1. In (A–D), $n = 2$ technical replicates per condition; in (E–H) $n = 3$ biological replicates per condition. Source data are available online for this figure.

IDS.SWAP-RAP12x2, but not IDS.RAP12x2, can effectively engage LRP-1. This is elaborated upon in the "Discussion".

Next, we assessed uptake of IDS.SWAP versions into MPS II fibroblasts after 24 h incubation in conditioned media generated by transduction of HMC3 cells (Fig. 2E). All the IDS-tagged versions exhibited $EC_{50}$ at lower values compared to untagged IDS, which resulted in increased uptake across the range of concentrations tested. Specifically, IDS.IGF2 and IDS.IGF2del uptake resulted in $EC_{50}$ at values that were 2.3 and 2.4 times lower than those of untagged IDS, respectively, while uptake of IDS.SWAP-ApoE and IDS.SWAP-RAP12x2 showed $EC_{50}$ at values that were 1.2 and 1.5 times lower than untagged IDS (Table EV1). We also assessed uptake of IDS.SWAP variants into a murine brain endothelial cell line (bEND.3), a human microglia cell line (HMC3), and SH-SY5Y cells differentiated into neurons (Fig. 2F–H). In these cell lines, IDS-tagged versions exhibited increased cellular uptake across the range of concentrations tested. Specifically, $EC_{50}$ values were 3.9, 1.4, 2.4, and 3.1 times lower than untagged IDS in bEND.3 cells for IDS.IGF2, IDS.SWAP-ApoE, IDS.SWAP-RAP12x2, and IDS.IGF2-del, respectively (Fig. 2F). In HMC3, $EC_{50}$ values were 3.0, 1.3, 2.0, and 2.5 times lower than untagged IDS for IDS.IGF2, IDS.SWAP-ApoE, IDS.SWAP-RAP12x2, and IDS.IGF2del, respectively (Fig. 2G). In SH-SY5Y, $EC_{50}$ values were 1.9, 1.1, 1.6, and 2.6 times lower than untagged IDS for IDS.IGF2, IDS.SWAP-ApoE, IDS.SWAP-RAP12x2, and IDS.IGF2del, respectively (Fig. 2H). Overall, IDS-SWAP variants mediated enhanced uptake compared to untagged IDS across various cell types at levels that varied depending on the inserted epitope. IDS.SWAP-ApoE variant showed the lowest enhancement of uptake levels over 24 h, similar to what we previously observed for IDS.ApoEII (Catalano et al, 2023).

These data show that the SWAP design decreases IGF2's binding affinity for IR-A and reduces binding to IGF1R depending on the inserted epitope, with only IDS.SWAP-RAP12x2 showing lower binding affinity for IGF1R. Importantly, the SWAP design maintained high-affinity binding to the CI-M6P/IGF2R and conferred binding to an additional receptor targeted by the inserted epitope, resulting in increased cellular uptake compared to untagged IDS across various cell lines.

## Ex vivo lentiviral gene therapy with IDS.SWAP variants

Next, pCCL lentiviral vectors with transgene expression driven by the MND promoter (Catalano et al, 2023) were used to compare *IDS.SWAP-ApoEco*, *IDS.SWAP-RAP12x2co* and *IDS.IGF2del_co* with untagged *IDSco* and *IDS.IGF2co* during hematopoietic stem and progenitor cell (HSPC)-mediated lentiviral gene therapy (LVGT) for their efficacy in correction of pathology in a murine model for Hunter disease ($Ids^{y/-}$) (Chen et al, 1999; Garcia et al, 2007). We transplanted *CD45.1-Ids$^{y/-}$* HSPCs transduced with

lentiviral vectors expressing either the test transgenes or *GFP* as a control into 8–11 week-old *CD45.2-Ids$^{y/-}$* mice and analyzed pathology in disease-relevant tissues six months after transplantation (Fig. 3A). Flow cytometry analysis of chimerism in bone marrow revealed efficient engraftment of transplanted cells 6-months after transplantation, with chimerism values that were comparable across the conditions tested at values of ~80–95% (Fig. 3B). Similarly, VCN in bone marrow varied among the treatment groups and ranged from as low as 0.5 copies per genome to 5 copies per genome, with some groups showing slightly higher (*IDS.IGF2co* and *IDS.SWAp-ApoEco*) or slightly lower (*IDS.IGF2-del_co* and GFP) average values (Fig. 3C). Gene therapy with all *IDS*-containing vectors resulted in supraphysiological levels of IDS activity in bone marrow, WBC and plasma at levels several times above *Ids$^{y/-}$* and WT animals (Figs. 3D–G, EV3A–C). IDS activity in bone marrow was comparable across the treatment groups at values ~50–200-fold above WT (Figs. 3D and EV3A; Table EV1). To correct expression values for VCN in BM, we plotted all data points of this and our previous study (Catalano et al, 2023) and found a hyperbolic relationship, with saturation of average expression levels around VCN 2 (Fig. EV3D). Saturation of protein expression upon lentiviral transduction has been reported previously (Zielske et al, 2004; Larson et al, 2017) and may depend on experimental conditions. Using a hyperbolic relationship, IDS activities per VCN in BM were similar for all constructs tested (Fig. EV3E). In WBC, IDS activity of *IDSco*-treated animals was higher compared to activity measured in WBC of *IDS.IGF2co*, *IDS.IGF2del_co*, *IDS.SWAP-ApoEco*, *IDS.SWAP-RAP12x2co*-treated animals (Figs. 3E and EV3B). A similar pattern was observed in plasma (Figs. 3F and EV3C), with all the IGF2-containing versions displaying a lower plasma IDS activity compared to untagged IDS (this is further elaborated upon in the discussion). By plotting IDS activity in plasma against VCN in bone marrow, we observed a linear increase of IDS activity per VCN, with slope and intercept values that confirmed a tendency toward lower plasma activity levels for the vectors encoding tagged IDS variants compared to the *IDSco* vector (Fig. 3G; Table EV1). Specifically, *IDS.SWAP-ApoEco* resulted in values comparable to those observed after *IDS.IGF2co* gene therapy, while higher values were recorded for *IDS.IGF2-del_co*, and lower levels for *IDS.SWAP-RAP12x2co* (Fig. 3G; Table EV1). Slightly lower plasma protein levels after gene therapy with *IDS.SWAP-RAP12x2co* compared to *IDS.IGF2co* were confirmed by IDS ELISA (Fig. EV3F).

At harvest (8.5 months), *Ids$^{y/-}$* mice showed body weights that were ~8% lower compared to WT mice (*Ids$^{y/-}$*: 37.98 g; WT: 41.22 g), as previously reported (Fig. EV3G) (Garcia et al, 2007). After gene therapy, *GFP*-treated mice showed an average body weight at even lower levels compared to *Ids$^{y/-}$* mice (*GFP*:34.27 g), which was likely caused by the preconditioning treatment. Gene therapy with the therapeutic vectors resulted in body weights of

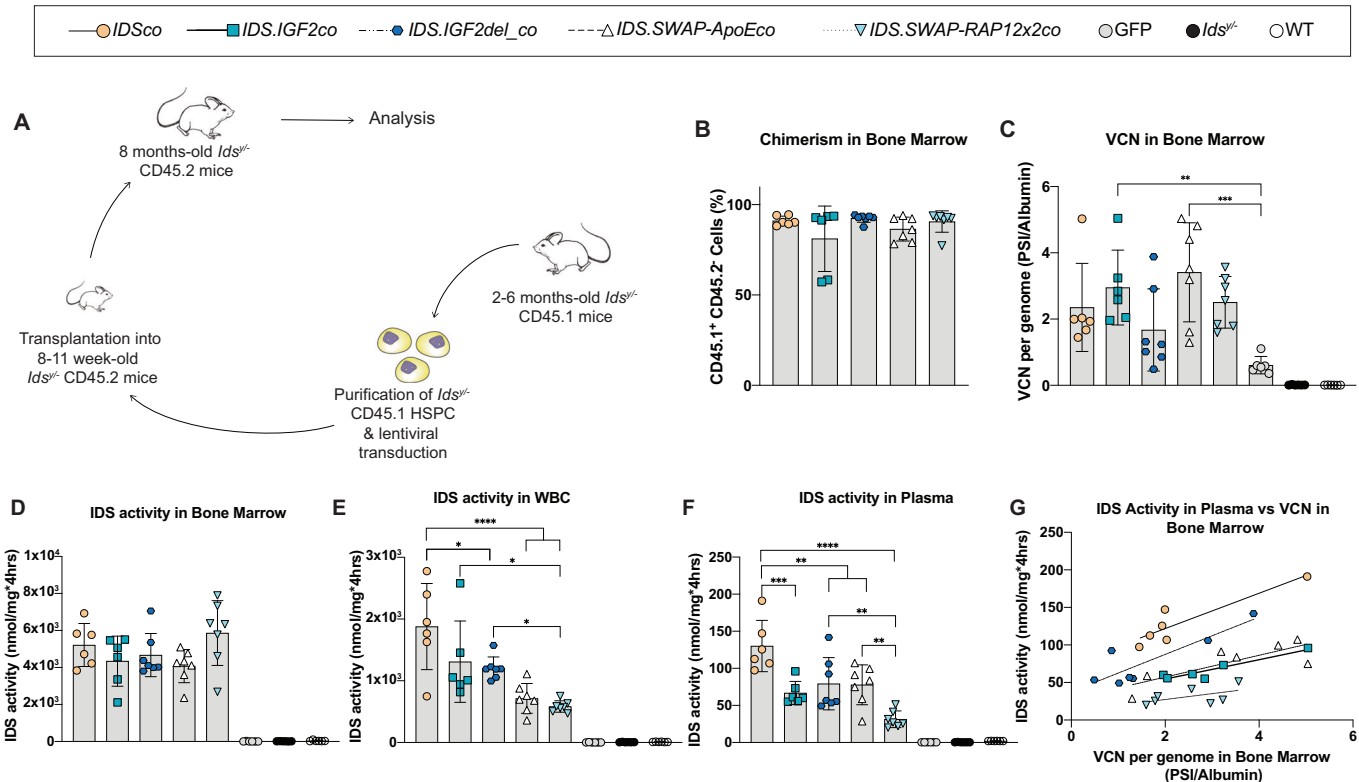

**Figure 3. Lentiviral gene therapy with IDS.SWAP variants results in supraphysiological transgene expression.**

(A) Cartoon of the experiment. 2-month-old *Ids*[y/−] mice were treated with HSPC-LVGT and analyzed at 8 months of age as previously described (Catalano et al, 2023, 2024). (B) Flow cytometry analysis of chimerism in bone marrow. Values are expressed as percentage of CD45.1[+]/ CD45.2[−] cells. (C) Analysis of VCN per genome in bone marrow by qPCR analysis on the *PSI* and *Albumin* loci. The adjusted *P* values were as follows: *IDS.IGF2co* vs. *GFP P* = 0.0048, *IDS.SWAP-ApoEco* vs. *GFP P* = 0.0002. (D–F) IDS enzyme activity in bone marrow (D), white blood cells (E, WBC), and plasma (F). In (D), all comparisons among the treatment groups showed *P* values > 0.9999, except for *IDS.IGF2co* vs. *IDS.SWAP-RAP12x2co* (*P* = 0.3553) and *IDS.SWAP-ApoEco* vs. *IDS.SWAP-RAP12x2co* (*P* = 0.0753). For (E, F), the adjusted *P* values were as follows: (E) *IDSco* vs. *IDS.SWAP-ApoEco P* = <0.0001, *IDSco* vs. *IDS.SWAP-RAP12x2co P* = <0.0001, *IDSco* vs. *IDS.IGF2del_co P* = 0.0317, *IDS.IGF2del_co* vs. *IDS.SWAP-RAP12x2co P* = 0.0492. (F) *IDSco* vs. *IDS.SWAP-RAP12x2co P* = <0.0001, *IDSco* vs. *IDS.IGF2del_co P* = 0.0030, *IDSco* vs. *IDS.SWAP-ApoEco P* = 0.0020, *IDSco* vs. *IDS.IGF2co P* = 0.0002, *IDS.IGF2del_co* vs. *IDS.SWAP-RAP12x2co P* = 0.0038, *IDS.SWAP-ApoEco* vs. *IDS.SWAP-RAP12x2co P* = 0.0058. (G) Relationship between VCN per genome in bone marrow and IDS enzyme activity in plasma. Regression analysis is shown in Table EV1. Data information: data represent means ± SD and were analyzed by one-way ANOVA followed by Bonferroni's multiple testing correction. *IDSco*, *GFP*, *Ids*[y/−] and WT *n* = 6; *IDS.IGF2del_co*, *IDS.SWAP-ApoEco* and *IDS.SWAP-RAP12x2co n* = 7. *\*P* ≤ 0.05; *\*\*P* ≤ 0.01; *\*\*\*P* ≤ 0.001; *\*\*\*\*P* ≤ 0.0001. Significant results are indicated by brackets. Source data are available online for this figure.

values raging from 36.78 g (*IDS.SWAP-RAP12x2co*) to 38.47 g (*IDSco*). The average body weight of animals treated with therapeutic vectors (37.6 g) was ~9% higher than that of *GFP*-treated animals, therefore showing a difference similar to that observed between *Ids*[y/−] and WT mice (Fig. EV3G). Post-prandial glucose levels in plasma were lower for *GFP*-treated and untreated *Ids*[y/−] animals compared to WT (Fig. EV3H). This could be caused by the lethargy that we observed in *GFP*-treated and untreated *Ids*[y/−] mice compared to WT, which could result in reduced feeding behavior. Importantly, treatment with the therapeutic vectors restored the postprandial plasma glucose at levels comparable to WT animals (Fig. EV3H). As IDS.IGF2 was the only fusion protein that could bind IR-A, albeit with low affinity, we tested whether gene therapy with the *IDS.IGF2co* vector could impact glucose homeostasis by performing a glucose tolerance test (GTT) in mice 6-month post transplantation (aged 8 months). GTT was performed in starved mice (6 h) by intraperitoneal injection of glucose and monitoring of the resulting glycemic levels in blood before and after glucose injection (Fig. EV3I). Overall, during the

GTT, glucose levels followed a similar curve across all conditions, with the *GFP*-treated group showing a faster return to basal levels compared to WT and *IDS.IGF2*-treated mice. At the 120 min time point, blood glucose levels returned to basal values for all groups tested. This indicates that HSPC-LVGT with *IDS.IGF2co* does not impact glucose tolerance under the conditions employed.

In conclusion, HSPC-LVGT with all the vectors tested resulted in supraphysiological expression in relevant hematological tissues, while plasma glucose levels were not affected.

## Gene therapy with IDS.SWAP variants corrects *Ids*[y/−] peripheral pathology

*GFP*-treated and untreated *Ids*[y/−] mice showed prominent pathology in liver and spleen characterized by increased alcian blue reactivity in sinusoidal structures (Fig. 4A). To assess correction of peripheral pathology using histology, three mice per treatment group were selected that had a similar VCN in BM to allow a fair comparison (Fig. EV4H). All tested gene therapy vectors resulted in

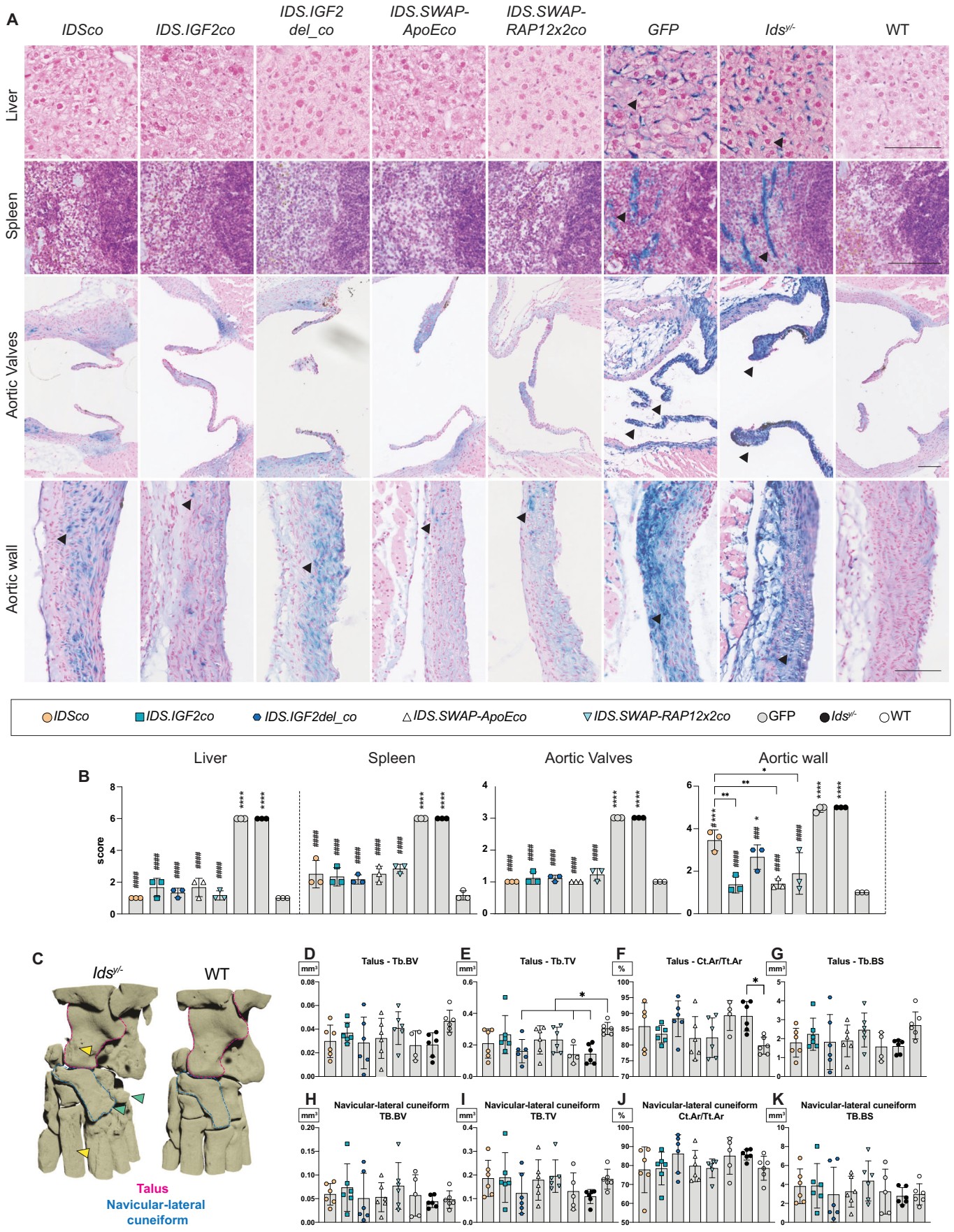

**Figure 4. Gene therapy with IDS.SWAP variants shows efficient correction of peripheral pathology.**

(A, B) Representative images of alcian blue staining of liver, spleen, aortic valves and aortic walls from gene therapy-treated mice and from controls. Arrowheads indicate alcian blue-positive cells. Scale bars = 100 μm. Scoring of alcian blue pathology is shown in (B). Scoring rules are shown in Table EV2. All significant comparisons in (B) showed $P$ values < 0.0001, except for: *Aortic Wall*, IDSco vs. IDS.IGF2co $P = 0.0016$, IDSco vs. IDS.SWAP-ApoEco $P = 0.0018$, IDSco vs. IDS.SWAP-RAP12x2co $P = 0.0243$, IDSco vs. WT $P = 0.0002$, IDS.IGF2del_co vs. WT $P = 0.0139$, IDSco vs. $Ids^{y/-}$ $P = 0.0264$, IDS.IGF2del_co vs. $Ids^{y/-}$ $P = 0.0004$. (C) Representative 3D renderings of reconstructed μCT images of the left hindlimb ankles are shown. Green arrowheads indicate osteophytes in untreated $Ids^{y/-}$ mice. Yellow arrowheads indicate bone surface features such as pores and irregular surface in untreated $Ids^{y/-}$ mice. (D–K) μCT quantification of bone microarchitecture of *talus* (D–G) and *navicular-lateral cuneiform* (H–K) in gene therapy-treated animals. Explanatory drawings are shown in Fig. EV4G. In (E, F), the adjusted $P$ values were as follows: (E) IDS.IGF2del_co vs. WT $P = 0.0297$, GFP vs. WT $P = 0.0241$, $Ids^{y/-}$ vs. WT $P = 0.0107$. (F) $Ids^{y/-}$ vs. WT $P = 0.0337$. Data information: data are presented as means ± SD. Data were analyzed by one-way ANOVA with Bonferroni's correction. In (D–K), comparisons were performed against WT. Asterisks (*) represent significance versus WT; hashes (#) represent significance versus $Ids^{y/-}$. Other significant comparisons are identified by brackets. In (A, B), $n = 3$; in (D–K) $n = 6$ for all groups except for GFP *navicular-lateral cuneiform* ($n = 5$) and GFP *talus* ($n = 4$). *$P \leq 0.05$; **$P \leq 0.01$; ***$P \leq 0.001$; ****$P \leq 0.0001$. #$P \leq 0.05$; ###$P \leq 0.001$; ####$P \leq 0.0001$. Source data are available online for this figure.

complete correction of alcian blue-related pathology in liver (Fig. 4A,B), and caused complete or near-complete reduction of alcian blue-related pathology in spleen (Fig. 4A,B). Strong alcian blue staining was present in aortic valves and the tunica media of the aortic wall of $Ids^{y/-}$ and GFP-treated mice (Fig. 4A,B), which was paralleled by increased levels of LAMP1 immunoreactivity in the same tissues (Fig. EV4A), as we previously reported (Catalano et al, 2024). Gene therapy with all the vectors tested caused a complete correction of alcian blue and LAMP1-related pathology in aortic valves (Figs. 4A,B and EV4A). Importantly, complete correction of pathology in aortic wall was achieved only after gene therapy with IDS.IGF2co and IDS.SWAP-ApoEco (Figs. 4A,B and EV4A). Among the other treatment groups, IDS.SWAP-RAP12x2co gene therapy caused a nearly complete correction, while significant residual pathology was observed after IDS.IGF2del_co and IDSco gene therapy (Figs. 4A,B and EV4A).

$Ids^{y/-}$ mice displayed a partially penetrant phenotype of the ankle that was characterized by swelling of the joint and limited mobility. This phenotype was previously characterized by micro-computed tomography (μCT) analysis in the original study describing this $Ids^{y/-}$ mouse model (Garcia et al, 2007). To analyse this in more detail, we performed μCT of the ankle bones of gene therapy-treated and control mice. μCT scans revealed the presence of abnormal bone surface features in $Ids^{y/-}$ mice compared to WT mice (Fig. 4C; yellow arrowheads indicate surface pores; green arrowheads indicate osteophytes). We also conducted a bone microarchitecture analysis of *talus*, *navicular-lateral cuneiform* and *medial cuneiform* (Figs. 4D–K and EV4B–F; explanatory drawings for the μCT parameters tested are shown in Fig. EV4G). In *talus* of $Ids^{y/-}$ animals, we observed a tendency toward a decreased trabecular bone volume (Tb.BV), with a 43% reduction compared to WT animals (Fig. 4D), while a significant 53% decrease of the trabecular tissue volume (Tb.TV) was observed in $Ids^{y/-}$ mice compared to WT (Fig. 4E). In addition, *talus* of $Ids^{y/-}$ mice showed a significant increase of the cortical bone area (Ct.Ar/Tt.Ar), with a 13% increase compared to WT animals (Fig. 4F), and a tendency toward a decreased trabecular bone surface (Tb.BS), with ~41% reduction compared to WT mice (Fig. 4G). The *navicular-lateral cuneiform* of $Ids^{y/-}$ mice showed a tendency toward decreased Tb.TV (41% reduction compared to WT; Fig. 4I) and increased Ct.Ar/Tt.Ar (10% increase compared to WT; Fig. 4J), while no differences were observed in Tb.BV and Tb.BS (Fig. 4H,K). The *medial cuneiform* of $Ids^{y/-}$ mice did not present noticeable differences for any of the parameters tested compared to WT animals (Fig. EV4B–E). Depending on the vector tested, gene

therapy had a variable effect on bone microarchitecture of the *talus* and the *navicular-lateral cuneiform*. Mock treatment with GFP vector had no impact on the bone microarchitecture parameters measured, and resulted in values comparable to those observed in untreated $Ids^{y/-}$ mice for all the bones examined (Figs. 4D–K and EV4B–E). IDSco gene therapy resulted in partial correction of Tb.TV and the Ct.Ar/Tt.Ar of the *talus* (Fig. 4E,F) and a more substantial correction of the same parameters in the *navicular-lateral cuneiform* (Fig. 4I,J), but had no effect on the Tb.BV and the Tb.BS of the *talus* (Fig. 4D,G). On the other hand, IDS.IGF2del_co gene therapy resulted in no correction for all the parameters analyzed in both *talus* and *navicular-lateral cuneiform* (Fig. 4D–K). Importantly, IDS.IGF2co and IDS.SWAP-RAP12x2co showed a general improvement of all the parameters tested compared to both IDSco and IDS.IGF2del_co. Specifically, when compared to IDSco, the correction provided by the IDS.IGF2co and IDS.SWAP-RAP12x2co vectors was more pronounced for the Tb.BV, the Tb.TV and the Tb.BS of the *talus* (Fig. 4D–G). In contrast, IDS.IDS.SWAP-ApoEco treatment was comparable to IDSco for most of the parameters, but not for the Ct.Ar/Tt.Ar of the *talus*, for which it resulted in an improved correction (Fig. 4D–J).

These results demonstrate that gene therapy with SWAP vectors effectively prevented alcian blue and LAMP1-related pathology in liver, spleen and cardiac muscle, and nearly fully restored those bone microarchitecture parameters that were significantly different between $Ids^{y/-}$ and wild-type mice, although these improvements did not reach statistical significance. This highlights the potential of SWAP vectors for addressing both peripheral and skeletal pathologies in lysosomal storage disorders.

## Superior correction of brain pathology by gene therapy with *IDS.SWAP* variants

Gene therapy with all vectors tested resulted in comparable IDS activity in brain homogenates at levels that were ~10–20 times higher compared to untreated $Ids^{y/-}$ animals, and ~10–30 times lower compared to untreated WT mice (Fig. 5A), as previously reported (Catalano et al, 2023; Gleitz et al, 2018). Using mass spectrometry, we observed a significant increase of the heparan sulfate levels in brain homogenates of $Ids^{y/-}$ mice compared to WT mice (~70-fold above WT; Fig. 5B,C), but not a significant increase of the dermatan sulfate levels (Fig. EV4I), as shown previously (Catalano et al, 2023). Gene therapy with all vectors tested caused a significant reduction of the heparan sulfate at levels depending on the vector tested (Fig. 5B,C). We plotted all data points of this and

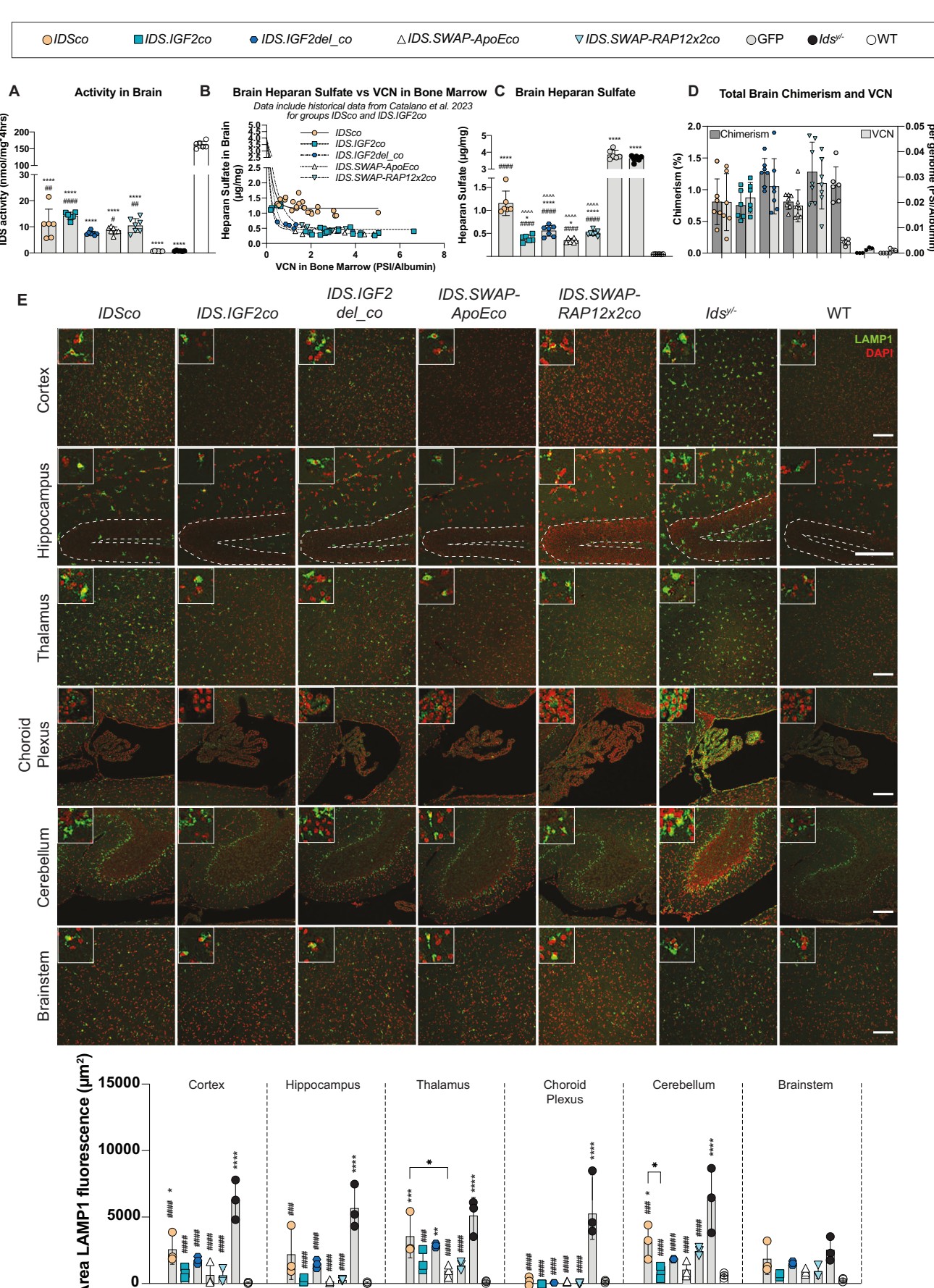

**Figure 5.  Correction of brain heparan sulfate and lysosomal pathology by gene therapy with IDS.SWAP variants.**

(A) IDS enzyme activity in brain. All significant comparisons in (A) showed *P* values < 0.0001, except for IDSco vs. *Ids*$^{y/-}$ *P* = 0.0031, IDS.SWAP-ApoEco vs. *Ids*$^{y/-}$ *P* = 0.0442, IDS.SWAP-RAP12x2 vs. *Ids*$^{y/-}$ *P* = 0.0042. (B) Relationship between heparan sulfate in brain after gene therapy and VCN in bone marrow. Regression analysis of (B) is shown in Table EV1. (C) Heparan sulfate quantification in total brain homogenates by mass spectrometry. All significant comparisons in (C) showed *P* values < 0.0001, except for IDS.IGF2co vs. WT *P* = 0.0113, IDS.SWAP-ApoEco vs. WT *P* = 0.0353. (D) Analysis of Chimerism and VCN per genome in brain by qPCR analysis on the *PSI* and *Albumin* loci (VCN) and by allele specific qPCR on the *Cd45.1* locus (Chimerism). (E) LAMP1 (green) staining of sagittal sections of cortex, hippocampus, thalamus, choroid plexus, cerebellum and brainstem. Scale bars = 100 μm. Nuclei are stained in red. Dashed lines outline the Cornu Ammonis (CA) fields 2 and 3. Quantification of the area of LAMP1 fluorescence is shown. VCN in bone marrow of the mice used for histology is shown in Fig. EV4J. An example of LAMP1 staining on *GFP*-treated *Ids*$^{y/-}$ mice is shown in Fig. EV5. All significant comparisons in (E) showed *P* values < 0.0001, except for: *Cortex*, IDSco vs. WT *P* = 0.0174. *Hippocampus*, IDSco vs. *Ids*$^{y/-}$ *P* = 0.0001. *Thalamus*, IDSco vs. WT *P* = 0.0002, IDS.IGF2del_co vs. WT *P* = 0.0058, IDS.IGF2co vs. *Ids*$^{y/-}$ *P* = 0.0002, IDSco vs. IDS.SWAP-ApoEco *P* = 0.0104. *Cerebellum*, IDSco vs. WT *P* = 0.0139, IDSco vs. *Ids*$^{y/-}$ *P* = 0.0007, IDSco vs. IDS.IGF2co *P* = 0.0475. Data information: data are presented as means ± SD and were analyzed by one-way ANOVA with Bonferroni's correction. Asterisks (*) represent significance versus WT; hashes (#) represent significance versus *Ids*$^{y/-}$; carets (ˆ) represent significance versus IDSco. In (A–D): IDSco, GFP, *Ids*$^{y/-}$ and WT n = 6; IDS.IGF2del_co, IDS.SWAP-ApoEco and IDS.SWAP-RAP12x2co n = 7. In (E), n = 3. *P ≤ 0.05; **P ≤ 0.01; ****P ≤ 0.0001. ###P ≤ 0.001; ####P ≤ 0.0001. Significant results are indicated by brackets. Source data are available online for this figure.

our previous study (Catalano et al, 2023) and found that heparan sulfate decreased exponentially per VCN in bone marrow (Fig. 5B; Table EV1). Gene therapy with the mock *GFP* vector had no effect on the heparan sulfate, resulting in levels that were comparable to untreated *Ids*$^{y/-}$ mice. IDSco gene therapy reduced heparan sulfate at levels that were three times lower than *Ids*$^{y/-}$ mice, but that remained 23 times higher than WT mice. Tagging of IDS with IGF2 versions caused a further reduction of heparan sulfate at levels that were two- to threefold lower than those observed after gene therapy with untagged IDS (Fig. 5C; IDS.IGF2co and IDS.SWAP-ApoEco ~ three times lower than IDSco, 10 times lower than untreated *Ids*$^{y/-}$ mice, and 7.5 times higher than WT mice; IDS.SWAP-RAP12x2co and IDS.IGF2del_co ~ 2.1 times lower than IDSco, 7 times lower than *Ids*$^{y/-}$ mice, and 11 times higher than WT mice).

As HSPC-LVGT is expected to result in engraftment of donor HSPC-derived cells in brain, we investigated the levels of engraftment in brain via measuring the levels of VCN and total chimerism (Fig. 5D) in brain homogenates. Following gene therapy, we observed comparable levels of brain VCN across all the tested vectors, which were ~0.025 copies per genome (Fig. 5D). Treatment with the *GFP* vector resulted in values that were approximately 5 times lower compared to the other groups (Fig. 5D) and that were consistent with the difference in bone marrow VCN between this treatment group and the others (Fig. 3D). This has likely a technical reason due to the virus titration process. Importantly, we note that a lower VCN would not compromise this experimental group's role as a control for the preconditioning procedure. Analysis of chimerism in brain demonstrated similar levels across the vectors tested of ~1% (Fig. 5D), ruling out that the transduction with lentiviral vectors affected engraftment of HSPC-derived cells in the brain.

To assess correction of brain pathology using histology, three mice per treatment group were selected that had a similar VCN in BM to allow a fair comparison (Fig. EV4J,K). To analyse correction of brain lysosomal pathology by gene therapy, we performed immunostaining of LAMP1 protein on brain sagittal sections. As previously reported (Catalano et al, 2023), we observed a widespread increased of LAMP1 immunoreactivity in brain sections of *Ids*$^{y/-}$ mice compared to WT mice (Fig. 5E), which was not affected by *GFP*-treatment (Fig. EV5). Gene therapy with all the tested vectors resulted in reduced LAMP1 fluorescence, with levels that varied among the different vectors, with IDS.IGF2co and IDS.SWAP variants showing the most significant reduction

(Fig. 5E). Specifically, gene therapy with IDSco caused a ~2–2.5-fold reduction of LAMP1 levels in cortex, hippocampus and cerebellum, and a milder reduction of LAMP1 levels in thalamus and brainstem (~1.4-fold). IDS.IGF2co led to further reductions in all regions (~8 fold in the cortex, ~21-fold in the hippocampus, ~3-fold in the thalamus, ~7-fold in the cerebellum, and ~4 fold in the brainstem), with levels comparable to those observed with IDS.SWAP-ApoEco and IDS.SWAP-RAP12x2co gene therapies, although these vectors demonstrated even greater correction in some areas. Specifically, LAMP1 immunoreactivity in hippocampus and thalamus was 1.76 times lower after IDS.SWAP-ApoEco treatment compared to IDS.IGF2co treatment, while no differences were observed in cortex, brainstem and cerebellum. A similar pattern was observed comparing IDS.SWAP-RAP12x2 and IDS.IGF2co treatments, with exceptions in thalamus—where LAMP1 immunoreactivity levels were comparable—and in cerebellum—where LAMP1 levels were 2.7 times higher after IDS.SWAP-RAP12x2 compared to IDS.IGF2co. Gene therapy with IDS.IGF2del_co caused a reduction of brain LAMP1 immunoreactivity at levels that ranged from comparable to slightly lower than those observed after IDSco treatment (cortex and hippocampus: ~1.5 times reduction compared to IDSco treatment; cerebellum: 1.73 times reduction compared to IDSco treatment), and ~2–5 times higher than the other vectors (cortex, hippocampus and brainstem: ~2 times increase compared to IDS.IGF2co treatment; thalamus: ~1.7 times increase compared to IDS.IGF2co treatment; hippocampus: 5 times increase compared to IDS.IGF2co treatment).

We also analyzed correction of neuroinflammation by performing immunostaining for CD68, a marker of activated microglia when used in brain (Figs. 6 and EV5), and GFAP, a marker for astrocytes (Figs. 7 and EV5) (Catalano et al, 2023). *Ids*$^{y/-}$ mice presented a widespread increased of the number of CD68-positive cells compared to WT animals (Fig. 6), which was not affected by the *GFP*-treatment (Fig. EV5). Gene therapy with all tested vectors resulted in a reduction of CD68 levels, with variations among the different vectors, and the lowest levels observed following therapy with IDS.IGF2co and IDS.SWAP vectors. Specifically, IDSco and IDS.IGF2del_co caused a reduction of CD68-positive cells in all areas analyzed and at levels that were on average ~3.5-fold lower than those observed in *Ids*$^{y/-}$ mice, with a more prominent reduction observed in hippocampus, cerebellum and brainstem (~4.5-fold reduction compared to *Ids*$^{y/-}$ mice). IDS.IGF2co gene therapy further reduced the CD68 pathology to levels that were on average 2-fold lower than IDSco and 7-fold lower than *Ids*$^{y/-}$ mice. Gene

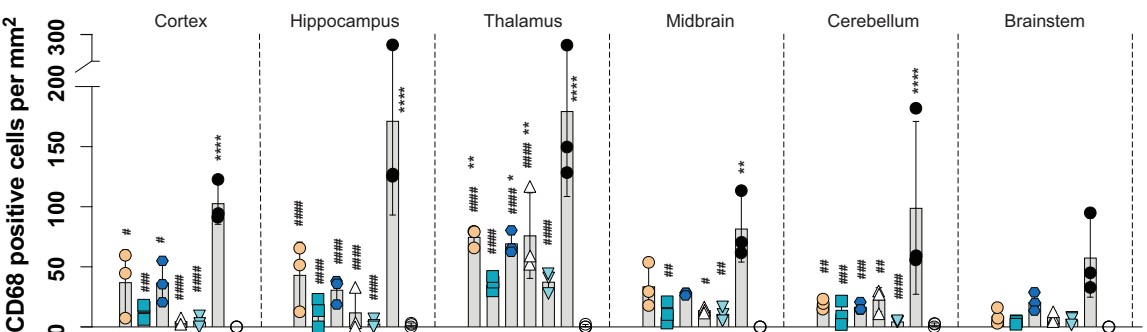

**Figure 6. Correction of CD68 pathology by gene therapy with IDS.SWAP variants.**

Representative images of CD68 staining of sagittal sections of cortex, hippocampus, thalamus, midbrain, brainstem and cerebellum of gene therapy-treated $Ids^{y/-}$ mice and controls. Scale bars = 100 μm. Quantification of CD68-positive cells is shown. VCN in bone marrow of the mice used for histology is shown in Fig. EV4K. An example of CD68 staining on GFP-treated $Ids^{y/-}$ mice is shown in Fig. EV5. All significant comparisons in showed P values < 0.0001, except for: Cortex, IDSco vs. $Ids^{y/-}$ P = 0.0210, IDS.IGF2co vs. $Ids^{y/-}$ P = 0.0003, IDS.IGF2del_co vs. $Ids^{y/-}$ P = 0.0201. Thalamus, IDSco vs. WT P = 0.0047, IDS.IGF2del_co vs. WT P = 0.0123, IDS.SWAP-ApoEco vs. WT p = 0.0038. Midbrain, IDS.IGF2co vs. $Ids^{y/-}$ P = 0.0093, IDS.SWAP-ApoEco vs. $Ids^{y/-}$ P = 0.0141, IDS.SWAP-RAP12x2co vs. $Ids^{y/-}$ P = 0.0081. Cerebellum, IDSco vs. $Ids^{y/-}$ P = 0.0016, IDS.IGF2co vs. $Ids^{y/-}$ P = 0.0003, IDS.IGF2del_co vs. $Ids^{y/-}$ P = 0.0010, IDS.SWAP-ApoEco vs. $Ids^{y/-}$ P = 0.0031. Data information: data are presented as means ± SD and were analyzed by one-way ANOVA with Bonferroni's correction. Asterisks (*) represent significance versus WT; hashes (#) represent significance versus $Ids^{y/-}$. n = 3. *P ≤ 0.05; **P ≤ 0.01; ****P ≤ 0.0001. #P ≤ 0.05; ##P ≤ 0.01; ####P ≤ 0.0001. Significant results are indicated by brackets. Source data are available online for this figure.

therapy with *IDS.SWAP* vectors resulted in a reduced number of CD68-positive cells in midbrain and cerebellum at levels that were comparable to those observed after *IDS.IGF2co* gene therapy, while resulting in a 5.6 times (*IDS.SWAP-ApoEco*) or 2.3 times (*IDS.SWAP-RAP12x2co*) further reduction in cortex, and 2 times (*IDS.SWAP-ApoEco*) or 5 times (*IDS.SWAP-RAP12x2co*) further reduction in hippocampus compared to *IDS.IGF2co* gene therapy. However, *IDS.SWAP-ApoEco* gene therapy was less effective in thalamus, where levels of CD68-positive cells were comparable to *IDSco* and *IDS.IGF2del_co* treatments and ~1.5 times higher than *IDS.IGF2co* and *IDS.SWAP-RAP12x2co* treatment.

Similarly, immunostaining of GFAP showed a regional increase of GFAP immunoreactivity in brain sections of $Ids^{y/-}$ mice (Fig. 7), which was not affected by *GFP*-treatment (Fig. EV5). Following gene therapy, GFAP fluorescence decreased with varying levels across treatments. The greatest reductions were observed with *IDS.IGF2co* and *IDS.SWAP* vectors. *IDSco* caused a ~2-fold reduction of GFAP fluorescence levels in cortex, but had no effect in thalamus, midbrain and brainstem. *IDS.IGF2co* and *IDS.SWAP* vectors gene therapy further reduced GFAP immunoreactivity in cortex, thalamus, midbrain and brainstem at levels that were ~4.4, ~3.6, ~2.6, ~3.9 times lower than those observed in $Ids^{y/-}$ mice, respectively. *IDS.IGF2del_co* treatment caused a reduction of GFAP fluorescence in thalamus, midbrain and brainstem—but not in cortex—at levels that were ~2.5-fold lower than *IDSco* treatment, but that were ~2-fold higher than after *IDS.IGF2co* treatment.

These findings indicate that IDS.SWAP variants and IDS.IGF2 achieved similar efficacy in the brain, showed a trend towards superior efficacy on histopathology markers compared to *IDS.IGF2del* in some brain areas and showed clearly enhanced efficacy compared to IDSco, as evidenced by greater reductions in cerebral heparan sulfate, alleviation of astrogliosis, and enhanced LAMP1 clearance.

## Discussion

In this study, we found that the central loop of IGF2 is not required for binding to CI-M6P/IGF2R but is essential for high-affinity binding to IR-A and IGF1R. This insight led to the development of the SWAP design, a structurally cohesive IGF2-based tag with a more favourable receptor interaction profile than the original IGF2 tag. We showed that the SWAP design retains binding to the CI-M6P/IGF2R, reduces binding to the IR-A (SWAP-ApoE and SWAP-RAP12x2) and IGF1R (only SWAP-RAP12x2), and includes the option for binding additional receptors (or protein) such as LRP-1. This offers a promising novel tagging platform to enhance gene therapy treatment of lysosomal storage disorders such as Hunter syndrome.

Upon deletion of the central loop of IGF2 (IGF2del) or its substitution with other epitopes (SWAP-ApoE or SWAP-RAP12x2), we observed a complete loss of binding affinity to the IR-A. On the other hand, binding to the IGF1R was reduced either upon deletion of the central loop or its substitution with RAP12x2, but not upon substitution with ApoE. For the IR-A, these findings align with structural and mutagenesis studies demonstrating that the central loop residues R37 and R38 are essential for high-affinity binding (An et al, 2024). Therefore, this loss of key interactions likely explains the reduced affinity of IGF2del to the IR-A, and indicates that the inserted epitopes (ApoE and RAP12x2) could not replicate these interactions. Similarly, for IGF1R, structural studies have suggested possible interactions between residues R30 and R38 of IGF2's central loop and IGF1R (Xu et al, 2020). As for the IR-A, this suggests that the loss of these interactions between the IGF2's central loop and IGF1R is responsible for the lack of IGF1R binding of IGF2del. However, SWAP-ApoE, but not SWAP-RAP12x2, retained IGF1R binding at levels similar to unmodified IGF2, suggesting that the ApoE, but not RAP12x2, could replicate these interactions. One possible explanation for this finding is the presence of the "RKL" and "KRL" motifs in ApoE, which mimic the "RRA" motif found in IGF1 (R36, R37 and A38), and known to be critical for IGF1 binding to the IGF1R (Blyth et al, 2020; Xu et al, 2020). This motif is absent in SWAP-RAP12x2, possibly explaining the lower binding affinity of SWAP-RAP12x2 for IGF1R compared to SWAP-ApoE. Alternatively, a structural, sequence-independent requirement for a flexible central loop in IGF2 and SWAP-derived tags may explain our findings regarding their binding to the IR-A and IGF1R. For example, An et al, 2024 showed that IGF2's central loop undergoes structural rearrangements to accommodate the IR-A's α-CT domain during binding. This could suggest that a loss of structural flexibility—due to the lack of a flexible central loop in the IGF2del, or the presence of a disulfide bond at the ends of the insertion in the SWAP design (predicted for the SWAP variants in Fig. 1A) —may be involved in the reduced affinity to the IR. Similarly, studies have proposed structural rearrangements of IGF2 during binding to the IGF1R. Examples are the folding out of IGF2's B-chain, or the opening of the central loop for accommodation of the αCT domain of IGF1R (Xu et al, 2020). Therefore, beside engaging the receptor via polar interactions, the central loop of IGF2 could play a critical role in facilitating the structural rearrangements needed for receptor engagement. In this context, the RAP12x2 insert—larger than ApoE insert (25 AA vs. 9 AA)—in the SWAP design, but not the ApoE insert, could increase the enthalpic cost required for the necessary structural rearrangements, thereby reducing binding affinity to the IGF1R.

Besides mediating a reduced binding affinity for the IR-A and IGF1R, the SWAP design proved to be an effective modular

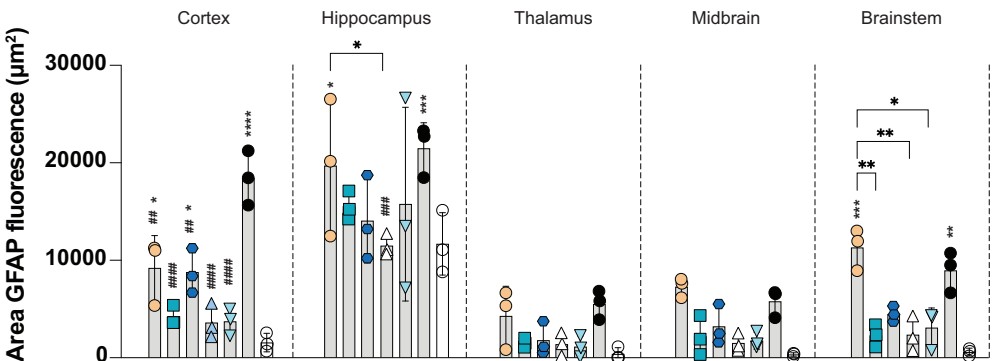

**Figure 7. Efficient correction of GFAP pathology by gene therapy with IDS.SWAP variants.**

Representative images of GFAP (green) staining of sagittal sections of cortex, hippocampus, thalamus, midbrain and brainstem of gene therapy-treated $Ids^{y/-}$ mice and controls. Scale bar = 100 μm. Nuclei are stained in red. Dashed lines outline the Cornu Ammonis (CA) fields 2 and 3. Quantification of GFAP fluorescence area is shown. VCN in bone marrow of the mice used for histology is shown in Fig. EV4J. An example of CD68 staining on GFP-treated $Ids^{y/-}$ mice is shown in Fig. EV5. All significant comparisons in showed P values < 0.0001, except for: Cortex, IDSco vs. WT = 0.0239, IDS.IGF2del_co vs. WT P = 0.0434, IDSco vs. $Ids^{y/-}$ = 0.0023, IDS.IGF2del_co vs. $Ids^{y/-}$ P = 0.0012. Hippocampus, IDSco vs. WT P = 0.0140, $Ids^{y/-}$ vs. WT P = 0.0010, IDS.SWAP-ApoEco vs. $Ids^{y/-}$ P = 0.0007, IDSco vs. IDS.SWAP-ApoEco P = 0.0106. Brainstem, IDSco vs. WT P = 0.0002, $Ids^{y/-}$ vs. WT P = 0.0087, IDSco vs. IDS.IGF2co P = 0.0033, IDSco vs. IDS.SWAP-ApoEco P = 0.0037, IDSco vs. IDS.SWAP-RAP12x2co P = 0.0110. Data information: data are presented as means ± SD and were analyzed by one-way ANOVA with Bonferroni's correction. Asterisks (*) represent significance versus WT; hashes (#) represent significance versus $Ids^{y/-}$. n = 3. *P ≤ 0.05; **P ≤ 0.01; ***P ≤ 0.001; ****P ≤ 0.0001. #P ≤ 0.05; ##P ≤ 0.01; ###P ≤ 0.001; ####P ≤ 0.0001. Significant results are indicated by brackets. Source data are available online for this figure.

platform for targeting multiple relevant receptors by switching inserted epitopes of different lengths. Specifically, we tested ApoE (9 AA) and RAP12x2 (25 AA) epitopes. These variants demonstrated simultaneous engagement of the CI-M6P/IGF2R and LRP-1 with high-affinity that was similar to single tagged IDS versions (IDS.IGF2 and IDS.ApoEII), without introducing the complexity or structural interference typically associated with double-tagging methods, or sometimes observed with simple N- or C-terminal fusion strategies. For example, we previously showed that C-terminal tagging of IDS with RAP12x2 failed to engage the LRP-1. As a result, IDS.RAP12x2 failed to provide a therapeutic advantage over untagged IDS in vivo (Catalano et al, 2023). In contrast, SWAP variants with either an ApoE or RAP12x2 inserts outperformed untagged IDS in correcting or preventing most peripheral and CNS manifestations in a murine Hunter syndrome model. Additionally, the IDS.SWAP variants more effectively normalized cardiac Alcian blue staining and LAMP1 pathology, matching the efficacy of IDS.IGF2, and showed a trend toward superior correction of histopathology compared with IDS.IGF2del. This might indicate that the C-domain of IGF2—deleted in IGF2del and involved in the binding to the insulin receptor and IGF1R (LeBowitz & Maga, 2012; Blyth et al, 2020)—partially exerts the IGF2-mediated enhancement of therapeutic efficacy during HSPC-LVGT, and that his role can be replaced by the epitopes inserted in the SWAP design.

In addition to testing IDS.RAP12x2, we also previously tested the IDS.ApoEII construct (Catalano et al, 2023). IDS.ApoEII comprises a tandem repeat of an ApoE-derived peptide (sequence LRKLRKRLL x 2), while IDS.SWAP-ApoE comprises a single repeat of the same sequence. While it was previously observed that the isolated single-repeat peptide has a reduced biological activity compared to the tandem-repeat ApoEII, this was mainly attributed to the failure of the short isolated single repeat to stably adopt the α-helical conformation characteristic of native Apolipoprotein E (Dobson et al, 2006; Minami et al, 2010). However, when embedded within a larger amino-acid context, such as in the native Apolipoprotein E sequence, a single LRKLRKRLL repeat can adopt an α-helical structure and binds LRP-1 with high affinity (Croy et al, 2004), a mechanism that may also operate in the SWAP design. Consistently, in Fig. EV2C we show that IDS.ApoEII and IDS.SWAP-ApoE bind Cluster IV of LRP-1 with very similar affinities, with the latter exhibiting only a slight decrease. In vivo, IDS.SWAP-ApoE largely mirrored IDS.IGF2 for the prevention of brain pathology, and showed a non-significant trend toward better cortex rescue. Both IDS.SWAP-ApoE and IDS-IGF2 showed near-complete rescue of disease hallmarks in brain, while IDS.ApoEII in previous studies showed complete rescue (Catalano et al, 2023; Gleitz et al, 2018). We noticed different pharmacokinetic properties of IGF2-based IDS including SWAP versions, which might be caused by scavenging to various tissues via the IGF2R and/or by binding plasma proteins, which should be assessed in future studies. We speculate that the lower plasma concentration of IGF2-based tags might limit the availability of transgene products to reach the brain via transcytosis, which should be addressed in future experiments (Catalano et al, 2023; Gleitz et al, 2018).

ERT with an IGF2-tagged GAA analog caused dose-dependent hypoglycemia shortly after drug administration in patients with Pompe disease. This was deemed to be caused by the pharmacological effect of the IGF2 moiety (Byrne et al, 2017). In this study, patients with hypoglycemic reactions showed a maximum plasma concentration ($C_{max}$, observed shortly after administration) of ~50000–100,000 ng/ml, corresponding to ~450–900 nM of IGF2 moieties (IGF2.GAA: ~110 KDa). In our studies, gene therapy with IDS.IGF2co resulted in plasma levels of ~200 ng/ml (Fig. EV3F), corresponding to ~2.5 nM of IGF2 moieties (IDS.IGF2: ~80 KD), which is ~175 to 350-fold lower compared to the $C_{max}$ IGF2 moieties values observed after bolus administration of IGF2.GAA. This suggests that, during gene therapy, the level of expression of IGF2 moieties may be several fold lower compared to the previously observed concentration range associated with hypoglycemia. Despite this, IGF2-related toxicity could still manifest in gene therapy settings due to the constant and long-term exposure to supraphysiological levels of IGF2 moieties, compared to single administration during bolus ERT. In this respect, E20 mouse embryos systemically overexpressing IGF2 at levels 65% above control showed increased body weight and pancreatic islet cells hyperplasia (Petrik et al, 1999). Additionally, 5–10-weeks old mice overexpressing IGF2 systemically showed increased body weight and impaired glycolipid metabolism (Zhang et al, 2023). Here, after gene therapy, we did not observe specific effects of IGF2-tagging on body weight, nor an effect on post-prandial glucose levels or on glucose tolerance. This suggests that the levels of expression of IGF2 moieties of IDS.IGF2 do not interfere with glucose metabolism during lentiviral gene therapy. Another scenario in which IGF2 overexpression could lead to adverse effects is a high level of local expression in proximity of insulin-sensitive cells. In the context of gene therapy, such an effect might be caused by transgene-expressing macrophages engrafting in proximity of, for example, pancreatic islets. In this context, Devedjian and colleagues showed that, when IGF2 overexpression is restricted to β-cells, mice developed hyperinsulinemia, hyperglycemia and impaired response to glucose tolerance tests, even though serum levels of IGF2 were only ~twofold higher than WT animals and comparable to the levels observed by us after IDS.IGF2co gene therapy (Devedjian et al, 2000). Additionally, species differences in glucose regulation between mice and humans might still pose a risk for using IGF2-based therapeutics for gene therapy in humans, especially in a single dosing gene therapy setting resulting in long-term expression (Bruce et al, 2021). In all these scenarios, using IGF2-derived tags with abolished or lowered binding to IR or IGF1R is favourable over unmodified IGF2 tag given similar therapeutic efficacy. For this reason, the SWAP design, and SWAP-RAP12x2 in particular, represents a favourable transgene candidate for lentiviral gene therapy for Hunter syndrome, as well as for approaches involving IGF2-based therapeutics where peak plasma levels strongly exceed those achieved by HSPC-LVGT (e.g., ERT).

In conclusion, this study extends our understanding of the role of the central loop of IGF2 in receptor binding and specificity, and builds on this to develop a modified version of IGF2 for enhancing efficacy of lentiviral gene therapies. This SWAP design enables modular targeting of therapeutically relevant receptors to achieve a more favourable receptor-targeting profile with reduced binding to IR-A and IGF1R and novel receptor binding properties, relevant for the clinical translation of lentiviral gene therapies for lysosomal disorders such as Hunter syndrome.

# Methods

### Reagents and tools table

| Reagent/resource | Reference or source | Identifier or catalog number |
|---|---|---|
| **Experimental models** | | |
| B6.SJL-Ptprca Pepcb/BoyJ mice | Jackson Laboratory | 2014 |
| B6N.Cg-Idstm1Muen/J mice | Jackson Laboratory | 24744 |
| bEND.3 cells | ATCC | CRL-2299 |
| C57BL/6J mice | Jackson Laboratory | 664 |
| HMC3 cells | ATCC | CRL-3304 |
| MPS II fibroblasts | Primary, male, mut: p.Leu182Cysfs*31 | N/A |
| SH-SY5Y cells | ATCC | CRL-2266 |
| **Recombinant DNA** | | |
| LV.IDSco | Catalano et al, 2023 | N/A |
| LV.IDS.IGF2co | Catalano et al, 2023 | N/A |
| **Antibodies** | | |
| Anti-Rabbit Alexa Fluor® 488 secondary antibody | ThermoFisher Scientific | A-11008 |
| Anti-Rat Alexa Fluor® 488 secondary antibody | ThermoFisher Scientific | A-11006 |
| Biotinylated goat anti-rat secondary antibody | BD Pharmingen | 554014 |
| FITC-anti-mouse CD45.2 | BD Biosciences | 553772 |
| Goat anti-human IDS primary antibody (dilution 1:1000) | R&D Systems | F2449 |
| IRDye 680RD secondary antibody (1:10,000) | LI-COR Biosciences | 925-68071 |
| Mouse anti-His tag antibody | R&D Systems | MAB050 |
| Mouse IgG2a-FITC isotype control | BD Biosciences | 349051 |
| Mouse IgG2a-PE isotype control | BD Biosciences | 349053 |
| PE-anti-mouse CD45.1 | BD Biosciences | 553776 |
| Rabbit anti-GFAP IgG | Sigma-Aldrich | G9269 |
| Rat anti-CD68 primary antibody | Bio-Rad | MCA1957T |
| Rat anti-LAMP1 primary antibody | Abcam | ab25245 |
| **Oligonucleotides and other sequence-based reagents** | | |
| Cd45.1 allele FW primer (5'-CTGAGCCTGCATCTAAACCTGATC-3') | IDT | N/A |
| Cd45.1 allele RV primer (5'-TCACCTTCATAAAAGCCTTGTAGCTC-3') | IDT | N/A |
| mouse Albumin FW primer (5'-ACTTTGAGTGTAGCAGAGAGGAACC-3') | IDT | N/A |
| mouse Albumin RV primer (5'-CTCTTCACTGACCTAAGCTACTCCC-3') | IDT | N/A |
| PSI FW primer (5'-CAGGACTCGGCTTGCTGAAG-3') | IDT | N/A |
| PSI RV primer (5'-TCCCCCGCTTAATACTGACG-3') | IDT | N/A |

| Reagent/resource | Reference or source | Identifier or catalog number |
|---|---|---|
| **Chemicals, enzymes and other reagents** | | |
| 3 M HCl-methanol | Merck | 90964 |
| 4-Methylumbelliferyl-α-L-idopyranosiduronic acid-2-sulphate (4-MU) | Biosynth/Carbosynth | M-5430 |
| Alcian Blue 8GX | Sigma | 33864-99-2 |
| Aluminum sulfate hexadecahydrate | Sigma | 16828-11-8 |
| Avidin/biotin blocking kit | Vector Laboratories | SP-2001 |
| B27 supplement (2%) | Gibco | 17504044 |
| Biotinylated IGF1 peptide | CellSciences | AQU050 |
| Biotinylated IGF2 peptide | CellSciences | AXU020 |
| Biotinylated insulin | CellSciences | AYU100 |
| Bovine serum albumin | Sigma | A3059 |
| Citrate Buffer (4%) | Sigma | 6132-04-3 |
| DMEM/F-12 | Gibco | 11320033 |
| Elaprase™ (idursulfase) | Takeda Pharmaceuticals | N/A |
| ELISA substrate solution | R&D Systems | DY999 |
| Entellan mounting medium | Sigma | 1079600500 |
| Equimolar dNTP mix, 25 mM | ThermoFisher | R0181 |
| Fetal bovine serum (10%) | Capricorn Scientific | FBS-12A |
| GlutaMAX (1%) | Gibco | 35050-061 |
| Goat serum | Thermo Fisher Scientific | 50197Z |
| Hoechst 33258 | Life Technologies | N/A |
| Human α-L-iduronidase (5 µg/mL) | R&D Systems | 4119-GH-010 |
| Hydrogen peroxidase | PHC Corporation | BMS-2110-1E |
| Impact DAB™ | Vector Laboratories | SK-4105 |
| iTaq Universal SYBR Green Supermix | Bio-Rad | 172-5120 |
| Mature human IGF2 peptide des1-6 | CellSciences | MU100 |
| Neurobasal medium | Gibco | 21103049 |
| Non-essential amino acids (NEAA) | Gibco | 11140050 |
| Nuclear Fast Red | Sigma | 6409-77-4 |
| Penicillin/Streptomycin (1%) | Gibco | 15070-063 |
| Phusion polymerase | NEB | M0530L |
| Recombinant murine Flt-3 ligand | R&D Systems | 427-FL-005/CF |
| Recombinant murine stem cell factor | R&D Systems | 455-MC-010/CF |
| Recombinant murine thrombopoietin | R&D Systems | 488-TO-005/CF |
| Retinoic acid (10 µM) | Sigma | R2625 |
| StemSpan™ SF Expansion Medium | Stemcell Technologies | 9600 |
| Streptavidin-HRP | R&D Systems | DY998 |
| T5 Exonuclease | NEB | M0663L |
| Triton X-100 | Thermo Fisher Scientific | T8787 |

| Reagent/resource | Reference or source | Identifier or catalog number |
|---|---|---|
| **Software** | | |
| AccuCT software (ASBMR module) | PerkinElmer | N/A |
| AlphaFold3 | Abramson et al, 2024 | N/A |
| CFX Manager 3.0 | Bio-Rad | N/A |
| FACS-DIVA software | BD Biosciences | N/A |
| Fiji 2.17 | ImageJ | N/A |
| FlowJo v10 | BD Biosciences | N/A |
| GraphPad Prism 9.0.0 | GraphPad Software | N/A |
| Image Studio 6.0 | LI-COR Biosciences | N/A |
| Quantum GX2 software | PerkinElmer | N/A |
| **Other** | | |
| Amicon protein concentrators (30 kDa cutoff) | Thermo Scientific | 88536 |
| BD LSRFortessa™ flow cytometer | BD Biosciences | N/A |
| CFX96 real-time PCR detection system | Bio-Rad | N/A |
| Cu 0.06 mm + Al 0.5 mm filter | PerkinElmer | N/A |
| DuoSet® ELISA kit - IDS | R&D Systems | DY449-05 |
| Gammacell® 40 irradiator | Atomic Energy of Canada LTD | N/A |
| Glucometer | Swisspointofcare | OGM-191 |
| Glucose solution (45% w/v) | Sigma | G8769 |
| Histokinette | N/A | N/A |
| Leica Stellaris5 confocal microscope | Leica | N/A |
| Lineage Cell Depletion Kit – mouse | Miltenyi Biotec | 130-110-470 |
| NanoZoomer 2.0 slide scanner | Hamamatsu Photonics | N/A |
| Odyssey Infrared Imaging System | LI-COR Biosciences | CLx |
| Pierce™ BCA Protein Assay Kit | Thermo Fisher Scientific | 23225 |
| Quantum GX μCT imaging system | PerkinElmer | N/A |
| Sciex 5500 QTrap mass spectrometer | Sciex | N/A |
| Trans-Blot Turbo Mini 0.2 μm Nitrocellulose Transfer | Bio-Rad | 1704158 |
| Varioskan microplate reader | Thermo Fisher Scientific | Flash |
| Waters Acquity UPLC system | Waters | N/A |

## Animals and procedures

The animal experiments were conducted as previously shown (Catalano et al, 2023). $Ids^{y/-}$ and WT mice were generated breeding heterozygous female B6N.Cg-Ids$^{tm1Muen}$/J mice with wild-type C57BL/6J males. CD45.1 donor $Ids^{y/-}$ were generated following the same procedures using the B6.SJL-Ptprc$^a$ Pepc$^b$/BoyJ background.(Gleitz et al, 2018). Mice were bred according to standard procedures as previously delineated (Catalano et al, 2023). Mice were anesthetized and perfused with 50 ml of phosphate-buffered saline (PBS). For the glucose tolerance test (GTT), animals were weighed in the morning and starved for 6 h. GTT was performed in the afternoon via intraperitoneal injection. Glucose solution was purchased from Sigma (G8769; 45% glucose solution w:v) and diluted in PBS to a concentration of 1 mg/g of body weight in a total volume of 200 μl. Glucose levels were measured using a glucometer (Swisspointofcare) on blood derived from tail puncture. All animal experiments in this study were approved by the Animal Experiments Committee (DEC) in the Netherlands and these complied with the Dutch legislature to use animals for scientific procedures.

## Lentiviral vector construction and production

*IDSco* and *IDS.IGF2co* constructs were generated as described before (Catalano et al, 2023). Briefly, the vector was based on a pCCL backbone, with transgene expression driven by the MND promoter. *IDS.SWAP* vectors were generated by Gibson assembly. The backbone for the Gibson assembly was prepared through PCR amplification of the entire *IDS.IGF2co* vector, excluding the sequence encoding amino acids 29–41 of the human IGF2 tag (Fig. 1B). The insert for the Gibson assembly was generated by PCR amplification of a primer encoding the inserted epitopes (Fig. 1B) and with 20 bp overlap at the 5' and the 3' of the backbone. The Gibson assembly reaction consisted in mixing of 10 μl of Gibson Mix, 50 ng of backbone, and the insert at 3 times the molar concentration of the backbone, totaling 20 μl in volume. This mixture was incubated for 1 h at 50 °C. The Gibson mix was prepared by combining 80 μl of 1 M Tris Base, pH 8 (648310-M), 8 μl of 1 M MgCl2 (M8266), 10 μl of Phusion polymerase (M0530L), 6.4 μl of a 25 mM equimolar dNTPs mix (R0181), and 0.64 μl of T5 Exonuclease (M0663L). The *IDS.IGF2del_co* variant was generated by PCR amplification of the entire *IDS.IGF2co* vector, excluding the sequence encoding amino acids 30–40 of the human IGF2 tag, using primers with a 20 bp overlap. Gibson assembly for the *IDS.IGF2del_co* vector was performed using only the backbone. *GFP* vector was generated as described previously (Catalano et al, 2023). Lentiviral particles were generated as previously shown (Liang et al, 2022a). Functional viral titers were measured by transduction of HeLa cells as previously shown (Liang et al, 2022a, 2022b).

## Transduction of HMC3 and uptake of IDS versions

For analysis of processing and specific activity, HMC3 were transduced at MOI 13.5 with lentiviral vectors encoding the IDS versions under study. For analysis of secretion, medium was refreshed 24 h before harvest. For measuring of the relative specific activity, IDS activity levels were measured in five 2-fold dilution of media. IDS activity was measured in media samples and cell lysate using 4-Methylumbelliferyl-α-L-idopyranosiduronic acid 2-sulphate disodium salt (4 MU)-analysis as described below. Western blot analysis was performed on cell lysate and media samples as described below. For uptake experiments, HMC3 cells were transduced at MOI 10 and cultured for a week in complete medium (DMEM supplemented with 1% penicillin/streptomycin (PS, Gibco 15070) and 10% fetal bovine serum (FBS-12A Capricorn Scientific)) before producing conditioned media by refreshing with complete medium 24 h before harvest. Conditioned medium containing secreted IDS proteins was centrifuged at 300× *g* for 5 min and filtered (0.45-μm filter, Millipore). Media were aliquoted and stored at −70 °C. Protein concentration was measured by IDS

sandwich ELISA. Uptake experiments were performed on primary MPS II fibroblasts, bEND.3 cells, HMC3 and SH-SY5Y (seeding density 24 h before start of uptake: 100,000 cells/cm²) via incubation in conditioned media for 24 h at the indicated IDS-protein concentration. SH-SY5Y were expanded in DMEM/F12 (Gibco, 11320033) supplemented with 10% FCS and 1% NEAA (Gibco, 11140050) and 1% Pen/Strep. Differentiated into neurons was performed via culturing in Neurobasal medium (Gibco, 21103049) supplemented with 2% B27 (Gibco, 17504044), 10 μM retinoic acid (Sigma, R2625) and 1% GlutaMax (Gibco, 35-050-061) and 1% Pen/Strep. Differentiation of SH-SY5Y cells was started at day 1, after plating at a confluency of 100,000 cells/cm² at day 0. Uptake in SH-SY5Y cells was performed at day 7 after start of differentiation.

## Western blotting

Protein extracts from HMC3 cells and media supernatant were obtained as described below (IDS activity section). Western blotting and protein concentration assay were performed as described previously (Catalano et al, 2023). Protein concentration was determined using a Pierce™ BCA Protein Assay Kit (Thermo Fisher Scientific) according to the manufacturer's instructions, while a total of 12.9 μg (transfection—cells) of total protein or 12 μl (transfection—medium) of samples were used for SDS-PAGE analysis. Proteins were transferred to nitrocellulose blotting membranes (GE Healthcare) and blocked with 5% non-fat milk powder in PBS and probed by overnight incubation at 4 °C with goat anti-human IDS (1:1000, R&D Systems) in 5% non-fat milk powder in PBS supplemented with 0.1% Tween. Proteins were detected with IRDye 680 RD secondary antibodies (1:10,000; LI-COR Biosciences, Lincoln, NE) and were imaged using the Odyssey Infrared Imaging System (LI-COR Biosciences, Lincoln, NE). Protein content was quantified using Fiji; equal loading was determined by quantification of the total bands using the stain-free signal on the same gel used for immunoblotting.

## IDS enzyme activity and postprandial glucose levels

Brain samples (right hemisphere) were disrupted as described previously (Catalano et al, 2023). Extracts from HMC3, HEK 293T, bone marrow and WBC were obtained in 100 μl of deionized water by snap-freezing on dry-ice and mechanical disruption. Debris was pelleted by centrifugation at 10,000 rpm for 5 min. Medium from HMC3 and HEK 293T was centrifuged at 10,000 rpm for 5 min to remove debris. To obtain plasma, blood samples were mixed 3:1 with 4% Citrate Buffer (6132-04-3 Sigma) and plasma was separated by centrifugation at $2000 \times g$ for 10 min at 4 °C. Postprandial glucose levels were measured in plasma using a glucometer (OGM-191, Swiss Point of Care). Lysate from HMC3 and HEK 293T cells was diluted 50 times in 0.2% bovine serum albumin (BSA) in water (BSA, Sigma) for measurement of IDS enzyme activity, and two times in water for measurement total protein. Medium from HMC3 and HEK 293T cells was diluted ten times in 0.2% BSA for measurement of IDS activity. Lysate from bone marrow was diluted 51 times in 0.2% BSA in water for measurement of IDS enzyme activity and 6 times in water for measurement of total protein. Plasma was diluted 21 times in 0.2% BSA in water for measurement of IDS enzyme activity and 31 times

in water for measurement of total protein. Lysate from WBC was diluted 21 times in 0.2% BSA in water for measurement of IDS enzyme activity and 3 times in water for measurement of total protein. IDS activity was measured as described in previously (Catalano et al, 2023) and using 4-Methylumbelliferyl-α-L-idopyranosiduronic acid 2-sulphate disodium salt (Biosynth, Carbosynth; 2.5 mM in 0.2 M Na-acetate buffer, pH 5) substrate and human α-L-Iduronidase (5 μg/ml in 0.1% BSA; R&D Systems) for 4 h at 37 °C (Voznyi et al, 2001). 1, 0.1, 0.01, 0.003 ng/μl of Elaprase were measured as a control. For normalization of enzyme activity in bone marrow per VCN in bone marrow (Fig. EV3D,E), IDS activity and VCN were fitted to a Michaelis–Menten model (Activity = $(5203 \ast VCN)/(0.6086 + VCN)$). The cohort average VCN ($VCN_{ave}$) was then used as a reference to normalize the measured activity of each sample using the formula: $Activity_{norm}$ = measured activity$\ast(VCN_{ave}/(0.6086 + VCN_{ave}))\ast((0.6086 + measured\ VCN)/$ measured VCN)).

## ELISA

IDS sandwich ELISA was performed using the DuoSet kit according to manufacturer's instructions (R&D, DY449-05). For IDS ELISA in plasma, plasma samples were diluted 1:700 in 1% BSA (Sigma) in PBS. LRP-1 direct ELISA was performed as described previously (Catalano et al, 2023). For competition ELISAs with the IGF2R (repeats 11-13; R&D, 2447-GR-050), IGF1R (R&D, 305-GR-050) and IR-A (R&D, 1544-IR-050/CF), plates were coated with mouse anti-HIS tag antibody (R&D, MAB050) at a concentration of 1 μg/ml (IGF1R and IGF2R) or 2 μg/ml (IR-A) and incubated O/N at RT. The following day, plates were washed three times with 0.05% Tween (Sigma) in PBS and blocked for 2 h at RT with 1% BSA (Sigma) in PBS. Receptors were added after dilution in 1% BSA in PBS at a concentration of 1 μg/ml (IGF2R and IGF1R) or 2 μg/ml (IR-A) and incubated 2 h at RT. After washing, 100 μl of conditioned media containing IDS-tagged versions at the concentration indicated and either 8 μM of biotinylated IGF2 peptide (AXU020, CellSciences), 4 μM of biotinylated IGF1 peptide (AQU050, CellSciences), or 0.8 μM of biotinylated insulin (AYU100, CellSciences) were applied to the wells and incubated 2 h at RT. Biotinylated ligands were detected by incubation with 100 μl of streptavidin-HRP (R&D, DY998) for 20 min at RT and incubation in 100 μl of substrate solution (R&D, DY999) for 20 min at RT. Reactions were stopped with 50 μl of 2 N HCl and signal was measured using a microplate reader. For IGF2R, IGF1R, IR-A and LRP-1R ELISAs (Fig. 2A–D), conditioned media was produced by transduction of HMC3 cells, and by incubation in DMEM 1% penicillin/streptomycin (PS, Gibco 15070) and 0.1% BSA (Sigma) for 24 h. Media were filtered through a 0.45-μm filter, and concentrated using protein concentrators (Thermo Scientific, 88536). Mature human IGF2 peptide was purchased (MU100, CellSciences).

## Lentiviral hematopoietic stem cell transduction and transplantation procedures

HSPC-LVGT was conducted in one experiment comprising all the experimental groups. In the same experiment, other experimental groups were included and are shown in the second set of experiments (*Experiment 2*) reported in Catalano et al, 2023. For this reason, this

article and Catalano et al, 2023 share the *IDSco, IDS.IGF2co* and *GFP* groups. Transplantation procedures were performed as previously shown (Catalano et al, 2023). Briefly, Hematopoietic stem and progenitor cells were enriched from 8-week to 4-month-old male *Ids^{y/−}* CD45.1 mice by lineage depletion using the Lineage Cell Depletion Kit – mouse (Miltenyi Biotec). Lin⁻ cells were cultured in StemSpan SF expansion medium (Stemcell Technologies) supplemented with recombinant murine thrombopoietin (10 ng/ml, R&D Systems), recombinant murine stem cell factor (100 ng/ml, R&D Systems) and recombinant murine FMS-like tyrosine kinase 3 murine ligand (50 ng/ml, R&D Systems). Cells were transduced without transduction enhancers over 24 h at the indicated multiplicity of infection (MOI) with concentrated lentiviral particles and incubated at 37 °C with 5% $CO_2$. The day after, $1 \times 10^6$ transduced Lin⁻ cells (200 μl of cells suspension in PBS) were transplanted intravenously into 8/11 weeks-old male *Ids^{y/−}* CD45.2 recipients, previously subjected to 9 Gy total body irradiation (TBI) using the Gammacell 40 irradiator (Atomic Energy of Canada LTD, Ontario, Canada). No normalization for body weight was applied to the number of cells transplanted.

## Mass spectrometry analysis of heparan sulfate and dermatan sulfate in brain

Brain samples were prepared as described previously using highly sensitive liquid chromatography tandem-mass spectrometry (LC-MS/MS) (Tanaka et al, 2018; Catalano et al, 2023). Internal standard for the analysis of heparan sulfate and dermatan sulfate were obtained by deuterio-methanolysis of heparan sulfate and dermatan sulfate as described in Catalano et al, 2023. An internal standard working solution was prepared by 25-fold dilution of HS and DS internal standards in 10 mM ammonium acetate in 90:10% (v/v) acetonitrile:water. 125 μl of brain homogenates were transferred into borosilicate tubes and dried under nitrogen. After addition of 25 μl of 2,2-dimethoxypropane and 300 μl of 3 M HCl-methanol (Merck, 90964), samples were incubated for 75 mins at 65 °C and dried under nitrogen. Samples were resuspended in 150 μl of internal standard working solution. In total, 13 μl of sample preparation were mixed with 187 μl 10 mM ammonium acetate in 90:10% (v/v) acetonitrile:water. LC-MS/MS was performed on a Sciex 5500 QTrap (tandem) mass spectrometer coupled to a Waters Acquity UPLC system.

## Histopathology and immunohistochemistry

After perfusion, brain, liver, spleen and heart were excised and the left hemisphere was fixed in methacarn (v/v – 60% absolute methanol, 30% chloroform, 10% glacial acetic acid), dehydrated in 50% and 70% ethanol for 24 h and processed in paraffin (histokinette). Embedded samples were sectioned at either 10 μm (liver and spleen) or 8 μm (brain and heart) and either stained with alcian blue or processed for immunohistological staining. For alcian blue staining, sections were rehydrated and equilibrated in 0.1 N hydrochloric acid (Sigma) for 30 s, followed by staining in 1% Alcian Blue 8GX (Sigma) pH 1 for 30 min. Sections were incubated in 0.1 N hydrochloric acid (Sigma) for 30 s and stained in 0.1% nuclear fast red (Sigma) in 0.06 M aluminum sulfate hexadecahydrate (Sigma) for 5 min. Sections were rinsed in 95% ethanol, dehydrated and mounted in Entellan mounting medium (Sigma). Alcian blue stainings were scored according to the rules shown in

Table EV2. For LAMP1 immunostaining in heart, sections were blocked for endogenous peroxidase in 3% hydrogen peroxidase (dilution 1:2 in $dH_2O$ of a 6% v/v solution BMS-2110-1E, PHC Corporation), and for endogenous avidin and biotin for 15 mins at room temperature according to the manufacturer's (SP-2001, Vector Laboratories). Sections were then blocked for 30 min in staining buffer (3% BSA, 3% goat serum, 0.3% Triton X-100 in PBS) at room temperature. Sections were stained with LAMP1 primary antibody (rat anti-LAMP1,1:500, ab25245 Abcam) in staining buffer O/N at 4 °C. The day after, sections were incubated with goat anti-rat antibody biotinylated (1:200, 554014 BD Pharmingen). Sections were then incubated with streptavidin-HRP (1:50, DY998 R&D systems) in staining buffer for 60 mins at room temperature and incubated in impact DAB (SK-4105) for 2 mins. Sections (LAMP1 in heart and alcian blue staining) were mounted in Entellan (Sigma) and scanned by a NanoZoomer 2.0 (Hamamatsu Photonics, Japan). For immunohistochemical staining in brain, sections were rehydrated and blocked for 30 mins in staining buffer (3% BSA, 3% goat serum, 0.3% Triton X-100 in PBS) at room temperature. Sections were stained with primary antibodies detecting either LAMP1 (rat anti-LAMP1,1:500, ab25245 Abcam), CD68 (rat anti-CD68 IgG, 1:300, MCA1957T Bio-Rad), or GFAP (rabbit anti-GFAP IgG, 1:500, Sigma-Aldrich, G9269). After incubation overnight at 4 °C, sections were washed with PBS and stained with the appropriate secondary antibody conjugated to Alexa Fluor® 488 or Alexa Fluor® 594 (1:500, ThermoFisher Scientific) for 1 h. All sections were counterstained with Hoechst33258 (1:15000, Life Technologies) to stain nuclei. Pictures were obtained using an ST5LIA_311 Leica Stellaris5 confocal microscope. CD68 staining was quantified by counting the number of CD68-positive cells in three areas per brain region. The obtained values were averaged and normalized by the area analyzed to obtain the final number of CD68-positive cells per mm². CD68 immunoreactivity was quantified by counting the number of CD68-positive cells in three distinct areas per brain region of three different mice. The resulting values per mouse were averaged and normalized by the analyzed area to determine the final number of CD68-positive cells per mm². For LAMP1 and GFAP stainings, fluorescence intensity was measured using ImageJ. Briefly, the channels displaying LAMP1 or GFAP signals in two distinct areas per brain region of three different mice were isolated, and the total fluorescence area was quantified after applying thresholding and LUT inversion. Final values were normalized by the area analyzed.

## Flow cytometry analysis of chimerism

Flow cytometry analysis of chimerism in bone marrow was performed as previously described (Gleitz et al, 2018). Briefly, frozen vials of bone marrow were isolated from the right hindlimb tibia and femur, and from the humerus bones. Staining was performed in 4% FCS in PBS with FITC-mouse anti-mouse CD45.2 (BD Bioscience, 553772) and PE-mouse anti-mouse CD45.1 (BD Bioscience, 553776). Every experiment was performed with single-staining controls, unstained controls and isotype-stained controls (mouse IgG2a-FITC, BD Bioscience 349051; mouse IgG2a-PE, 349053). Measurement of chimerism was performed using a BD LSRFortessa and a FACS DIVA software recording 20,000 or more events per sample, while analysis was performed using FlowJo v10.

### Quantitative polymerase chain reaction of vector copy number and chimerism

Vector copy number (VCN) and chimerism in bone marrow and brain were determined by quantitative polymerase chain reaction (qPCR) as described previously (Catalano et al, 2023). Briefly, VCN was measured using iTaq Universal SYBR Green Supermix (Bio-Rad, Hercules, CA) and primers specific for *PSI* (FW: 5'-*CAGGACTCGGCTTGCTGAAG*-3'; RV: 5'-*TCCCCCGCTTAATAC TGACG*-3') and mouse *Albumin* (FW: 5'-*ACTTTGAGTGTAGCA-GAGAGGAACC*-3'; RV: 5'-*CTCTTCACTGACCTAAGCTACTCCC*-3'). Chimerism was determined as described previously (Catalano et al, 2023) using primers specific for the *Cd45.1* allele (FW: 5'-*CTGAGCCTGCATCTAAACCTGATC*-3'; RV: 5'-*TCACCTTCATA AAAGCCTTGT AGCTC*-3'). Reactions were performed and measured in a CFX96 real-time PCR detection system and analyzed by CFX Manager 3.0 (Bio-Rad, Hercules, CA).

### Micro-computed tomography (μCT) analysis of bone microarchitecture

Micro-computed tomography (μCT) imaging of the right ankle was performed using Quantum GX imaging system (PerkinElmer). For CT imaging, samples were fixed in 4% paraformaldehyde in PBS at 4 °C for 24 h, followed by washing in PBS. Samples were scanned while in PBS in 0.5 -ml polypropylene tubes. During scans, the following parameters were applied: 90 kV, 88 μA, 36-μm isotropic voxel size, 18 mm Field of View (FOV) and a Cu 0.06 mm + Al 0.5 mm standard filter. The acquired raw data were reconstructed with a resolution of 10 μm using Quantum GX2 software (PerkinElmer). Segmentation of *talus, navicular-lateral cuneiform* and *medial cuneiform* from the CT scans and 3D analysis were performed using the automated ASBMR measurement of the AccuCT software (PerkinElmer).

### Statistics

Statistical analysis was performed GraphPad Prism (version 9.0.0. for Windows, San Diego, California USA, www.graphpad.com). All results are presented as mean ± SD and each data point is shown. Normality tests were performed by Shapiro–Wilk Test. Multiple comparison analysis was performed by one-way ANOVA with Bonferroni's correction. Lysosomal pathology and neuroinflammation quantification in brain was analyzed by two-way ANOVA with Bonferroni's correction using brain area and viral vector as categorical variables. Non-linear regression models were used to describe the relationship between heparan sulfate and other variables such as, VCN, IDS activity in brain, IDS activity in plasma, as well the relationship between VCN and activity in bone marrow, and signal and input concentration during functional ELISA. This analysis was performed using GraphPad Prism built-in models, such as one-phase decay and Michaelis–Menten. Linear regression analysis was performed using GraphPad Prism.

## Data availability

This study includes no data deposited in external repositories.

The source data of this paper are collected in the following database record: biostudies:S-SCDT-10_1038-S44321-025-00314-3.

## Peer review information

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

## Acknowledgements

This work was supported by the "Finding a cure for Hunter Disease" Foundation, the Sophia Foundation (grant S18-59), Metakids (grants 2018-083 and 2021-094), the Prinses Beatrix Spierfonds (grant W.OP20-04), the Hersenstichting (grant DR-2021-00386), Dioraphte (project 20210367), the For Wishdom Foundation, and by Cure4Life: a Dutch Research Agenda grant from NWO (grant *NWA.1389.20.049*). This work was supported through the use of imaging equipment provided by the Applied Molecular Imaging Erasmus MC facility. We thank Prof. Dr. Brian Bigger for kindly providing the transgenes *IDSco* and *IDS.ApoEIIco*. We thank Eva Niggl for kindly providing SH-SY5Y cells.

## Author contributions

**Fabio Catalano**: Conceptualization; Data curation; Formal analysis; Investigation; Methodology; Writing—original draft; Project administration; Writing—review and editing. **Dejan Stevic**: Investigation. **Giacomo Zundo**: Investigation. **Tessa F Huizer**: Data curation. **Zina Dammou**: Data curation. **Eva C Vlaar**: Data curation. **Drosos Katsavelis**: Data curation. **Jeroen C van den Bosch**: Data curation. **Hannerieke J M P van den Hout**: Resources. **Esmeralda Oussoren**: Resources. **Ans T van der Ploeg**: Resources. **George J G Ruijter**: Resources. **Gerben Schaaf**: Supervision. **W W M Pim Pijnappel**:

Conceptualization; Data curation; Funding acquisition; Investigation; Methodology; Writing—original draft; Project administration; Writing—review and editing.

Source data underlying figure panels in this paper may have individual authorship assigned. Where available, figure panel/source data authorship is listed in the following database record: biostudies:S-SCDT-10_1038-S44321-025-00314-3.

## Disclosure and competing interests statement

Ans T van der Ploeg (AvdP) has received consulting fees from Sanofi Genzyme and has provided consulting services, participated in advisory boards meetings and received grants for premarketing studies and research from industries via agreements between Erasmus MC and the industry. WWM Pim Pijnappel (WP) is cofounder and scientific advisor of LentiCure.

# Expanded View Figures

**Figure EV1.  IDS.SWAP processing and secretion.**

(A, B) Binding of IDS.IGF2del to CI-M6P/IGF2R and IR-A was assessed using two methods: a direct CI-M6P/IGF2R ELISA (A) and a competitive IR-A ELISA using biotinylated insulin (B). (C) Cellular uptake test of SWAP variants containing an ApoE insert into MPS II fibroblasts (24-h uptake). IDS.IGF2 and IDS.ApoEII refer to versions of the IDS protein tagged at the C-terminus with either IGF2 or ApoEII (tandem repeat of ApoE) tags via a flexible linker, as previously described (Gleitz et al, 2018; Catalano et al, 2023). IDS.SWAP-ApoE_No Cys refers to a SWAP version without a cysteine residue at each end of the insertion. IDS.SWAP-ApoE_with Cys refers to a SWAP version with a cysteine residue at each end of the insertion. The adjusted $P$ values were as follows: IDS.IGF2 vs. IDS.ApoEII $P = <0.0001$, IDS.ApoEII vs. IDS.SWAP-ApoE_No Cys $P = 0.0085$, IDS.ApoEII vs. IDS.SWAP-ApoE_with Cys $P = 0.0004$, IDS.SWAP-ApoE_No Cys vs. IDS.SWAP-ApoE_with Cys $P = 0.0395$. (D) Immunoblot analysis of IDS proteins using an anti-IDS antibody. The upper panel shows SDS-PAGE analysis of the cell lysate of HMC3 cells 10 days after transduction at MOI 13.5 with lentiviral vectors encoding the indicated fusion proteins. The lower panel shows SDS-PAGE analysis of the medium supernatant of HMC3 transduced as above. Both upper and lower panels were separated to eliminate an irrelevant condition. Medium was refreshed 24 h before sample collection. HMC3 cells transduced with a GFP vector and 0.1 μg of Elaprase served as negative and positive controls, respectively, during SDS-PAGE analysis. (E) Protein load for the western blot in Fig. EV1D and relative quantification (shown in Fig. 1C). Bottom panel was separated to eliminate an irrelevant condition. The stain free signal was quantified from the same gel used for immunoblot analysis to determine the total protein load. (F) Immunoblot analysis of IDS protein in five 2-fold dilutions of medium supernatant from HMC3 transduced with the indicated lentiviral vectors. 1 μg of Elaprase served as control. This was used to measure specific IDS activity in Fig. 1E. Lower panel was separated to eliminate an irrelevant condition. (G) Secreted IDS protein activity per VCN after transduction of HMC3 cells at MOI 5 and MOI 50. The adjusted $p$ value for *MOI 5* IDS.IGF2 vs. IDS.SWAP-ApoE was 0.0283. Data information: data are presented as means ± SD. In (C), a total of 8 conditions were analyzed by one-way ANOVA; significant pairwise differences among the 4 relevant conditions are reported. In (G) data were analyzed by two-way ANOVA followed by Bonferroni's multiple testing correction. In (A–C) $n = 2$. In (G) $n = 3$. *$P \le 0.05$; **$P \le 0.01$; ***$P \le 0.001$; ****$P \le 0.0001$. Significant comparisons are indicated by brackets.

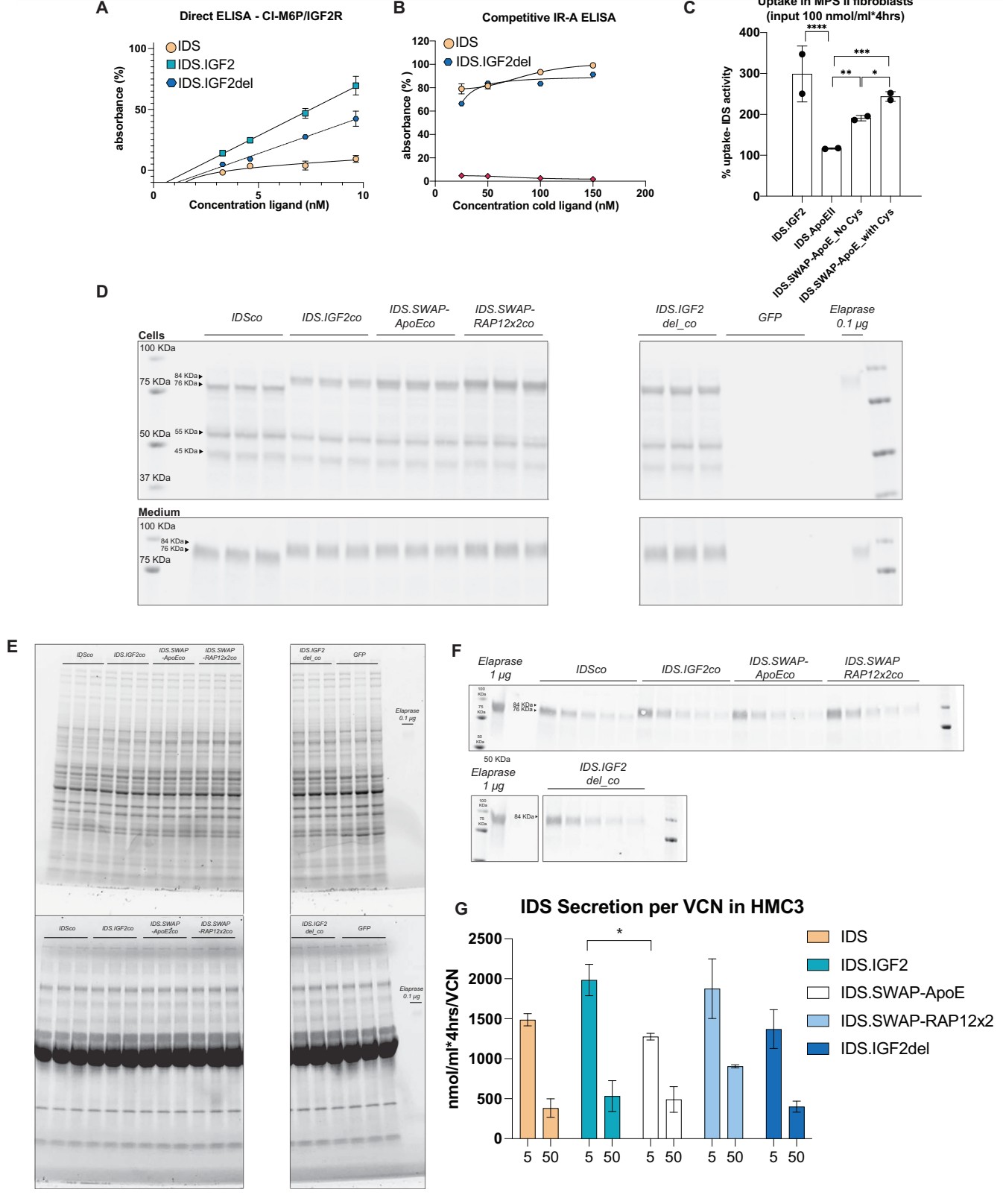

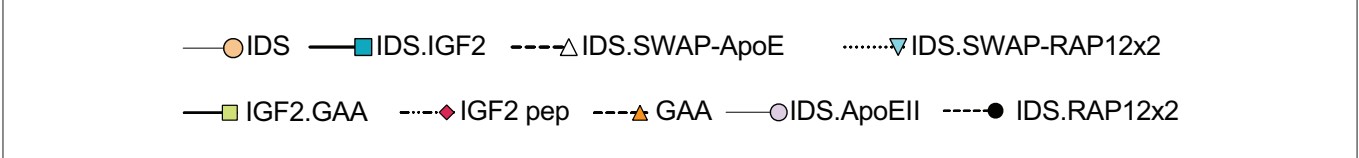

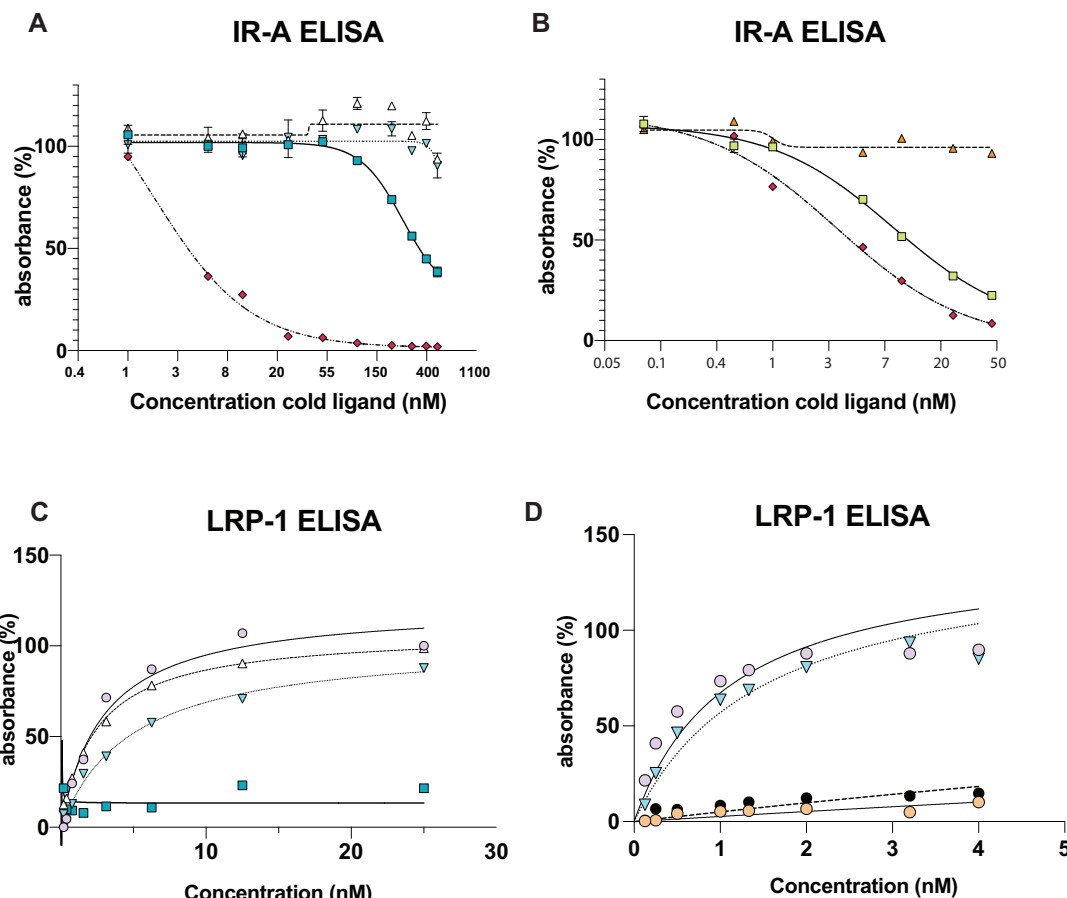

**Figure EV2. CI-M6P/IGF2IGF2R, IR-A, LRP-1 ELISAs.**

(A, B) competitive IR-A ELISA using biotinylated insulin. Concentrations of competing ligands are indicated. Ligands were derived from conditioned media produced in HEK 293T cells. (C, D) direct LRP-1 ELISA. Concentrations of ligands are indicated. Data information: data are means ± SD. Regression analysis is shown in Table EV1. $n = 2$ (A–C) or $n = 1$ (D).

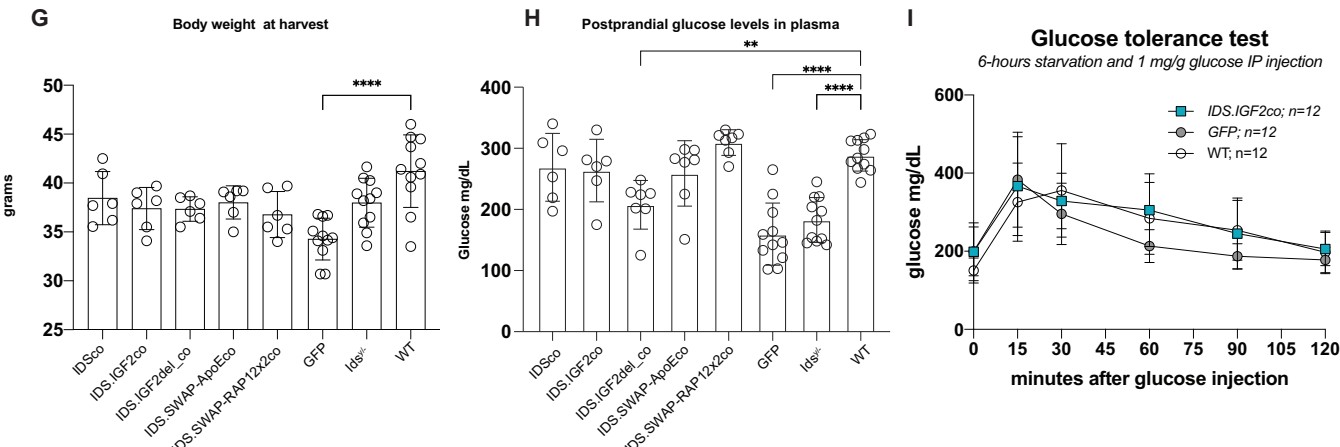

◀ **Figure EV3. Supraphysiological levels of IDS activity in hematological samples, body weight, and glucose homeostasis.**

(A–C) IDS enzyme activity in hematological tissues. Fold-increase versus WT in bone marrow (A), WBC (B), and plasma (C). In (B, C) the adjusted *P* values were as follows: (B) *IDSco* vs. *IDS.SWAP-ApoEco P* = <0.0001, *IDSco* vs. *IDS.SWAP-RAP12x2co P* = <0.0001, *IDSco* vs. *IDS.IGF2del_co P* = 0.0317, *IDS.IGF2co* vs. *IDS.SWAP-RAP12x2 P* = 0.0146, *IDS.IGF2del_co* vs. *IDS.SWAP-RAP12x2 P* = 0.0492. (C) *IDSco* vs. *IDS.IGF2co P* = 0.0002, *IDSco* vs. *IDS.IGF2del_co P* = 0.0030, *IDSco* vs. *IDS.SWAP-ApoEco P* = 0.0020, *IDSco* vs. *IDS.SWAP-RAP12x2co P* = <0.0001, *IDS.IGF2del_co* vs. *IDS.SWAP-RAP12x2co P* = 0.0038, *IDS.SWAP-ApoEco* vs. *IDS.SWAP-RAP12x2co P* = 0.0058. (D, E) Relationship between IDS activity in bone marrow and VCN in bone marrow. (F) Sandwich ELISA to quantify IDS protein levels in plasma. (G) Body weight at harvest after gene therapy. The adjusted *P* value for *GFP* vs. WT was <0.0001. (H) Postprandial glucose levels at harvest. The adjusted *P* values were as follows: *IDS.IGF2del_co* vs. WT *P* = 0.0015, *GFP* vs. WT *P* = <0.0001, *Ids*$^{y/-}$ vs. WT *P* = <0.0001. (I) Glucose tolerance test. Data information: data are means ± SD. Regression analysis of (D) is shown in Table EV1. In (A–C, E, G, H) data were analyzed by one-way ANOVA followed by Bonferroni's multiple testing correction. In (H), comparisons were made against WT. In (F) data were analyzed by *t* test. In (A–C, E) *n* = 6 for *IDSco, IDS.IGF2co, GFP, Ids*$^{y/-}$, and *WT*, *n* = 7 for *IDS.IGF2del_co, IDS.SWAP-ApoEco, IDS.SWAP-RAP12x2co*. In (F) *n* = 6. In (G, H) for *IDSco, IDS.IGF2co, IDS.IGF2del_co, IDS.SWAP-ApoEco, IDS.SWAP-RAP12x2co n* = 6; for *GFP, Ids*$^{y/-}$, and *WT n* = 11. In (I) *n* = 12. *$P \leq 0.05$; **$P \leq 0.01$; ***$P \leq 0.001$; ****$P \leq 0.001$. Significant comparisons are indicated by brackets.

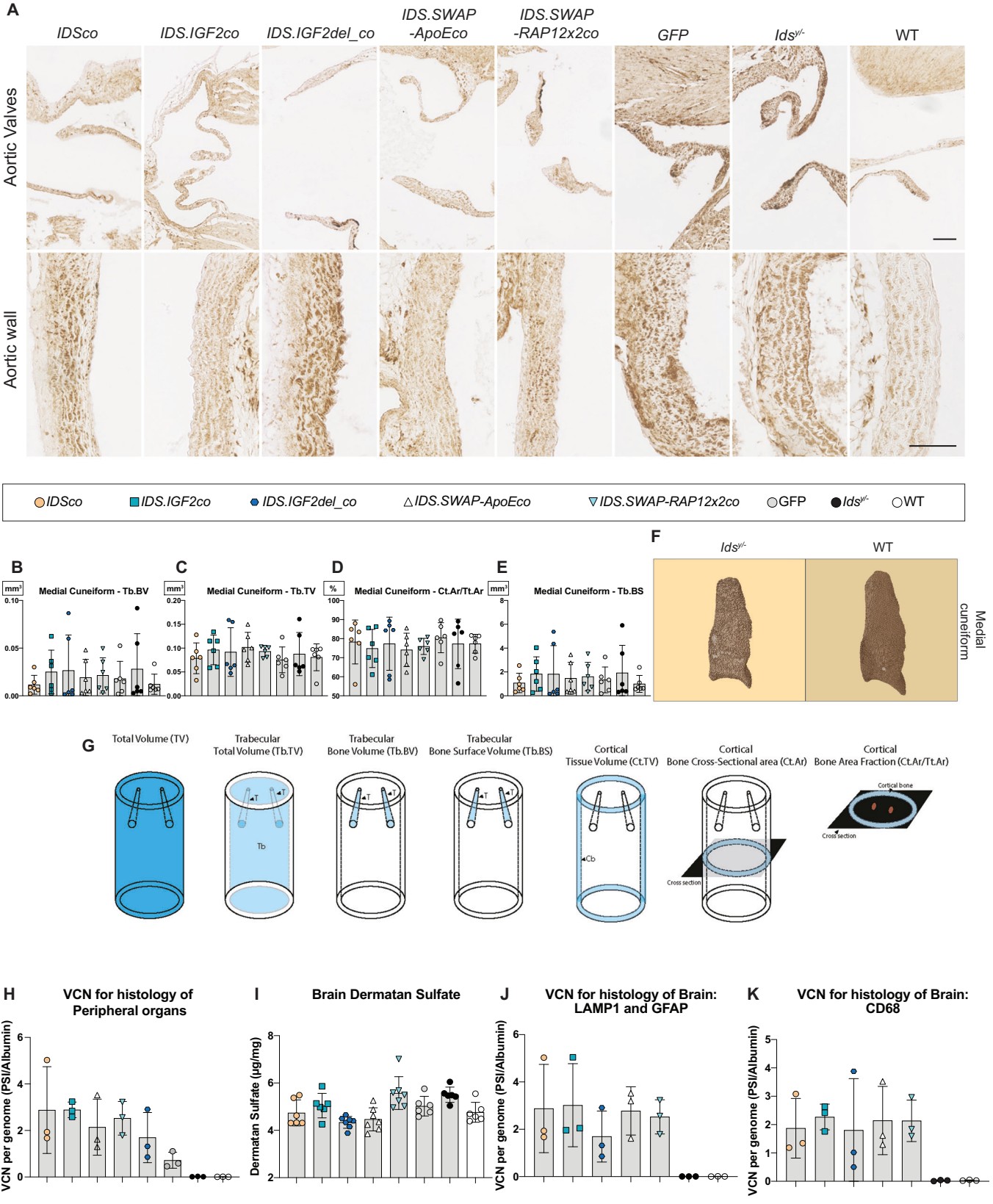

◀ **Figure EV4.  LAMP1 staining in heart, microarchitecture of the *medial cuneiform*, brain dermatan sulfate levels, and bone marrow VCN for histology.**

(A) LAMP1 staining in heart valves and aortic wall. $n = 3$. Scale bar $= 100$ µm. (B–E) Microarchitecture analysis of the *medial cuneiform*. (F) 3D rendering of reconstructed µCT scans of the *medial cuneiform*. (G) Explanatory drawings of µCT parameters. The volume/area measured by the µCT parameters is highlighted in blue. (H) VCN per genome in bone marrow of mice used for histological analysis of peripheral tissues. (I) Dermatan sulfate levels in total brain homogenates measured by mass-spectrometry. (J, K) VCN per genome in bone marrow of mice used for immunostaining of LAMP1 and GFAP (J), or CD68 (K) in brain. Data information: data are presented as means ± SD and were analyzed by one-way ANOVA with Bonferroni's correction. In (B–E) $n = 6$; in (H, J, K) $n = 3$; in (I) $n = 6$ (*IDSco, IDS.IGF2co, GFP, Ids$^{y/-}$, WT*) or $n = 7$ (*IDS.IGF2del_co, IDS.SWAP-ApoEco, IDS.SWAP*-RAP12x2co). Tb: trabecular bone; T: trabeculae; Cb: cortical bone. Data are presented as means ± SD and were analyzed by one-way ANOVA with Bonferroni's correction.

GFP treated mice

LAMP1 staining          GFAP staining          CD68 staining

**Figure EV5. Full scans of sagittal brain sections of GFP-treated mice stained for LAMP1, GFAP or CD68.**

Example of sagittal brain sections from GFP-treated mice stained for LAMP1 (**left panel**), GFAP (**central panel**) or CD68 (**right panel**). $n = 3$. Scale bar $= 1$ mm.

