## [Peer Review File · EMBO Molecular Medicine]

Domain-substituted IGF2 tag modulates targeting of lentiviral gene therapy for Hunter syndrome

Fabio Catalano, Dejan Stevic, Giacomo Zundo, Tessa Huizer, Zina Dammou, Eva Vlaar, Drosos Katsavelis, Jeroen van den Bosch, Hannerieke van den Hout, Esmeralda Oussoren, Ans Van der Ploeg, George Ruijter, Gerben Schaaf, and W.W.M. Pim Pijnappel

Corresponding author: W.W.M. Pim Pijnappel (w.pijnappel@erasmusmc.nl)

Review Timeline:

Submission Date:	10th Jan 25
Editorial Decision:	21st Feb 25
Revision Received:	27th May 25
Editorial Decision:	8th Jul 25
Appeal:	14th Aug 25
Editorial Decision:	25th Aug 25
Revision Received:	12th Sep 25
Accepted:	16th Sep 25

Editor: Zeljko Durdevic

Transaction Report:

21st Feb 2025

Dear Dr. Pijnappel,

Thank you for the submission of your manuscript to EMBO Molecular Medicine, and please accept my apologies for the unusual delay in getting back to you. We have now received feedback from two of the three reviewers who agreed to evaluate your manuscript. As the referee #2 will unfortunately not be able to return his/her report in a timely manner, and given that both reviewers provide similar recommendations, we prefer to make a decision now in order to avoid further delay in the process. Should referee #2 provide a report, we will send it to you, with the understanding that we will not ask for an additional revision.

As you will see from their reports pasted below, both referees recognize potential interest of the manuscript but also raise serious and partially overlapping concerns that should be addressed in a major revision. If you would like to discuss further the points raised by the referees, I am available to do so via email or video. Let me know if you are interested in this option.

We would welcome the submission of a revised version within three months for further consideration. Please let us know if you require longer to complete the revision.

I look forward to receiving your revised manuscript.

Yours sincerely,

Zeljko Durdevic

We require:

2) Individual production quality figure files as .eps, .tif, .jpg (one file per figure). For guidance, download the 'Figure Guide PDF': (<https://www.embopress.org/page/journal/17574684/authorguide#figureformat>).

3) A .docx formatted letter INCLUDING the reviewers' reports and your detailed point-by-point responses to their comments. As part of the EMBO Press transparent editorial process, the point-by-point response is part of the Review Process File (RPF), which will be published alongside your paper.

4) A complete author checklist, which you can download from our author guidelines (<https://www.embopress.org/page/journal/17574684/authorguide#submissionofrevisions>). Please insert information in the checklist that is also reflected in the manuscript. The completed author checklist will also be part of the RPF.

6) It is mandatory to include a 'Data Availability' section after the Materials and Methods. Before submitting your revision, primary datasets produced in this study need to be deposited in an appropriate public database, and the accession numbers and database listed under 'Data Availability'. Please remember to provide a reviewer password if the datasets are not yet public (see <https://www.embopress.org/page/journal/17574684/authorguide#dataavailability>).

12) Author contributions: You will be asked to provide CRediT (Contributor Role Taxonomy) terms in the submission system. These replace a narrative author contribution section in the manuscript.

13) A Conflict of Interest statement should be provided in the main text.

14) Every published paper now includes a 'Synopsis' to further enhance discoverability. Synopses are displayed on the journal webpage and are freely accessible to all readers. They include a short stand first (maximum of 300 characters, including space) as well as 2-5 one-sentences bullet points that summarizes the paper. Please write the bullet points to summarize the key NEW findings. They should be designed to be complementary to the abstract - i.e. not repeat the same text. We encourage inclusion of key acronyms and quantitative information (maximum of 30 words / bullet point). Please use the passive voice. Please attach these in a separate file or send them by email, we will incorporate them accordingly.

15) Include a Reagents and Tools Table as part of the Methods section, which can be downloaded from our author guidelines (<https://www.embopress.org/page/journal/17574684/authorguide#structuredmethods>)

***** Reviewer's comments *****

Referee #1 (Comments on Novelty/Model System for Author):

Although the initial data are compelling, some of the comparisons and claims made in the latter part of the paper about the superiority of SWAP tags over the existing IGF2 tag are overstated and need to be qualified and other sections to be clarified.

Referee #1 (Remarks for Author):

Catalano et al describe a novel engineered IGF2 peptide that has reduced off target binding to insulin receptor and IGF1 receptor, whilst maintaining its IGF2R binding capability. This has been a long-standing question mark in the lysosomal disease field for those employing enzyme replacement therapy with IGF2 tagged enzymes. IGF2 tags bind to the M6PR/IGFR protein (at distinct domains) meaning that exogenously delivered lysosomal enzymes that are typically taken up by M6PR binding, are more effectively bound to the same protein, facilitating increased uptake. The concern being that off target binding to insulin receptor and IGF1 receptor will block effective insulin or IGF1 trafficking and dangerously disrupt homeostasis. Catalano et al have removed the central loop of IGF2R binding domain (the so called SWAP domain), to insert other targeting sequences, and demonstrate in a genetic setting using hematopoietic stem cell gene therapy, that this reduces binding to Insulin receptor, and to IGF1 receptors, and can be used to insert other targeting peptides with novel receptor binding capabilities. They have further demonstrated the practical application of this in a genetic context using IGF2-SWAP tagged idursulfase enzyme to treat the brains of mice with Hunter disease (a childhood dementia) via hematopoietic stem cell gene therapy.

Although the initial data are compelling, some of the comparisons and claims made in the latter part of the paper about the superiority of SWAP tags over the existing IGF2 tag are overstated and need to be qualified and other sections to be clarified. Importantly, as this paper shares three experimental groups with Catalano et al.(2023). *Mol Ther Methods Clin Dev*; more comparison back to that paper should be made in the discussion. This reduces enthusiasm for the efficacy of the SWAP tags in the Hunter disease model, compared to existing tags, but does not detract from the novel and interesting findings described.

Major

1) Page 4 introduction - mention is made that replacement of the central loop of IGF2 is made with ApoE2, R_{apx12x2} and ApoB. Firstly, the described SWAP-ApoE2 sequence in Figure 1B is in fact the sequence for what is commonly known as ApoE peptide as it is only a single sequence of the receptor binding domain of apolipoproteinE. ApoEII refers to a tandem repeat of the ApoE receptor binding domain (Gleitz 2018 *EMM*) - thus the sequence used here: LRKLRKRL should be referred to as ApoE and LRKLRKRL-LRKLRKRL referred to as ApoEII. The nomenclature ApoE2, used in this paper, should be avoided in any case to avoid confusion with isoforms of the ApoE gene (e.g. ApoE2 that describes the $\epsilon 2$ allele of apolipoproteinE, conferring decreased risk to Alzheimer disease and increased longevity (Shinohara 2020 *elife*, Reimann 2020 *Nat comms*).

Unless the sequence in figure 1 was in fact an error and should be a tandem repeat of LRKLRKRL, we propose that the IDS-SWAPApoE2 construct is renamed throughout as IDS-SWAPApoE to reflect this and avoid further confusion.

2) A second point is that no data are presented for ApoB in the main paper. We would suggest removal of this or full presentation of ApoB data (rather than the snippets in the supplementary figure).

3) A further problem with the SWAP ApoE2 tag being LRKLRKRL, inserted into the central loop of the IGF2 binding domain peptide, is that it is therefore not comparable to the "IDS.ApoE2" tag described in Catalano et al 2023 - which is in fact LRKLRKRL-LRKLRKRL. These are distinctly different domains and this distinction needs to be in place throughout the paper where comparisons are currently made between these products as if they are the same product either outside or within the SWAP context. This is not the case.

4) I understand that this experiment leads on from a previous paper - but there is no description here of the lentiviral vector design used (Presumably the MND promoter driven LV used in reference 10) (Catalano et al.(2023). *Mol Ther Methods Clin Dev*, 101149.). The construction of the LV and details of the overlap of groups should be clearly stated within the introduction and detailed in methods.

5) On the same point I understand that the IDSc_o, IDS.IGF2c_o and WT groups are drawn from the same previous paper

(Catalano et al.(2023). Mol Ther Methods Clin Dev, 101149). Why are comparisons made only to these groups, when the previous paper contained IDS.ApoE2co and IDS.RAP12x2co groups (without the SWAP tags). Surely this would have been a better comparison?

6) Looking at Figure 3G of the current paper, its interesting that the functional IDS assay in plasma of transplanted mice vs VCN doesn't match the relative activity seen in HMC3 cells in vitro (Figure 1E). In figure 3G, the addition of any SWAP tag to coIDS appears to decrease the relationship between plasma IDS activity compared to VCN in bone marrow. I notice also that there is a consistent drop in the IDS activity in WBC vs bone marrow which is unusual as they should share similar VCNs. Notably if you look at the same figure(2F) in (Catalano et al.(2023). Mol Ther Methods Clin Dev, 101149), it is clear that the nonSWAP tagged versions of IDS.ApoE2co (which appears to be a tandem repeat in this case), and the IDS.RAP12x2co perform similarly to IDScO in the IDS activity vs VCN per genome in bone marrow graph (Figure 2F in the former paper). This implies that the SWAP motif of IGF2 actually decreases secreted IDS activity overall compared to when the tags are used without the SWAP. This should be noted and discussed.

7) It should also be noted that all SWAP and IGF2 tags used in this paper give reduced IDS activity per VCN of BM than the native IDScO (Figure 3G), (unlike the IDS.ApoE2 or IDS.Rap12x2). This is a significant downside to the SWAP tags and is not stated in results, nor in discussion.

8) On page 10 a statement is made "IDS.SWAPRAP12x2 can effectively engage LRP-1, in contrast to IDS.RAP12x2, which, as we previously showed¹⁰, failed to bind LRP-1 in a similar ELISA assay." (Figure S2D). Although the supplementary figure compares several different peptides, including the Apoe2 tag alone, RAP12x2 is the only one missing (referenced out to the previous Catalano paper). Due to the importance of this claim, as this would imply increased potential binding to and crossing of LRP receptors on the blood brain barrier of RAP12x2, this assay needs to be repeated including RAP12x2 alongside all the other SWAP peptides, Apoe2, IGF2del, and IGF2 tagged IDS.

9) A statement is made on page 11 that "gene therapy with SWAP vectors effectively prevents peripheral pathology and significantly improves bone abnormalities in Idsy/- mice, with IDS.IGF2co and IDS.SWAP-RAP12x2co achieving near-complete normalization of key parameters." Although the cardiac data are convincing in this regard and the statement could be modified to refer only to the alcian blue data, the bone data presented are not. Notably a significant difference between WT and IDS null is only presented for Talus Tb.TV and Talus Ct.Ar/Tt.Ar (Figure 4 E and F). No significant differences between IDS and any treatment group are shown for talus measurements, therefore the best that can be said is that there is a trend for improvement. This statement could say that where differences in bone pathology were shown there was a trend for improvement, It should not say that peripheral pathology is prevented (except for the specific organs shown -liver, spleen, aortic valves) as this is not shown here.

10) Supplementary figures 4 and 5 add little to the paper and should probably be removed.

11) Figure 5E The data are fairly convincing, but DAPI staining is quite inconsistent across the figure. Were all the sections stained at the same time? This is critical for determining LAMP1 contribution via thresholding.

12) A conjecture on why CD68 in the thalamus (Figure 6) was poorly corrected by all the vectors would be interesting. This didn't seem to be true for GFAP staining but there was residual LAMP1 in the thalamus. I wonder if a marker of activated microglia might have been more informative?

13) The statement in the last paragraph of results is overclaiming. (P20). There is very little (no?) statistical evidence from figures 5-6 that there are differences between SWAP variants and IDS.IGF2co and only some limited data to show that these are better than IDS.IGF2del_co or IDScO. It should be rephrased to "These results indicate that in brain, gene therapy with IDS.SWAP variants and IDS.IGF2 had similar efficacy to each other, all slightly better than IDS.IGF2del_co but with superior efficacy to IDScO, demonstrable in some specific areas such as brain HS reduction and brainstem astrogliosis as well as general improvements in LAMP1 clearance in the brain.

14) Similar claims in the discussion claiming superiority of SWAP tags to IDS.IGF2 and also in the abstract need to be toned down. E.g. Page 22 "SWAP tags...demonstrated a therapeutic advantage in correcting or preventing most peripheral or CNS manifestations compared to untagged IDS and IDS.IGFdel" - this statement is inaccurate.

15) Little comparison to the previous tags used in Catalano 2023 Mol ther clin meth is made. Given that these mice share 3 of the same groups with that paper, a comparison and conjecture should be made with the previous tags described.

Minor

Page 8 - there should be a reference to figure 1E at the end of this sentence I think "which confirmed that SWAP tagging does not affect specific activity of IDS"

Figure S1C has an almost illegible label - please enlarge.

In figure 2 E-H there seems to be some discrepancy between uptake into Bend3 and MPSII fibroblasts between the assay performed here for IDScO and IDS.IGF2co and the same products compared in the same assay in the Catalano 2023 paper (Figure 2G and H.). In the previous paper there was apparently a much greater difference between the two groups. Can this be discussed please.

Figures 3D and 3E and 3F show IDS activity in bone marrow and WBC and plasma respectively. As WT IDS levels are much lower than that of lentiviral vector transduced groups the current graph axis does not allow distinction between the levels of GFP transduced, WT and IDS null mice. Either the Y axis should be split or a log axis presented so the WT IDS activity levels can be seen and compared to LV transduced groups.

Figure 5D shows chimerism levels in the brain of around 1%. Please clarify if this refers to total chimerism as a percentage of total brain cells or microglial chimerism.

Figure 6 should be split into two. It will be illegible in a paper as it is.

Figure S7 adds little to the paper and should probably be removed.

Referee #3 (Remarks for Author):

Pijnappel and colleagues reported novel IGF2-based tag designs, namely SWAP, designed to enhance the efficacy and safety of IGF2 tag-based gene therapy for lysosomal storage disorders, with a specific focus on MPSII. They demonstrate the critical role of IGF2's central loop in binding the insulin and IGF1 receptors, while demonstrating its dispensability for interaction with mannose 6-phosphate /IGF2 receptor, which is a crucial target for enzyme delivery to the lysosomes. The SWAP tags replace the central loop with alternative epitopes to preserve structural integrity and enable high-affinity multimodal receptor targeting. The authors affirm that data in MPSII mice show the superiority of IDS fused to SWAP variants (compared with traditional IGF2 tags) in correcting disease pathology in liver, spleen, heart, bone, and brain. This is an important study in the field, addressing a relevant issue for LSD patients. The authors provided solid data to establish SWAP as an innovative, safer and brain penetrating tag for ex-vivo lentiviral gene therapy for MPSII. However, some results are overstated in the abstract and some points need to be better and clearly addressed/discussed.

- One major flaw of the study is IDS.IGF2co not affecting glucose levels in vivo. This is a bit underwhelming, as the main study rationale for the SWAP design is to have improved tags which still allow for good tissue uptake but do not alter glucose homeostasis. This is a crucial point, which would require some toning down. What is the relevance of these newly developed SWAPs' tags over the IDS.IGF2 (and IDS.IGF2_del), if the latter does not cause glucose levels' fluctuation and has better binding affinity/uptake? Maybe the authors could highlight more SWAPs' enhanced tissue-specific targeting or clarify whether this might be more relevant for ERT more than for HSC-GT.
- Are SWAPs showing lower binding ability and uptake over IDS.IGF2 and IDS.IGF2_del? Please elaborate on this.
- What is the rationale for testing uptake in fibroblasts? Similarly, why choosing a microglia cell line for IDS processing and IDS activity experiments in Figure 1 ? Please explain rationale for these choices
- Fig S1A: Why wasn't IDS.IGF2 tested as well for a direct comparison of efficiency?
- Fig 1D: The level of transduction achieved with each vector is unknown, making interpretation difficult - VCN should have been calculated and activity in cells should be correlated with VCN.
- In Fig 1C I could see a skewing toward the precursor IDS isoform (76KDa) in the SWAP-RAP12x2 IDS. What is the relevance of this and is this significant?
- I don't seem to fully grasp relevance and rationale of using ApoB, as it is then relegated in the Supplementary and not discussed any further.
- The authors should remark more clearly that some SWAP designs can still bind IGFR1, as this is an important point.
- Fig 3D: IDS activity should potentially be normalized to VCN, as VCN values in Fig 3C are not so similar, especially for relevant groups as IDS.IGF2delco,IDS.SWAP-ApoE2co, IDS.SWAP-RAP12x2co. Moreover, IDS.IGF2del_co. variant has the lowest VCN, which likely explains its lower activity - should be clarified.
- Fig 3B: Where is the GFP group?
- Fig 5B: no clear superior reduction in heparan sulphate is observed from the graphs- please reformulate statement on this matter.
- In general, data quality is very good, however data presentation is not always at service of data interpretation. The figures are not cited in numerical order, making difficult to follow the results flow, as many elements in graphs/figures are discussed later in the text even if belonging in the same figure/graph. Additionally, jumping between figures disrupts understanding. Consider restructuring figures for clarity.
- Some graphs in supplementary figures are very little and it is difficult to see symbols/colours well (Fig S2). Moreover, using the same colour for upward and downward triangles is confusing, please consider changing shapes or colours to better differentiate conditions.

Referee #1 (Comments on Novelty/Model System for Author):

Although the initial data are compelling, some of the comparisons and claims made in the latter part of the paper about the superiority of SWAP tags over the existing IGF2 tag are overstated and need to be qualified and other sections to be clarified.

Referee #1 (Remarks for Author):

Catalano et al describe a novel engineered IGF2 peptide that has reduced off target binding to insulin receptor and IGF1 receptor, whilst maintaining its IGF2R binding capability. This has been a long-standing question mark in the lysosomal disease field for those employing enzyme replacement therapy with IGF2 tagged enzymes. IGF2 tags bind to the M6PR/IGFR protein (at distinct domains) meaning that exogenously delivered lysosomal enzymes that are typically taken up by M6PR binding, are more effectively bound to the same protein, facilitating increased uptake. The concern being that off target binding to insulin receptor and IGF1 receptor will block effective insulin or IGF1 trafficking and dangerously disrupt homeostasis. Catalano et al have removed the central loop of IGF2R binding domain (the so called SWAP domain), to insert other targeting sequences, and demonstrate in a genetic setting using hematopoietic stem cell gene therapy, that this reduces binding to Insulin receptor, and to IGF1 receptors, and can be used to insert other targeting peptides with novel receptor binding capabilities. They have further demonstrated the practical application of this in a genetic context using IGF2-SWAP tagged idursulfase enzyme to treat the brains of mice with Hunter disease (a childhood dementia) via hematopoietic stem cell gene therapy. Although the initial data are compelling, some of the comparisons and claims made in the latter part of the paper about the superiority of SWAP tags over the existing IGF2 tag are overstated and need to be qualified and other sections to be clarified. Importantly, as this paper shares three experimental groups with Catalano et al.(2023). *Mol Ther Methods Clin Dev*; more comparison back to that paper should be made in the discussion. This reduces enthusiasm for the efficacy of the SWAP tags in the Hunter disease model, compared to existing tags, but does not detract from the novel and interesting findings described.

Major

1) Page 4 introduction - mention is made that replacement of the central loop of IGF2 is made with AopE2, Raux12x2 and ApoB. Firstly, the described SWAP-ApoE2 sequence in Fig 1B is in fact the sequence for what is commonly known as ApoE peptide as it is only a single sequence of the receptor binding domain of apolipoproteinE. ApoEII refers to a tandem repeat of the ApoE receptor binding domain (Gleitz 2018 EMM) - thus the sequence used here: LRKLRKRLI should be referred to as ApoE and LRKLRKRLI-LRKLRKRLI referred to as ApoEII. The nomenclature ApoE2, used in this paper, should be avoided in any case to avoid confusion with isoforms of the ApoE gene (e.g. ApoE2 that describes the ϵ 2 allele of apolipoproteinE, conferring decreased risk to Alzheimer disease and increased longevity (Shinohara 2020 *elife*, Reimann 2020 *Nat comms*).

Unless the sequence in Fig 1 was in fact an error and should be a tandem repeat of LRKLRKRLI, we propose that the IDS-SWAPApoE2 construct is renamed throughout as IDS-SWAPApoE to reflect this and avoid further confusion.

We thank the reviewer for this remark. The ApoE-derived sequence used in the SWAP design refers to a single repeat of the sequence LRKLRKRLI, while the ApoE sequence used in IDS.ApoE2, as shown in Catalano et al 2023, is the same as the one used in Gleitz et al 2018, and refers to a tandem repeat of the sequence LRKLRKRLI. We have modified the manuscript and the Figures, referring to a single repeat as ApoE, and to a tandem repeat as ApoEII.

We would like to note that the nomenclature for ApoE-derived peptides remains inconsistent. One of the earliest direct fusion protein studies involving ApoEII referred to this peptide as “E1” tag (see Wang and colleagues Supplementary Table 1).¹ Croy and colleagues characterized several ApoE-derived peptides, referring to these as *apoE* followed by the number of the aminoacidic range in the native Apolipoprotein E (e.g. apoE(130-149), with aminoacid 1 referring to the first aminoacid after the signal peptide).² Hoe and colleagues referred to ApoEII as “ApoE dimer tandem repeat”³ and later as “apoE peptide₍₁₄₁₋₁₄₉₎”⁴. Dobson and colleagues referred to ApoEII as ApoEdp⁵, and Minami and colleagues described mono-, di-, and tri-repeats of LRKLRKRLI as “ApoE-monomer,” “ApoE-dimer,” and “ApoE-trimer,” respectively.⁶

As suggested by the reviewer and because our previously adopted nomenclature ApoE2 closely resembles the designation for the ApoE2 protein produced by the *APOE- ϵ 2* allele, we have now adopted the term ApoE or ApoEII to denote a single or a tandem repeat of the sequence LRKLRKRLI.

2) A second point is that no data are presented for ApoB in the main paper. We would suggest removal of this or full presentation of ApoB data (rather than the snippets in the supplementary Figure).

When designing the SWAP, we selected a long peptide such as the 39 aminoacids-long ApoB to test cargo capacity. However, we did not have the capacity to evaluate this variant *in vivo*

during the original experiments. In agreement with this reviewer and reviewer 3, we have removed the *in vitro* ApoB data.

3) A further problem with the SWAP ApoE2 tag being LRKLRKRL, inserted into the central loop of the IGF2 binding domain peptide, is that it is therefore not comparable to the "IDS.ApoE2" tag described in Catalano et al 2023 - which is in fact LRKLRKRL-LRKLRKRL. These are distinctly different domains and this distinction needs to be in place throughout the paper where comparisons are currently made between these products as if they are the same product either outside or within the SWAP context. This is not the case.

We thank the reviewer for this remark. While it is true that an isolated single-repeat peptide (LRKLRKRL) often shows reduced biological activity compared to the tandem-repeat ApoEII sequence – largely attributed to the inability of the short peptide to adopt the α -helical conformation^{5,6} – this limitation applies only to isolated peptides. When embedded within a larger amino-acid context, as in the native Apolipoprotein E sequence (or our SWAP-ApoE construct), a single LRKLRKRL repeat can adopt a helical structure and bind LRP-1 with high affinity.² Consistently, in Fig EV2C we show that both IDS.ApoEII and IDS.SWAP-ApoE bind Cluster IV of LRP-1 with very similar affinities, with the latter exhibiting only a slight decrease. Therefore, although one versus two repeats of isolated peptides may differ in helix stability and downstream biological effects, a single ApoE repeat in the SWAP design remains fully capable of LRP-1 binding.

We have now clarified this throughout the manuscript, and added it to the discussion section (lines 541-562).

4) I understand that this experiment leads on from a previous paper - but there is no description here of the lentiviral vector design used (Presumably the MND promoter driven LV used in reference 10) (Catalano et al.(2023). Mol Ther Methods Clin Dev, 101149.) The construction of the LV and details of the overlap of groups should be clearly stated within the introduction and detailed in methods.

As noted by the reviewer, the lentiviral backbone used here is identical to that in Catalano et al. (2023), and comprises an MND promoter within a pCCL backbone. We have now clarified this in the introduction (lines 90-91), results (lines 263-264), and methods sections (lines 661-662); information on the overlapping groups was already included in the Materials and Methods (lines 663-664).

5) On the same point I understand that the IDSco, IDS.IGF2co and WT groups are drawn from the same previous paper (Catalano et al.(2023). Mol Ther Methods Clin Dev, 101149). Why are comparisons made only to these groups, when the previous paper contained IDS.ApoE2co and IDS.RAP12x2co groups (without the SWAP tags). Surely this would have been a better comparison?

We thank the reviewer for the suggestion. The goal of this work was to modulate the receptor-binding profile of the IGF2 tag, which is why we originally compared the SWAP tag to both the unmodified IGF2 tag and the central loop-deleted IGF2 tag as controls. However, we agree that directly comparing the SWAP constructs to their corresponding non-SWAP

versions (i.e., IDS.ApoEII and IDS.RAP12x2), mainly on their efficacy and plasma levels, would provide additional insight for the reader. We have now extensively compared the SWAP tag with the respective single tags in the discussion section (lines 518-600).

6) Looking at Fig 3G of the current paper, its interesting that the functional IDS assay in plasma of transplanted mice vs VCN doesn't match the relative activity seen in HMC3 cells *in vitro* (Fig 1E). In Fig 3G, the addition of any SWAP tag to coIDS appears to decrease the relationship between plasma IDS activity compared to VCN in bone marrow. I notice also that there is a consistent drop in the IDS activity in WBC vs bone marrow which is unusual as they should share similar VCNs. Notably if you look at the same Figure(2F) in (Catalano et al.(2023). *Mol Ther Methods Clin Dev*, 101149), it is clear that the nonSWAP tagged versions of IDS.ApoE2co (which appears to be a tandem repeat in this case), and the IDS.RAP12x2co perform similarly to IDSco in the IDS activity vs VCN per genome in bone marrow graph (Fig 2F in the former paper). This implies that the SWAP motif of IGF2 actually decreases secreted IDS activity overall compared to when the tags are used without the SWAP. This should be noted and discussed.

We appreciate the reviewer's observation that IGF2-tagged IDS exhibits reduced steady-state plasma levels – a phenomenon we previously hypothesized to be attributed to the high-affinity binding between the IGF2 tag and the CI-M6P/IGF2R.⁷ This hypothesis is supported by an *in vitro* transfection experiment in HEK 293T cells demonstrating that IDS.IGF2 secretion is reduced relative to untagged IDS, but it is fully rescued by co-incubation with excess free IGF2 peptide in the culture medium (Catalano et al. 2023, Fig 1F, Fig S1E).⁷

However, decreased secretion of IDS.IGF2 was not observed after transduction of HMC3 (this work, Fig 1D). At the moment, we believe that this is caused by lower levels of CI-M6P/IGF2R expression in HMC3 compared to HEK 293T cells. Additional experiments are needed to test this hypothesis, which is supported by the observation that the HMC3 cell line exhibited the lowest uptake levels among those cell line tested (Fig 2E–H, note the differences in the Y axes).

We also agree with the reviewer that the reduced IDS activity in WBCs compared to BM remains unexplained. We observed a trend toward similar IDS activity patterns in plasma and WBCs, which could be caused by interference of the tag with protein expression and, therefore, secretion. Whether this pattern is causal must be determined in future experiments.

We thank once again the reviewer for prompting these discussion points, which we have now added to the discussion section (lines 563-600). The relevant text is reported below:

“[...] In general, fusion to any moiety with strong, widespread receptor binding properties can alter the biodistribution of the resulting fusion protein through receptor-mediated clearance pathways. However, decreased plasma IDS activity levels were not observed after gene therapy with IDS.RAP12x2 and IDS.ApoEII. In the case of IDS.RAP12x2¹, the RAP12x2 moiety was shown to not be functional (no effect *in vivo* and unable to bind LRP-1 in an ELISA assay) and likely to behave like untagged IDS, explaining the unchanged plasma IDS activity levels. On the other hand, IDS.ApoEII was shown to be fully functional *in vivo*^{1,2} and

to bind with high affinity to LRP-1 *in vitro*.¹ Despite this, plasma IDS activity levels were unchanged in the IDS.ApoEII group compared to untagged IDS. This finding is per-se surprising giving the widespread distribution of LDL-receptor family members, including LRP-1. One possibility is that IDS.ApoEII, like native Apolipoprotein E, after delivering lipids to target cells, is routed into recycling endosomes and then re-secreted, thereby maintaining the circulating ApoE pool.³⁻⁸ Consistent with this, we previously showed that ApoEII tagging of IDS only marginally increases cellular uptake in MPS II fibroblasts and bEND.3 cells over 24 hours¹, and, in this work, the presence of an ApoE moiety only marginally increased the uptake levels of IDS.SWAP-ApoE compared to untagged IDS. Importantly, this pathway does not apply to IGF2 and RAP, which, upon binding to CI-M6P/IGF2R (IGF2) and LDL receptor family members (RAP), are routed to late endosomes and lysosomes for degradation.⁹⁻¹¹ Looking at plasma IDS activity levels (Fig 3G), our SWAP constructs, which combine IGF2 with either ApoE or RAP12x2, seem to combine characteristics of each moiety. The IGF2 tag – present in IDS.IGF2, IDS.SWAP-ApoE, IDS.SWAP-RAP12x2, and IDS.IGF2del – is scavenged by the CI-M6P/IGF2R with higher affinity compared to untagged IDS, thereby decreasing plasma IDS activity levels compared to untagged IDS. Incorporation of an ApoE-derived peptide in IDS.SWAP-ApoE does not further alter plasma IDS levels compared to those of IDS.IGF2, paralleling the behaviour of IDS.ApoEII versus untagged IDS and consistent with the proposed re-secretion mechanism.³⁻⁸ In contrast, the presence of a functional RAP12x2 moiety expands the plasma scavenging pathways for IDS.SWAP-RAP12x2, thereby decreasing further the plasma IDS activity levels compared to IDS.IGF2. This might suggest that modulating the contributions of each of the moieties in the SWAP tag, by mutagenesis or modulation of the plasma half-life, might be required in future work to reach a complete additive effect of each moiety. Alternatively, another hypothesis to explain reduced plasma IDS activity levels for all the IGF2-based versions of IDS is that the tagging strategy employed affects IDS protein expression, as we noticed reduced expression levels in WBCs – which might represent the main source of therapeutic protein in plasma – that parallel the plasma IDS activity pattern (Fig 3E, F). Further studies will be needed to clarify the underlying causes of reduced secretion of IGF2-based versions of IDS.”

7) It should also be noted that all SWAP and IGF2 tags used in this paper give reduced IDS activity per VCN of BM than the native IDSc0 (Fig 3G), (unlike the IDS.ApoE2 or IDS.Rap12x2). This is a significant downside to the SWAP tags and is not stated in results, nor in discussion.

We have elaborated on this point in the discussion (see answer to question 6 and lines 563-600), while in the result section we expanded our previous observation of reduced IDS plasma activity relative to untagged IDS (lines 288-290).

8) On page 10 a statement is made "IDS.SWAPRAP12x2 can effectively engage LRP-1, in contrast to IDS.RAP12x2, which, as we previously showed¹⁰, failed to bind LRP-1 in a similar ELISA assay." (Fig EV2D). Although the supplementary figure compares several different peptides, including the Apoe2 tag alone, RAP12x2 is the only one missing (referenced out to the previous Catalano paper). Due to the importance of this claim, as this would imply increased potential binding to and crossing of LRP receptors on the blood brain barrier of

RAP12x2, this assay needs to be repeated including RAP12x2 alongside all the other SWAP peptides, Apoe2, IGF2del, and IGF2 tagged IDS.

We thank the reviewer for this remark. We have now repeated the assay, and directly compared IDS.RAP12x2 with IDS.SWAP-RAP12x2, including untagged IDS as negative control, and IDS.ApoEII as positive control (Fig EV2D). This confirmed our previous observation that IDS.RAP12x2 failed to bind LRP-1, while RAP12x2 in a SWAP conformation efficiently bound this receptor. All the other versions were tested in previous ELISAs (as shown in Fig EV2D and EV2C), which have been repeated multiple times.

9) A statement is made on page 11 that "gene therapy with SWAP vectors effectively prevents peripheral pathology and significantly improves bone abnormalities in *Idsy*^{-/-} mice, with IDS.IGF2co and IDS.SWAP-RAP12x2co achieving near-complete normalization of key parameters." Although the cardiac data are convincing in this regard and the statement could be modified to refer only to the alcian blue data, the bone data presented are not.

Notably a significant difference between WT and IDS null is only presented for Talus Tb.TV and Talus Ct.Ar/Tt.Ar (Fig 4 E and F). No significant differences between IDS and any treatment group are shown for talus measurements, therefore the best that can be said is that there is a trend for improvement. This statement could say that where differences in bone pathology were shown there was a trend for improvement, It should not say that peripheral pathology is prevented (except for the specific organs shown -liver, spleen, aortic valves) as this is not shown here.

We agree with the reviewer's suggestion. We have adjusted the text accordingly (lines 374-379).

10) Supplementary Figures 4 and 5 add little to the paper and should probably be removed.

Supplementary Fig 4 extends our peripheral pathology analysis by presenting LAMP1 stainings and μ CT analysis of bone microarchitecture of the medial cuneiform. Although the latter did not show statistical significance, it provides insight into which skeletal elements are, or not, affected in MPS II mice. This Fig also includes schematic illustrations of the microarchitecture parameters assessed, which should aid researchers less familiar with μ CT analysis. Supplementary Fig 5 shows orthogonal projections of the segmented bones to show the *bona fide* segmentation for the μ CT analysis. While Fig 5 might be disposable, we believe Supplementary Fig 4 is essential as it provides additional data and aids data interpretation. We have therefore removed Supplementary Fig 5.

11) Fig 5E The data are fairly convincing, but DAPI staining is quite inconsistent across the Fig ure. Were all the sections stained at the same time? This is critical for determining LAMP1 contribution via thresholding.

We performed immunofluorescent stainings in parallel for all experimental conditions. Following staining, brains were imaged on a Leica STELLARIS 5 confocal laser scanning microscope (Leica Microsystems) using identical acquisition parameters across all samples. Whole-brain images were first acquired as a tile scan, and regions of interest were then

imaged in three dimensions by acquiring Z-stacks spanning the full thickness of the tissue, beginning and ending at the first and last focal planes where the marker signal (LAMP1 or GFAP) reached background threshold levels. Z-stacks were then processed in ImageJ using the *Z-Project* function with max projection for each channel prior to quantification.

Although we acknowledge that DAPI staining intensity was elevated in certain sections of cortex, hippocampus and cerebellum of IDS.SWAP-RAP12x2 and *ids*^{y/-} groups, this did not affect the analysis, as demonstrated by the minimal variation within these treatment groups in the specified brain regions.

12) A conjecture on why CD68 in the thalamus (Fig 6) was poorly corrected by all the vectors would be interesting. This didn't seem to be true for GFAP staining but there was residual LAMP1 in the thalamus. I wonder if a marker of activated microglia might have been more informative?

We thank the reviewer for raising this point. CD68 is a pan-macrophage marker and, in the brain, stains for activated microglia, as previously shown.¹⁸

In general, in brain, CD68 labels both microglia and any infiltrating macrophages – which are though present at very low levels – marking cells with enhanced phagocytic activity.¹⁸ CD68 in brain effectively stains activated (general markers CD68, IBA1, CD74, MHCII, ferritin)¹⁸ and amoeboid (general markers IBA1, CD74, CD68, MHCII)¹⁸ microglia. Therefore, in this context CD68 can detect the presence of microgliosis. Future studies should be focused on the analysis of additional markers to characterize other subpopulation of microglia cells, such as resting microglia cells (markers IBA1, P2RY12, TMEM119, CD74).¹⁸

To discuss further on the thalamus correction:

In our hands, the thalamus was indeed the least effectively targeted region by all of our SWAP-tagged constructs. This points at lower levels of therapeutic IDS protein reaching this brain region. SWAP and IGF2 variants effectively engage the IGF2R, IGF1R (IDS.IGF2 and IDS.SWAP-ApoE), the IR-A (IDS.IGF2) and the LRP-1 (IDS.SWAP-ApoE and IDS.SWAP-RAP12x2). All these receptors might be involved in the delivery of therapeutic IDS to different brain regions, including the thalamus.¹⁹ For example, after intravenous infusion of radiolabelled IGF2 peptide, accumulation of IGF2 was observed – although at a lesser extent compared to other brain areas – in capillaries and parenchymal cells of the supraoptic nucleus and anterior thalamic nucleus.²⁰ This shows that the IGF2 tag can reach at least some thalamic areas. The transport of IGFs to the brain occurs through the blood–CSF barrier, via interactions between IGF receptors and LRP2. From the CSF, IGFs then diffuse rapidly into periventricular regions such as the hippocampus. Delivery to more deep brain regions likely requires additional transport mechanisms that remain incompletely understood.¹⁹ A plausible route is transcytosis across the blood–brain barrier, as it was shown that proteolysis of circulating IGF-binding proteins releases free IGFs, which then engage IGF receptors, which, in turn, interact with LRP-1 on the endothelial surface to facilitate transport through the blood-brain barrier.¹⁹ Given that IGF¹⁹ receptors and LRP-1 are broadly expressed throughout the brain, one would therefore expect IDS.IGF2 and SWAP-tagged IDS to achieve similarly widespread distribution, including thalamus.

Therefore, one possible explanation is that Hunter syndrome itself induces a localized state of IGFs-resistance in the thalamus. It was indeed previously shown that pro-inflammatory cytokines, oxidative stress, glutamate excitotoxicity, or amyloid aggregates can all impair IGF receptor signalling in various neurodegenerative contexts²¹, potentially lowering uptake of IDS.IGF2-based constructs in affected areas.

As the reviewer noted, the pattern of poor thalamic correction was observed in CD68-stained sections (microgliosis) but not in GFAP-stained sections (astrogliosis). One possibility is that the therapeutic threshold needed to resolve astrogliosis is lower than that required for microgliosis, or that activated microglia are themselves resistant to correction with IGF2-based constructs, and that increase transport to the brain – or increased plasma steady-state concentration resulting in increased brain delivery – might be required to reach therapeutic efficacy. Therefore, the precise mechanisms behind persistent pathology in thalamus remain to be defined.

13) The statement in the last paragraph of results is overclaiming. (P20). There is very little (no?) statistical evidence from Figures 5-6 that there are differences between SWAP variants and IDS.IGF2co and only some limited data to show that these are better than IDS.IGF2del_co or IDSco. It should be rephrased to "These results indicate that in brain, gene therapy with IDS.SWAP variants and IDS.IGF2 had similar efficacy to each other, all slightly better than IDS.IGF2del_co but with superior efficacy to IDSco, demonstrable in some specific areas such as brain HS reduction and brainstem astrogliosis as well as general improvements in LAMP1 clearance in the brain.

We thank the reviewer for the comment. We have now adjusted the last paragraph of the results (lines 471-475).

14) Similar claims in the discussion claiming superiority of SWAP tags to IDS.IGF2 and also in the abstract need to be toned down. E.g. Page 22 "SWAP tags...demonstrated a therapeutic advantage in correcting or preventing most peripheral or CNS manifestations compared to untagged IDS and IDS.IGFdel" - this statement is inaccurate.

We thank the reviewer for the comment. We have now also adjusted the abstract, introduction (lines 89-96) and discussion (starting from line 529).

15) Little comparison to the previous tags used in Catalano 2023 Mol ther clin meth is made. Given that these mice share 3 of the same groups with that paper, a comparison and conjecture should be made with the previous tags described.

We have now compared the SWAP variants with the previously tested single tags in the discussion (lines 518-600).

Minor

Page 8 - there should be a reference to Fig 1E at the end of this sentence I think "which confirmed that SWAP tagging does not affect specific activity of IDS"

This is now adjusted.

Fig EV1C has an almost illegible label - please enlarge.
This is now adjusted.

In Fig 2 E-H there seems to be some discrepancy between uptake into Bend3 and MPSII fibroblasts between the assay performed here for IDSco and IDS.IGF2co and the same products compared in the same assay in the Catalano 2023 paper (Fig 2G and H.). In the previous paper there was apparently a much greater difference between the two groups. Can this be discussed please.

We thank the reviewer for this observation. Both experiments were conducted using the same IDS concentration range (as determined by sandwich ELISA) and assessed uptake in MPS II fibroblasts and bEND.3 cells during 24 hours. The only difference, detailed in the Materials and Methods of both studies, was the source of conditioned medium. We found that receptor ELISAs (particularly for IGF2R and IGF1R) produced cleaner signals when using input from HMC3 cells versus HEK 293T cells. Consequently, all results presented here were generated with HMC3-derived conditioned medium, while in Catalano et al 2023, uptake was performed using conditioned medium from HEK 293T cells. We cannot rule out that this change in medium source may have influenced the observed differences in IDS.IGF2 uptake between the two assays.

Figures 3D and 3E and 3F show IDS activity in bone marrow and WBC and plasma respectively. As WT IDS levels are much lower than that of lentiviral vector transduced groups the current graph axis does not allow distinction between the levels of GFP transduced, WT and IDS null mice. Either the Y axis should be split or a log axis presented so the WT IDS activity levels can be seen and compared to LV transduced groups.

Besides these main Figures, we also presented the same data but expressed as fold-increase over WT in Fig EV3A-C. Here, readers can see how the IDS activity levels compare with the other groups, including those with very low IDS activity (*ids*^{Y/-}, WT and GFP-treated).

Fig 5D shows chimerism levels in the brain of around 1%. Please clarify if this refers to total chimerism as a percentage of total brain cells or microglial chimerism.

This refers to total chimerism, and it is expressed a percentage of CD45.1 genomes in total CD45.2 genomes. We have clarified this in the result section and the Figure.

Fig 6 should be split into two. It will be illegible in a paper as it is.
We have adjusted this according to the reviewer's suggestion.

Fig EV7 adds little to the paper and should probably be removed.
We have removed Fig EV7 according to the reviewer's suggestion.

Referee #3 (Remarks for Author):

Pijnappel and colleagues reported novel IGF2-based tag designs, namely SWAP, designed to enhance the efficacy and safety of IGF2 tag-based gene therapy for lysosomal storage

disorders, with a specific focus on MPSII. They demonstrate the critical role of IGF2's central loop in binding the insulin and IGF1 receptors, while demonstrating its dispensability for interaction with mannose 6-phosphate /IGF2 receptor, which is a crucial target for enzyme delivery to the lysosomes. The SWAP tags replace the central loop with alternative epitopes to preserve structural integrity and enable high-affinity multimodal receptor targeting. The authors affirm that data in MPSII mice show the superiority of IDS fused to SWAP variants (compared with traditional IGF2 tags) in correcting disease pathology in liver, spleen, heart, bone, and brain. This is an important study in the field, addressing a relevant issue for LSD patients. The authors provided solid data to establish SWAP as an innovative, safer and brain penetrating tag for ex-vivo lentiviral gene therapy for MPSII. However, some results are overstated in the abstract and some points need to be better and clearly addressed/discussed.

- One major flaw of the study is IDS.IGF2co not affecting glucose levels in vivo. This is a bit underwhelming, as the main study rationale for the SWAP design is to have improved tags which still allow for good tissue uptake but do not alter glucose homeostasis. This is a crucial point, which would require some toning down. What is the relevance of these newly developed SWAPs' tags over the IDS.IGF2 (and IDS.IGF2_del), if the latter does not cause glucose levels' fluctuation and has better binding affinity/uptake? Maybe the authors could highlight more SWAPs' enhanced tissue-specific targeting or clarify whether this might be more relevant for ERT more than for HSC-GT.

The disturbance of glucose regulation risks associated with IGF2-based therapeutics is well documented in humans²², and this alone highlights the need for alternative strategies. In our HSPC-LVGT model of Hunter syndrome, systemic IDS.IGF2 expression did not alter glucose homeostasis in a glucose tolerance test. As noted in the discussion, plasma IGF2 concentrations following HSPC-LVGT are substantially lower than, for example, those achieved by bolus administration of IGF2.GAA, the GAA analog that caused hypoglycemic events shortly after drug infusion in late onset Pompe disease patients. Despite this, IGF2-related toxicity could still manifest in gene therapy settings due to the constant and long-term exposure to supraphysiological levels of IGF2 moieties – without possibilities for re-dosing – compared to single administration during bolus ERT. For example, E20 mouse embryos overexpressing IGF2 at levels ~65% above wild-type developed increased body weight and pancreatic islet hyperplasia²³, and 5–10-week-old mice with systemic IGF2 overexpression exhibit both weight gain and impaired glycolipid metabolism.²⁴ Another scenario is that localized transgene-over-expression mediated by, for example, macrophages engrafted adjacent to pancreatic islets, could expose β -cells to high local IGF2 concentrations. In this context, Devedjian et al. showed that β -cell-specific IGF2 overexpression induces hyperinsulinemia, hyperglycaemia, and impaired glucose tolerance despite only ~2-fold increases in circulating IGF2 compared to WT²⁵, and at levels comparable to those observed for our IDS.IGF2 construct during gene therapy. Additionally, species differences in glucose regulation between mice and humans warrant for caution when translating results from mice to humans, especially in a non-re-dosing gene therapy setting.²⁶ Finally, as the reviewer suggests, SWAP tags may also benefit other delivery modalities, such as intravenous ERT, where peak plasma levels strongly exceed those achieved by HSPC-LVGT.

We have incorporated/revised these points in the discussion section (lines 601-635). We thank the reviewer for this discussion point.

- Are SWAPs showing lower binding ability and uptake over IDS.IGF2 and IDS.IGF2_del? Please elaborate on this.

We thank the reviewer for this observation. The SWAP variant with an ApoE insert, but not the RAP12x2 insert, exhibited consistently lower uptake than IDS.IGF2 and IDS.IGF2del across all cell lines tested. This is surprising, since both SWAP-ApoE and SWAP-RAP12x2 are designed to engage additional receptors to those normally engaged by IGF2 given saturating conditions, thereby expanding the pathways for receptor-mediated uptake.

We have speculated about this in the discussion section (lines 575-600), and the relevant text is reported below:

“[...]IDS.ApoEII, like native Apolipoprotein E, after delivering lipids to target cells, is routed into recycling endosomes and then re-secreted, thereby maintaining the circulating ApoE pool.³⁻⁸ Consistent with this, we previously showed that ApoEII tagging of IDS only marginally increases cellular uptake in MPS II fibroblasts and bEND.3 cells over 24 hours¹, and, in this work, the presence of an ApoE moiety only marginally increased the uptake levels of IDS.SWAP-ApoE compared to untagged IDS. Importantly, this pathway does not apply to IGF2 and RAP, which, upon binding to CI-M6P/IGF2R (IGF2) and LDL receptor family members (RAP), are routed to late endosomes and lysosomes for degradation.⁹⁻¹¹ Looking at plasma IDS activity levels (Fig 3G), our SWAP constructs, which combine IGF2 with either ApoE or RAP12x2, seem to combine characteristics of each moiety. The IGF2 tag – present in IDS.IGF2, IDS.SWAP-ApoE, IDS.SWAP-RAP12x2, and IDS.IGF2del – is scavenged by the CI-M6P/IGF2R with higher affinity compared to untagged IDS, thereby decreasing plasma IDS activity levels compared to untagged IDS. Incorporation of an ApoE-derived peptide in IDS.SWAP-ApoE does not further alter plasma IDS levels compared to those of IDS.IGF2, paralleling the behaviour of IDS.ApoEII versus untagged IDS and consistent with the proposed re-secretion mechanism.³⁻⁸ In contrast, the presence of a functional RAP12x2 moiety expands the plasma scavenging pathways for IDS.SWAP-RAP12x2, thereby decreasing further the plasma IDS activity levels compared to IDS.IGF2. This might suggest that modulating the contributions of each of the moieties in the SWAP tag, by mutagenesis or modulation of the plasma half-life, might be required in future work to reach a complete additive effect of each moiety. Alternatively, another hypothesis to explain reduced plasma IDS activity levels for all the IGF2-based versions of IDS is that the tagging strategy employed affects IDS protein expression, as we noticed reduced expression levels in WBCs – which might represent the main source of therapeutic protein in plasma – that parallel the plasma IDS activity pattern (Fig 3E, F). Further studies will be needed to clarify the underlying causes of reduced secretion of IGF2-based versions of IDS.”

Additionally, we speculate that, in order to engage concomitantly multiple receptors, reaching saturating conditions is necessary. Thereby, the input concentration range tested might not be sufficient to saturate the available receptors. These observations warrant further experiments to test these hypotheses.

- What is the rationale for testing uptake in fibroblasts? Similarly, why choosing a microglia cell line for IDS processing and IDS activity experiments in Fig 1? Please explain rationale for these choices

We thank the reviewer for the interesting question. MPS II fibroblasts are the only primary cell source available in our center from MPS II patients. Therefore, we considered this an important cell model to test uptake. Additionally, because microglia-like cells constitute one of the primary sources of therapeutic transgene expression during HSPC-LVGT – following the rationale that, during HSPC-LVGT, circulate monocyte engraft the brain and differentiate into microglia like cell upon preconditioning procedures²⁷ – we chose the HMC3 microglia cell line to assess expression.

- Fig EV1A: Why wasn't IDS.IGF2 tested as well for a direct comparison of efficiency?

We thank the reviewer for this question. Although we did initially include IDS.IGF2 construct in the analysis in Fig EV1A, we decided to omit its presentation at this point of the story in order to maintain focus on the characterization of IDS.IGF2del. The complete graph is shown in the figure below, which now replaces Fig EV1A in the revised version. In this direct ELISA, we observed a slightly reduced binding affinity of IDS.IGF2del versus IDS.IGF2, consistent with the results in same range of concentrations obtained in a competitive ELISA (Fig 2A).

- Fig 1D: The level of transduction achieved with each vector is unknown, making interpretation difficult - VCN should have been calculated and activity in cells should be correlated with VCN.

We have now added VCN data to the graph in Fig 1D. VCN values varied across the conditions, and the variability observed in the VCN explained some of the variability observed in the intracellular activity. This was not the case for the medium. For this reason, we have performed a new experiment and analyzed secreted activity per VCN at low (MOI 5) and high (MOI 50) MOI, and the resulting graph is shown in Fig EV1G. Secretion was similar across the conditions, although IDS.SWAP-ApoE showed a slightly reduced secretion at low MOI (lines 161-166). We hope these additions aid the interpretation of these data.

- In Fig 1C I could see a skewing toward the precursor IDS isoform (76KDa) in the SWAP-RAP12x2 IDS. What is the relevance of this and is this significant?

We thank the reviewer for this comment. To investigate this, we have now performed a two-way ANOVA which revealed a modest but significant skew toward the precursor IDS isoform for all constructs tested, including IDS.IGF2 and IDS.IGF2del. We have added this to the results section. This might suggest that the current tagging strategy (using linker L(GGGGS)x4) for IGF2, IGF2del and SWAP tags may be suboptimal for proper IDS maturation. We have updated the corresponding Fig to include statistical analysis, and revised the results section (lines 153-157).

- I don't seem to fully grasp relevance and rationale of using ApoB, as it is then relegated in the Supplementary and not discussed any further.

When designing the SWAP, we selected a long peptide such as the 39 aminoacids-long ApoB to test cargo capacity. However, we did not have the capacity to evaluate this variant *in vivo* during the original experiments. As suggested by reviewer 1, we have removed the *in vitro* ApoB data.

- The authors should remark more clearly that some SWAP designs can still bind IGFR1, as this is an important point.

We agree with the reviewer that this is an important point. We have now clearly stated this in the abstract (lines 34-35), introduction (lines 88-89), results (lines 206-220, 256-257) and discussion sections (lines 486-517).

- Fig 3D: IDS activity should potentially be normalized to VCN, as VCN values in Fig 3C are not so similar, especially for relevant groups as IDS.IGF2delco, IDS.SWAP-ApoE2co, IDS.SWAP-RAP12x2co. Moreover, IDS.IGF2del_co. variant has the lowest VCN, which likely explains its lower activity - should be clarified.

We thank the reviewer for this observation. As we have previously demonstrated that the relationship between IDS enzyme activity in bone marrow and VCN in bone marrow is non-

linear with saturation levels reached at VCN 2 (Fig 2D of Catalano et al, 2023)⁷, enzyme activity in bone marrow does not rise in direct proportion to VCN once saturation is reached. For this reason, we have used the previously determined relationship to normalize the activity data in bone marrow per VCN in bone marrow (model: $Activity = \frac{5203 * VCN}{0.6086 + VCN}$; normalization formula: $Normalized\ Activity = \frac{measured\ activity * (average\ VCN)}{(0.6086 + average\ VCN) * ((0.6086 + measured\ VCN) / measured\ VCN)}$). The resulting graph is now included in supplementary Fig 3 (Fig EV3D), and it is displayed

here, while description of the normalization method is detailed in materials and methods (lines 741-747).

As the reviewer correctly noted, the graph revealed group-specific differences in activity per VCN, with IDS.IGF2del and IDS.SWAP-RAP12x2 slightly outperforming IDS, and IDS.IGF2 and IDS.SWAP-ApoE2 slightly underperforming IDS, although without reaching statistical significance. Nevertheless, plasma IDS activity – fundamental for transcytosis-based mechanisms of brain correction⁷ – was higher in the IDS.IGF2del group compared to the SWAP groups and the IDS.IGF2 group, while brain IDS activity, chimerism and VCN – which are driven largely by engrafted microglia-like cells, the other mechanism of correction during HSPC-LVGT – were comparable across all groups.

We have highlighted these differences in the result section (lines 279-286).

- **Fig 3B: Where is the GFP group?**

Although GFP-treated recipient (CD45.2) were also transplanted with CD45.1-derived HSPCs, chimerism analysis by CD45.1 staining was technically unfeasible as the anti-CD45.1 antibody was FITC-conjugated, which overlaps with the emission spectrum of the GFP expressed by HSPC of this group. Therefore, we omitted CD45.1 staining during flow cytometry analysis of this group. Nevertheless, we have measured chimerism in brain for this group using a qPCR-based assay that we have previously established, which resulted to be comparable to the other groups (Fig 5D). This suggests that overall chimerism for this group was comparable to the other groups.

- **Fig 5B: no clear superior reduction in heparan sulphate is observed from the graphs-please reformulate statement on this matter.**

We agree with the reviewer. We have now revised the relative statements (lines 471-475).

- In general, data quality is very good, however data presentation is not always at service of data interpretation. The Figures are not cited in numerical order, making difficult to follow the results flow, as many elements in graphs/Figures are discussed later in the text even if belonging in the same Figure/graph. Additionally, jumping between Figures disrupts understanding. Consider restructuring Figures for clarity.

We thank the reviewer for this suggestion. We agree that in the first 2 paragraphs of the results section we referred to experiments described later in the text. Therefore, we have now adjusted these sections and we hope these changes improve the results flow.

- Some graphs in supplementary Figures are very little and it is difficult to see symbols/colours well (Fig EV2). Moreover, using the same colour for upward and downward triangles is confusing, please consider changing shapes or colours to better differentiate conditions.

We have now adjusted this throughout the paper.

References

1. Wang, D., El-Amouri, S.S., Dai, M., Kuan, C.-Y.Y., Hui, D.Y., Brady, R.O., and Pan, D. (2013). Engineering a lysosomal enzyme with a derivative of receptor-binding domain

- of apoE enables delivery across the blood-brain barrier. *Proc Natl Acad Sci U S A* *110*, 2999–3004. <https://doi.org/10.1073/pnas.1222742110>.
2. Croy, J.E., Brandon, T., and Komives, E.A. (2004). Two apolipoprotein E mimetic peptides, apoE(130-149) and apoE(141-155) 2, bind to LRP1. *Biochemistry* *43*, 7328–7335. <https://doi.org/10.1021/bi036208p>.
 3. Hoe, H.S., and Rebeck, G.W. (2005). Regulation of ApoE receptor proteolysis by ligand binding. *Molecular Brain Research* *137*, 31–39. <https://doi.org/10.1016/J.MOLBRAINRES.2005.02.013>.
 4. Hoe, H.S., Pocivavsek, A., Dai, H., Chakraborty, G., Harris, D.C., and Rebeck, G.W. (2006). Effects of apoE on neuronal signaling and APP processing in rodent brain. *Brain Res* *1112*, 70–79. <https://doi.org/10.1016/j.brainres.2006.07.035>.
 5. Dobson, C.B., Sales, S.D., Hoggard, P., Wozniak, M.A., and Crutcher, K.A. (2006). The Receptor-Binding Region of Human Apolipoprotein E Has Direct Anti-Infective Activity. *J Infect Dis* *193*, 442–450. <https://doi.org/10.1086/499280>.
 6. Minami, S.S., Cordova, A., Cirrito, J.R., Tesoriero, J.A., Babus, L.W., Davis, G.C., Dakshanamurthy, S., Turner, R.S., Pak, D.T., Rebeck, G.W., et al. (2010). ApoE mimetic peptide decreases A production in vitro and in vivo. *Mol Neurodegener* *5*, 16. <https://doi.org/10.1186/1750-1326-5-16>.
 7. Catalano, F., Vlaar, E.C., Katsavelis, D., Dammou, Z., Huizer, T.F., van den Bosch, J.C., Hoogeveen-Westerveld, M., van den Hout, H.J.M.P., Oussoren, E., Ruijter, G.J.G., et al. (2023). Tagged IDS causes efficient and engraftment-independent prevention of brain pathology during lentiviral gene therapy for Mucopolysaccharidosis type II. *Mol Ther Methods Clin Dev*, 101149. <https://doi.org/10.1016/J.OMTM.2023.101149>.
 8. Gleitz, H.F., Liao, A.Y., Cook, J.R., Rowston, S.F., Forte, G.M., D'Souza, Z., O'Leary, C., Holley, R.J., and Bigger, B.W. (2018). Brain-targeted stem cell gene therapy corrects mucopolysaccharidosis type II via multiple mechanisms. *EMBO Mol Med* *10*, 1–19. <https://doi.org/10.15252/emmm.201708730>.
 9. Heeren, J., Grewal, T., Laatsch, A., Rottke, D., Rinninger, F., Enrich, C., and Beisiegel, U. (2003). Recycling of apoprotein E is associated with cholesterol efflux and high density lipoprotein internalization. *Journal of Biological Chemistry* *278*, 14370–14378. <https://doi.org/10.1074/jbc.M209006200>.
 10. Fazio, S., Linton, M.F., Hasty, A.H., and Swift, L.L. (1999). Recycling of apolipoprotein E in mouse liver. *Journal of Biological Chemistry* *274*, 8247–8253. <https://doi.org/10.1074/jbc.274.12.8247>.
 11. Heeren, J., Beisiegel, U., and Grewal, T. (2006). Apolipoprotein E recycling: Implications for dyslipidemia and atherosclerosis. *Arterioscler Thromb Vasc Biol* *26*, 442–448. <https://doi.org/10.1161/01.ATV.0000201282.64751.47>.
 12. Heeren, J., Weber, W., and Beisiegel, U. (1999). Intracellular processing of endocytosed triglyceride-rich lipoproteins comprises both recycling and degradation. *J Cell Sci* *112*, 349–359. <https://doi.org/10.1242/JCS.112.3.349>.
 13. Hasty, A.H., Plummer, M.R., Weisgraber, K.H., Linton, M.F., Fazio, S., and Swift, L.L. (2005). The recycling of apolipoprotein E in macrophages: influence of HDL and apolipoprotein A-I. *J Lipid Res* *46*, 1433–1439. <https://doi.org/10.1194/JLR.M400418-JLR200>.
 14. Braun, N.A., Mohler, P.J., Weisgraber, K.H., Hasty, A.H., Linton, M.F., Yancey, P.G., Yan, R.S., Fazio, S., and Swift, L.L. (2006). Intracellular trafficking of recycling apolipoprotein

- E in Chinese hamster ovary cells. *J Lipid Res* 47, 1176–1186.
<https://doi.org/10.1194/jlr.M500503-JLR200>.
15. Laatsch, A., Panteli, M., Sornsakrin, M., Hoffzimmer, B., Grewal, T., and Heeren, J. (2012). Low Density Lipoprotein Receptor-Related Protein 1 Dependent Endosomal Trapping and Recycling of Apolipoprotein E. *PLoS One* 7, e29385.
<https://doi.org/10.1371/JOURNAL.PONE.0029385>.
 16. Czekay, R.P., Orlando, R.A., Woodward, L., Lundstrom, M., and Farquhar, M.G. (1997). Endocytic trafficking of megalin/RAP complexes: Dissociation of the complexes in late endosomes. *Mol Biol Cell* 8, 517–532. <https://doi.org/10.1091/MBC.8.3.517>,
 17. Willnow, T.E., Orth, K., and Herz, J. (1994). Molecular dissection of ligand binding sites on the low density lipoprotein receptor-related protein. *Journal of Biological Chemistry* 269, 15827–15832. [https://doi.org/10.1016/S0021-9258\(17\)40755-1](https://doi.org/10.1016/S0021-9258(17)40755-1).
 18. Lier, J., Streit, W.J., and Bechmann, I. (2021). Beyond Activation: Characterizing Microglial Functional Phenotypes. *Cells* 2021, Vol. 10, Page 2236 10, 2236.
<https://doi.org/10.3390/CELLS10092236>.
 19. Fernandez, A.M., and Torres-Alemán, I. (2012). The many faces of insulin-like peptide signalling in the brain. *Nat Rev Neurosci* 13, 225–239.
<https://doi.org/10.1038/NRN3209>.
 20. Reinhardt, R.R., and Bondy, C.A. (1994). Insulin-like growth factors cross the blood-brain barrier. *Endocrinology* 135, 1753–1761.
<https://doi.org/10.1210/ENDO.135.5.7525251>.
 21. Carro, E., and Torres-Aleman, I. (2004). The role of insulin and insulin-like growth factor I in the molecular and cellular mechanisms underlying the pathology of Alzheimer's disease. *Eur J Pharmacol* 490, 127–133.
<https://doi.org/10.1016/j.ejphar.2004.02.050>.
 22. Byrne, B.J., Geberhiwot, T., Barshop, B.A., Barohn, R., Hughes, D., Bratkovic, D., Desnuelle, C., Laforet, P., Mengel, E., Roberts, M., et al. (2017). A study on the safety and efficacy of reveglucosidase alfa in patients with late-onset Pompe disease. *Orphanet J Rare Dis* 12. <https://doi.org/10.1186/S13023-017-0693-2>.
 23. Petrik, J., Pell, J.M., Arany, E., McDonald, T.J., Dean, W.L., Reik, W., and Hill, D.J. (1999). Overexpression of Insulin-Like Growth Factor-II in Transgenic Mice Is Associated with Pancreatic Islet Cell Hyperplasia. *Endocrinology* 140, 2353–2363.
<https://doi.org/10.1210/ENDO.140.5.6732>.
 24. Zhang, Q., Qin, S., Huai, J., Yang, H., and Wei, Y. (2023). Overexpression of IGF2 affects mouse weight and glycolipid metabolism and IGF2 is positively related to macrosomia. *Front Endocrinol (Lausanne)* 14, 1030453.
<https://doi.org/10.3389/FENDO.2023.1030453/BIBTEX>.
 25. Devedjian, J.-C., Gros, L., and Bosch, F. (2000). Transgenic mice overexpressing insulin-like growth factor-II in β cells develop type 2 diabetes. *J Clin Invest* 105.
<https://doi.org/10.1172/JCI5656>.
 26. Bruce, C.R., Hamley, S., Ang, T., Howlett, K.F., Shaw, C.S., and Kowalski, G.M. (2021). Translating glucose tolerance data from mice to humans: Insights from stable isotope labelled glucose tolerance tests. *Mol Metab* 53, 101281.
<https://doi.org/10.1016/J.MOLMET.2021.101281>.
 27. Colella, P., Sayana, R., Suarez-Nieto, M.V., Sarno, J., Nyame, K., Xiong, J., Pimentel Vera, L.N., Arozqueta Basurto, J., Corbo, M., Limaye, A., et al. (2024). CNS-wide repopulation by hematopoietic-derived microglia-like cells corrects progranulin deficiency in mice.

Nature Communications 2024 15:1 15, 1–26. <https://doi.org/10.1038/s41467-024-49908-4>.

7th Jul 2025

Decision on your manuscript EMM-2025-21220-V2

Dear Dr. Pijnappel,

Thank you for the submission of your manuscript to EMBO Molecular Medicine, and please accept my apologies for the unusual delay in getting back to you, which is due to the fact that one referee needed more time to complete his/her review. We have now received feedback from the two reviewers who agreed to re-evaluate your manuscript.

As you will see from their reports pasted below, referee #1 acknowledges the improvements of the revised manuscript but remains critical particularly regarding the points about the secretion of IDS.IGF and reduced IDS activity per VCN in bone marrow. Referee #3 recognizes the value and quality of the study but raises the question about the added value of SWAP tags considering the lack of in vivo effect of IDS.IGF on glucose levels. During our cross-commenting session both referees agreed that the study is not suitable for publication in EMBO Molecular Medicine.

From our side, we appreciate additional work done to address referees' criticism and we do recognize the potential interest of your findings; however, we agree with the referees and are not persuaded that your manuscript provides the sort of conceptual advance we would expect in an EMBO Molecular Medicine article. Therefore, I am afraid that we cannot offer to consider the manuscript further.

While we cannot pursue this manuscript further, we encourage you to transfer your study to our not-for-profit open-access sister journal, Life Science Alliance (LSA). We shared your manuscript and the accompanying reviews with LSA Executive Editor, Tim Fessenden, who is interested in these findings. He is pleased to offer publication of this manuscript at LSA pending the following revisions:

- Reduce the discussion as requested by Reviewer 1.
- Address the remaining request by Reviewer 1 related to previous point 7: Plot activity vs. VCN in bone marrow in Fig 3D using the hyperbolic relationship as noted for EV1 and EV3. Adjust claims on activity levels as needed to reflect these observations.

We encourage you to use the link below to transfer your manuscript to LSA. You do not need to revise the manuscript before transferring it to LSA. Once you transfer, Dr. Fessenden will email you an invitation to revise and resubmit, listing the same revision requests as mentioned above. Please feel free to reach out at t.fessenden@life-science-alliance.org if you have any questions about the LSA journal, the transfer process, or the revisions requested.

I understand that this is disappointing and regret that I could not bring better news this time. Please rest assured that this is not a judgment of the quality or interest of your work, but a decision based on appropriateness for EMBO Molecular Medicine. I hope you will view the encouragement of a transfer favorably.

Yours sincerely,

Zeljko Durdevic

Zeljko Durdevic
Senior Editor
EMBO Molecular Medicine

Referee #1 (Remarks for Author):

Overall the revised manuscript is much longer, less clear and the added text should be shortened. Points 6 secretion and 7 - BM enzyme vs VCN are not adequately addressed and need to be reworked. Points 1- 4, 8-13 have been addressed satisfactorily and minor points.

The response to points 5 and 15 which appears to be lines 549-612? is probably too long.

Point 6 - Although all points are captured, the response 576-612 is far too long and what has been added to the paper should be drastically shortened as it distracts and obscures the most likely explanation.

The most likely explanation for poor secretion seen in Figure 3G (lower plasma IDS activity per VCN with any tag) is that modification of IDS protein leads to reduced secretion in WBC, and this should be acknowledged. I appreciate that this is not seen in HMC3 cells, which is encouraging, but they do not reflect WBC populations. Indeed, some of the tags also appear to show lower protein production in BM per VCN (Figure 3C and 3D - and see point 7 below), which would tally with this. Amino acid mutations in proteins for example can result in misfolding, ubiquitination and degradation, remain blocked in the ER, have inappropriate secretion, amongst a myriad of other factors (e.g. Gerasimavicius 2022 Nat Comms). AlphaFold often has low structural confidence for end tagging approaches such as this.

Shorten the discussion drastically to note that secreted enzyme levels in some tags are lower than expected in WBC but not in HMC3, that this could be poor initial expression (BM) and/or secretion (WBC) from tagged products, or potentially endosomal recycling leading to re-secretion or capture at the receptor.

Point 7 - We and reviewer 3 noted that the IDS activity per VCN of BM of tagged proteins was lower than that of native coIDS (Fig 3G) and for intracellular comparing figure 3C to 3D. This is a serious concern as substantial differences in enzyme activity per VCN would mean that groups might not be as comparable and significantly detracts from the findings.

The authors claimed a hyperbolic relationship between enzyme activity and vector copy number and have added in new methods in lines 756-761. Indeed, the claim made to reviewer 3 is that saturation was reached at VCN 2 in Catalano2023. There is no evidence of saturation being reached at VCN of 2 in figure 3C or Fig 3G in this paper, thus this doesn't seem reasonable. Whilst it is well known that enzymes often have hyperbolic relationships with substrate, linear relationships between protein expression and VCN have been well documented in HSC gene therapy by multiple authors, perhaps most eloquently by Charrier 2010 Gene Therapy 18:479-487.

I find it hard to believe that a hyperbolic relationship exists between gene expression (VCN) and protein expression (activity) except in the case that the protein itself is toxic, which seems unlikely in this context.

The authors appear to have used a linear relationship between BM VCN and plasma enzyme activity (Figure 3D), but appear to have used a hyperbolic relationship in Expanded view figure EV1G and figure EV3D as stated in the new methods. The Y axis values in Figure EV3D do not correspond well to that achieved if one divides the average of 3D (BM IDS activity) by the average of 3C (BM VCN) and this skews the data in an unacceptable manner.

It would appear that IDS IGF2 and IDSSWAPAOEII will both be about half of IDS, IDSIGF2del and IDS SWAP Rap12, which would reduce the impact of the paper.

This graph should be presented as BM enzyme activity vs BM VCN, then a graph of BM activity per VCN should be presented and groups compared statistically.

A re-review to assess the claims would be needed, dependent on the outcome.

Referee #3 (Remarks for Author):

Following my review of the revised manuscript, I feel the authors have not addressed a fundamental issue (but only added more discussion) - namely, the lack of an in vivo effect of IDS.IGF2co on glucose levels in the mouse model. While the rationale is supported by human data, the model used does not replicate this problem (increased glucose levels), making it difficult to evaluate whether the new construct resolves it.

This limitation raises questions about the added value of SWAP tags. In my view, without a model that allows proper testing of the intended mechanism, the study does not provide sufficient evidence to support one of its main claims and, while recognising its value and quality, is not suitable for publication in EMBO Molecular Medicine.

Additionally, my requests to normalize VCN in Figure 1D and IDS activity in bone marrow (Figure 3C) were not addressed adequately, limiting interpretation of the data.

=====

As a service to authors, EMBO provides authors with the possibility to transfer a manuscript that one journal cannot offer to publish to another EMBO publication. The full manuscript and if applicable, reviewers reports are automatically sent to the receiving journal to allow for fast handling and a prompt decision on your manuscript. For more details of this service, and to transfer your manuscript to another EMBO title please click on Link Not Available

Dear Dr. Durdevic,

In response to our telephone call of July 23 in which we discussed the reviewers' comments to our manuscript EMM-2025-21220-V2, I would like to appeal to the decision that you mailed on July 7th 2025.

After the first reviewing round, we were pleased with the reviewers' comments, which were in general positive. As always with scientific progress, questions were raised but in this case all requests from the referees only concerned specific modifications of text and/or figures, except in one case, where reviewer 1 requested an additional experiment. In the resubmission, we have addressed all these points and we have performed and included the requested experiment. Therefore, we were surprised by the rather negative response to our resubmission, which may have been caused by a misinterpretation of our data. In addition, reviewer 3 requested new experimental evidence in reviewing round 2, which was not requested in reviewing round 1. Below, we outline this in a point-by-point manner.

Reviewer 1

The reviewer states that modification of IDS protein with SWAP tags leads to reduced secretion in WBCs.

In our opinion, this is an incorrect conclusion. The reviewer bases this suggestion on the reduced enzyme activity in plasma. However, plasma activity is the result of pharmacokinetics, which depends on many factors including uptake, clearance, and potentially re-secretion by the liver, uptake and re-secretion by other target cells and tissues, and binding to plasma proteins. It is well known that secretion and plasma activity are not coupled, this can also be seen for the previously published IDS-ApoE2 protein in lentiviral gene therapy (for example see Gleitz et al, EMBO Mol Med 2018 Jul;10(7):e8730, compare Fig 1D (showing similar secretion) with Fig 2D and 2J (showing very different plasma activities).

The reviewer states that there is lower protein production in Bone Marrow (BM) per VCN for SWAP tags, and argues for a linear relationship between VCN in BM and expression.

As per the reviewer's suggestion, we here provide 2 ways of correction BM activity for VCN: using a linear or a hyperbolic relationship. As can be seen from the figures, in both the linear and hyperbolic corrections, activities of IDS.SWAP proteins were not significantly different from the activity of untagged IDS or from IDS.IGF2 (which was the basis for designing the SWAP tags).

Figure 1. Linear (left panel) and hyperbolic (right panel; now Fig included as EV3E) normalization of IDS activity in bone marrow per VCN in bone marrow.

Reviewer 1 argues that we should have used a linear relationship between VCN and activity in BM for normalization. Above (Figure 1) we showed that it actually does not matter how you normalize (no significant differences), one could argue that using a linear relationship, there seems to be a trend toward lower expression of IDS.SWAP-ApoE, compared to untagged IDS, but not compared to IDS.IGF2. However, we argue against using a linear relationship between VCN and activity in BM, thereby largely removing the impression of the trend as well, based on our data (Figure 2, now included as Fig EV3D) and the literature: It has been shown by us and by colleagues in the field that upon higher VCNs, expression values of transgene proteins can saturate. The curves are dependent on the cell type analysed and likely on experimental conditions as well. We now provide a curve with all datapoints for all vectors that we collected so far, combining data from the current manuscript and data from Catalano et al 2023 (reference number 10 of the manuscript), which clearly show a hyperbolic relationship and saturation around VCN 2 under the conditions employed (Figure 2; now included as Fig EV3D).

Figure 2. Relationship between IDS activity in bone marrow and VCN in bone marrow.

In addition, several publications have reported a similar conclusion:

- 1) Larson et al, HUMAN VACCINES & IMMUNOTHERAPEUTICS 2017, VOL. 13, NO. 5, 1094–1104, Figure 1B:

- 2) Zielske et al, MOLECULAR THERAPY Vol. 9, No. 6, June 2004, p923:

“Our results indicate that transgene expression increased linearly up to a limited number of integrations but reached a plateau above that threshold.”

The publication cited by the reviewer (Charrier et al, Gene Therapy (2011) 18, 479–487 shows a correlation between VCN and GFP expression in only three individual clones (representing only 3 datapoints) of a human fibrosarcoma HT1080 cell line, which in our opinion is insufficient to conclude that there is a linear relationship between expression and VCN.

The point raised by the reviewer is crucial, because the reviewer states further that ‘This is a serious concern as substantial differences in enzyme activity per VCN would mean that groups might not be as comparable and significantly detracts from the findings.’ The reviewer addresses the expression data in BM and plasma, and request a re-evaluation performing statistical testing, which we have shown above and now include in the manuscript (using hyperbolic normalization). However, we respectfully disagree with the view to base the conclusion of the value of the SWAP tags on protein expression per VCN in BM only (despite the fact that these are not significantly different). This would ignore the in vivo data, which clearly demonstrate the efficacy of the SWAP tags per VCN in BM:

- 1) In the figure below (Figure 3, now included as main Fig 5B), we clearly demonstrate that **reduction of heparan sulfate in brain per VCN in bone marrow is similar for IDS.SWAP versions compared to IDS.IGF2 (and all are superior to untagged IDS).**

Figure 3. Relationship between heparan sulfate in brain and VCN in bone marrow.

- 2) In the original manuscript, we showed similar efficacies in brain of IDS-SWAP versions compared to IDS.IGF2 (all superior to untagged IDS) with respect to lysosomal pathology (LAMP1 expression, Fig 5E), inflammation (CD68 expression, Fig 6), and astrogliosis (GFAP expression, Fig 7). In these experiments, **mice were selected for histology based on similar VCNs in BM to allow for a fair comparison**, as was shown in Extended View Fig. 4J, K. **Perhaps this crucial information was missed by the reviewers.** In some cases (cortex, hippocampus), there was even a non-significant trend towards higher efficacy of SWAP-tagged IDS compared to IDS.IGF2.

We conclude from these data that, per VCNs in BM, SWAP-tagged IDS shows at least similar efficacy in vivo compared to IGF2-tagged IDS, and both SWAP and canonical IGF2-tagged IDS show superior efficacy compared to untagged IDS.

As per the reviewer's comments on the revised document, we have now strongly reduced the text where requested.

Reviewer 3.

The first point made by the reviewer in review round 2 appears to be a new requirement compared to review round 1. In round 1, the reviewer requested to 'tone down' or 'highlight or clarify' the relevance of the SWAP tags relative to the IGF2 tag, which we did in the modified text, but in round 2 the reviewer requested an experiment using a model that demonstrates the added value of the SWAP tag for maintaining glucose levels. To our knowledge, new requests are not allowed in a second review round.

The second point from reviewer 3 refers to the normalization of expression in BM per VCN; we have discussed this above.

Taken together, in our opinion our manuscript was rejected for publication based on a conclusion on the value of the SWAP tag that appears to be incorrect, and based on a new requirement for experimental testing that is not allowed. For these reasons, and based on the rather positive reviews in round 1, we are asking you to reconsider your decision to reject our manuscript for publication. We send you along with this letter our revised manuscript in which we addressed all the reviewers original requests and in which we clarified the misunderstandings. Revised figures are sent via surffilesender.

Yours sincerely,

Prof. Dr. W.W.M. Pim Pijnappel.

25th Aug 2025

Dear Dr. Pijnappel,

Thank you for submitting your manuscript to EMBO Molecular Medicine. I have now carefully read your manuscript, point-by-point response to the referees' comments and discussed it with the other members of our editorial team. In addition, I have sought external advice on the study from an expert in the field. Our advisor reached a conclusion that the study is suitable for publication in EMBO Molecular Medicine. Therefore, I am pleased to inform you that we will be able to accept your manuscript pending the following final amendments:

1) Figures:

- Please remove text "Figure 1; Catalano et al., 2025" etc. and "Expanded View Figure 1; Catalano et al, 2025" etc. from figure and EV figure files.
- We note that western blots in Figure EV1D are over-contrasted. Please provide source data and replace western blots with less processed images.

2) In the main manuscript file, please do the following:

- Please address all comments suggested by our data editors listed below:

o Figure legends:

1. Please note that the exact p values are not provided in the legends of figures 1C, D; 3C-F; 4B, E, F; 5A, B, E; 6, 7; EV1 C, G; EV3 B, C, F, G.
2. Please indicate the statistical test used for data analysis in the legends of figures EV3 B, C, F, G.
3. Please note that information related to n is missing in the legends of figures EV2 A, B; EV4 B, C, D, E, H, I, J.
4. Please note that the scale bar needs to be defined for figures 5E, 6.
5. Please note that the dotted borders are not defined in the legend of figure 7. This needs to be rectified.

- Limit keywords to max. 5 and place it after Abstract.

- Indicate in legends exact n and exact p values, not a range, along with the statistical test used. To keep the figures "clear" some authors found providing an Appendix table Sx with all exact p-values preferable. You are welcome to do this if you want to.
- Please remove Reagents and Tools Table from the manuscript file and upload it as a separate file. More information on how to adhere to this format as well as downloadable templates (.docx) for the Reagents and Tools Table can be found in our author guidelines:

<https://www.embopress.org/page/journal/17574684/authorguide#structuredmethods>

An example of a paper with Structured Methods can be found here:

<https://www.embopress.org/doi/full/10.1038/s44320-024-00037-6#sec-4>

- Rename "Conflict of interests" to "Disclosure Statement & Competing Interests". We updated our journal's competing interests policy in January 2022 and request authors to consider both actual and perceived competing interests. Please review the policy <https://www.embopress.org/competing-interests> and update your competing interests if necessary.

- Author contributions: Please remove it from the manuscript and specify author contributions in our submission system. CRediT has replaced the traditional author contributions section because it offers a systematic machine-readable author contributions format that allows for more effective research assessment. You are encouraged to use the free text boxes beneath each contributing author's name to add specific details on the author's contribution. More information is available in our guide to authors:

<https://www.embopress.org/page/journal/17574684/authorguide#authorshipguidelines>

- In data availability statement replace the current text with "This study includes no data deposited in external repositories".
- Please correct the reference citation in the text and reference list. In the text a reference should be cited by author and year of publication. Include a space between a word and the opening parenthesis of the reference that follows. In the reference list, citations should be listed in alphabetical order. Where there are more than 10 authors on a paper, 10 will be listed, followed by "et al.". Also, please remove DOIs. DOIs should only be used for preprints and datasets that have not been published. Please check "Author Guidelines" for more information.

<https://www.embopress.org/page/journal/17574684/authorguide#referencesformat>

3) Tables: Please upload EV tables as excel files and add their legends to the files.

4) Source Data: Please upload one zipped file per figure.

5) The Paper Explained: Please provide "The Paper Explained" and add it to the main manuscript text. Please check "Author Guidelines" for more information. <https://www.embopress.org/page/journal/17574684/authorguide#researcharticleguide>

6) Synopsis:

- Synopsis image: Please resize the image to 550 pixels wide x 300-600 pixels high and upload it as a high-resolution jpeg file
- Please check your synopsis text and image before submission with your revised manuscript. Please be aware that in the proof stage minor corrections only are allowed (e.g., typos).

7) As part of the EMBO Publications transparent editorial process initiative (see our Editorial at

<http://embomolmed.embopress.org/content/2/9/329>), EMBO Molecular Medicine will publish online a Review Process File (RPF) to accompany accepted manuscripts. This file will be published in conjunction with your paper and will include the anonymous referee reports, your point-by-point response and all pertinent correspondence relating to the manuscript. Let us know whether you agree with the publication of the RPF and as here, if you want to remove or not any figures from it prior to publication.

8) Please provide a point-by-point letter INCLUDING my comments as well as the reviewer's reports and your detailed responses (as Word file).

I look forward to reading a new revised version of your manuscript as soon as possible.

Yours sincerely,

Zeljko Durdevic

Zeljko Durdevic
Senior Editor
EMBO Molecular Medicine

*** Instructions to submit your revised manuscript ***

To submit your manuscript, please follow this link:

<https://embomolmed.msubmit.net/cgi-bin/main.plex>

- 1) a .docx formatted version of the manuscript text (including Figure legends and tables)
- 2) Separate figure files*
- 3) supplemental information as Expanded View and/or Appendix. Please carefully check the authors guidelines for formatting Expanded view and Appendix figures and tables at <https://www.embopress.org/page/journal/17574684/authorguide#expandedview>
- 4) a letter INCLUDING the reviewer's reports and your detailed responses to their comments (as Word file).
- 5) The paper explained: EMBO Molecular Medicine articles are accompanied by a summary of the articles to emphasize the major findings in the paper and their medical implications for the non-specialist reader. Please provide a draft summary of your article highlighting
 - the medical issue you are addressing,
 - the results obtained and
 - their clinical impact.This may be edited to ensure that readers understand the significance and context of the research. Please refer to any of our published articles for an example.

6) Author contributions: the contribution of every author must be detailed in a separate section.

7) EMBO Molecular Medicine now requires a complete author checklist (<https://www.embopress.org/page/journal/17574684/authorguide>) to be submitted with all revised manuscripts. Please use the checklist as guideline for the sort of information we need WITHIN the manuscript. The checklist should only be filled with page numbers where the information can be found. This is particularly important for animal reporting, antibody dilutions (missing) and exact values and n that should be indicated instead of a range.

8) Every published paper now includes a 'Synopsis' to further enhance discoverability. Synopses are displayed on the journal webpage and are freely accessible to all readers. They include a short stand first (maximum of 300 characters, including space) as well as 2-5 one sentence bullet points that summarise the paper. Please write the bullet points to summarise the key NEW findings. They should be designed to be complementary to the abstract - i.e. not repeat the same text. We encourage inclusion of key acronyms and quantitative information (maximum of 30 words / bullet point). Please use the passive voice. Please attach these in a separate file or send them by email, we will incorporate them accordingly.

You are also welcome to suggest a striking image or visual abstract to illustrate your article. If you do please provide a jpeg file 550 px-wide x 300-600px high.

9) A Conflict of Interest statement should be provided in the main text

10) Please note that we now mandate that all corresponding authors list an ORCID digital identifier. This takes <90 seconds to complete. We encourage all authors to supply an ORCID identifier, which will be linked to their name for unambiguous name identification.

Currently, our records indicate that the ORCID for your account is 0000-0002-7042-2482.

Link Not Available

11) Include a Reagents and Tools Table as part of the Methods section, which can be downloaded from our author guidelines (<https://www.embopress.org/page/journal/17574684/authorguide#structuredmethods>)

Photos 400-800 DPI

*Additional important information regarding figures and illustrations can be found at

<https://bit.ly/EMBOPressFigurePreparationGuideline>. See also figure legend preparation guidelines:

<https://www.embopress.org/page/journal/17574684/authorguide#figureformat>

The authors addressed the remaining editorial issues.

16th Sep 2025

Dear Dr. Pijnappel,

We are pleased to inform you that your manuscript is accepted for publication and is now being sent to our publisher to be included in the next available issue of EMBO Molecular Medicine.

Zeljko Durdevic
Senior Editor
EMBO Molecular Medicine
